# Probabilistic Pretraining for Improved Neural Regression

**Boris N. Oreshkin**                                    *oreshkin@amazon.com*
*SCOT Forecasting*
*Amazon Science*

**Shiv Tavker**                                          *tavker@amazon.com*
*Pricing and Promotions*
*Amazon Science*

**Dmitry Efimov**                                        *defimov@amazon.com*
*SCOT Forecasting*
*Amazon Science*

**Reviewed on OpenReview:** *https://openreview.net/forum?id=F6BTATGXaf*

## Abstract

While transfer learning has revolutionized computer vision and natural language processing, its application to probabilistic regression remains underexplored, particularly for tabular data. We introduce NIAQUE (Neural Interpretable Any-Quantile Estimation), a novel permutation-invariant architecture that enables effective transfer learning across diverse regression tasks. Through extensive experiments on 101 datasets, we demonstrate that pre-training NIAQUE on multiple datasets and fine-tuning on target datasets consistently outperforms both traditional tree-based models and transformer-based neural baseline. On real-world Kaggle competitions, NIAQUE achieves competitive performance against heavily hand-crafted and feature-engineered solutions and outperforms strong baselines such as TabPFN and TabDPT, while maintaining interpretability through its probabilistic framework. Our results establish NIAQUE as a robust and scalable approach for tabular regression, effectively bridging the gap between traditional methods and modern transfer learning.

## 1 Introduction

Tabular data underpins high-stakes decision-making across healthcare (Rajkomar et al., 2018), real estate valuation (De Cock, 2011), energy systems (Olson et al., 2017a), and e-commerce (McAuley et al., 2015). In these settings, models must be accurate, data-efficient and, crucially for operational use, able to quantify predictive uncertainty. Historically, tree-based ensembles such as Random Forests (Breiman, 2001), XG-Boost (Chen & Guestrin, 2016), LightGBM (Ke et al., 2017), and CatBoost (Prokhorenkova et al., 2019) have been the de facto standard due to their robustness, ease of use, and strong performance with heterogeneous features and modest sample sizes. Recent work has revisited deep learning for tabular data, motivated by the promise of end-to-end representation learning, joint optimization with downstream objectives, and straightforward multimodal fusion. Architectures such as TabNet (Arik & Pfister, 2021) and TabTransformer (Huang et al., 2021) demonstrate that appropriately constrained inductive biases (e.g., learned feature selection, attention over categorical embeddings) can close the gap to boosted trees on many benchmarks while enabling capabilities such as differentiable feature learning and integration with text or images. Yet, unlike computer vision and natural language processing, where transfer learning via large-scale pretraining across heterogenous real datasets is now foundational, the corresponding paradigm for tabular data remains comparatively under-explored, particularly for *probabilistic regression* tasks that require calibrated predictive distributions rather than point estimates (Hollmann et al., 2023; Levin et al., 2023).

**Challanges.** This work addresses several fundamental challenges in tabular regression. *Transfer Learning Gap:* While recent work has explored transfer learning for tabular classification, probabilistic regression remains underexplored, lacking frameworks that effectively transfer knowledge across diverse regression tasks. *Scalability–Interpretability Trade-off:* Existing approaches often sacrifice either scalability or interpretability, limiting their practical utility in real-world applications requiring both robust performance and explainable predictions. *Benchmark Limitations:* Current multi-dataset tabular benchmarks predominantly focus on classification tasks, hindering systematic evaluation of regression models.

**Our approach: NIAQUE.** We introduce *NIAQUE* (Neural Interpretable Any-Quantile Estimation), a probabilistic regression model designed to resolve these challenges. First, NIAQUE enables effective *co-training across multiple disjoint datasets*, exhibiting positive transfer and strong scalability when fine-tuned on new regression tasks. Second, its probabilistic framework supports *interpretable* analysis via marginal posterior distributions that yield feature-level importance summaries, enhancing transparency and reliability in decision-critical settings. Third, we provide *theoretical guarantees* showing that, under appropriate regularity conditions, NIAQUE approximates the inverse of the posterior distribution, thereby formalizing its ability to recover task-conditional quantiles. To enable comprehensive assessment, we introduce a new multi-dataset regression benchmark comprising *101* diverse datasets spanning multiple domains. We evaluate with proper scoring rules (e.g., CRPS), calibration diagnostics, and standard accuracy metrics. Across this suite, NIAQUE outperforms strong tree-based baselines and neural approaches while maintaining interpretability. Furthermore, on real-world Kaggle regression challenges, NIAQUE achieves competitive performance against highly engineered solutions. Therefore, our primary contributions are as follows. (1) We propose a novel deep probabilistic regression model (**NIAQUE**) that enables effective transfer learning across diverse tabular datasets via co-training and task-adaptation. (2) We provide theoretical analysis establishing NIAQUE's convergence to inverse posterior distribution. (3) We demonstrate superior performance over strong baselines, both boosted trees and modern neural methods, in transfer learning settings on a new benchmark suite for tabular regression (101 datasets).

## 2 Related Work

**Probabilistic Regression.** This work builds on probabilistic time-series modeling approach (Smyl et al., 2024), refining its theoretical underpinnings and extending architectural design for tabular transfer learning applications. Alternative methods, such as Neural Processes (Garnelo et al., 2018b) and Conditional Neural Processes (Garnelo et al., 2018a), offer conditional probabilistic solutions to regression but are constrained to fixed-dimensional input spaces, limiting their applicability to cross-dataset, multi-task regression. Our approach effectively transfers knowledge across datasets with varying feature spaces and target domains, establishing a flexible and scalable framework for conditional probabilistic regression.

**Transfer Learning in Tabular Data.** Transfer learning has driven major advances in computer vision (Sun et al., 2021; Radford et al., 2021) and language modeling (Devlin et al., 2019) by leveraging shared representations across tasks. Recent successes in time-series forecasting (Garza & Mergenthaler-Canseco, 2023; Ansari et al., 2024) demonstrate transfer learning's potential for numerical prediction tasks. However, probabilistic transfer learning for tabular data remains largely unexplored, with existing work primarily focusing on classification tasks or point estimates. Our work bridges this gap by introducing a framework specifically designed for probabilistic transfer across diverse tabular datasets.

**Deep Learning vs. Tree-based Models.** The comparison between deep learning and tree-based approaches for tabular data has been extensively studied, particularly for classification tasks. Previous evaluations include: transformer architectures across 20 classification datasets, MLPs versus TabNet and tree-based models on 40 classification datasets, Comprehensive comparison of various architectures across 45 datasets (Grinsztajn et al., 2022). Most relevant to our work are TabPFN (Hollmann et al., 2023) and TabDPT (Ma et al., 2024), which introduce Transformer-based approaches for synthetic pre-training. While these methods share similar goals, our work differs in three key aspects. First, we propose a novel architectural approach based on deep prototype aggregation, which scales linearly in the number of input features. Second, we focus specifically on probabilistic pretraining on real public datasets and transfer learning. Finally, we demonstrate

superior empirical accuracy on real-world Kaggle competitions through direct probabilistic pretraining and fine-tuning on downstream regression tasks.

**Permutation-invariant Architectures.** Our architectural design builds upon advances in permutation-invariant representations, crucial for handling variable feature spaces in multi-task learning. We extend the prototype-based architecture proposed by (Oreshkin et al., 2022) for pose completion to support any-quantile modeling in tabular regression. This approach relates to several key developments in the field. PointNet (Qi et al., 2017) and DeepSets (Zaheer et al., 2017) pioneered pooling techniques for variable-dimensional inputs. ResPointNet (Niemeyer et al., 2019) generalized these approaches with residual connections. Prototypical Networks (Snell et al., 2017) demonstrated effectiveness of prototype-based learning by leveraging average-pooled embeddings for few-shot classification. Transformer architecture (Vaswani et al., 2017) established flexible processing of variable-length sequences.

## 3 Background and Problem Formulation

Let $\mathbb{R}$ denote the set of real numbers and $\mathcal{U}(0,1)$ the uniform distribution over the interval $(0,1)$. For a vector $\mathbf{x}$, we denote its dimensionality as $|\mathbf{x}|$. For a random variable $Y$ with cumulative distribution function (CDF) $F(y) = P(Y \leq y)$, the $q$-th quantile $q \in (0,1)$ is defined as:

$$F^{-1}(q) = \inf\{y \in \mathbb{R} : F(y) \geq q\}.$$

### 3.1 Problem Formulation

Let $\mathcal{X}$ be the input feature space and $\mathcal{Y} \subseteq \mathbb{R}$ be the space of the target variable. We consider a probability distribution $\mathcal{D}$ over $\mathcal{X} \times \mathcal{Y}$. For any instance $\mathbf{x} \in \mathcal{X}$, the relationship between features and target variable is given by:

$$y = \Psi(\mathbf{x}, \varepsilon) \tag{1}$$

where $\Psi : \mathcal{X} \times \mathcal{E} \to \mathcal{Y}$ is an unknown non-linear function and $\varepsilon \in \mathcal{E}$ represents stochastic noise with unknown distribution.

Given a finite training sample $S = \{(\mathbf{x}_i, y_i)\}_{i=1}^{N}$ drawn i.i.d. from $\mathcal{D}$, we aim to learn a probabilistic regression function $f_\theta : \mathbb{R}^{|\mathbf{x}| \times Q} \to \mathbb{R}^Q$, parameterized by $\theta \in \Theta$, which maps an input $\mathbf{x}$ to a $Q$-tuple of quantiles $(q_1, ..., q_Q)$, where $q_i \in (0,1)$ for $i \in [Q]$, thereby capturing the conditional distribution of $y|\mathbf{x}$.

### 3.2 Performance Metrics

Let $y_i$ denote the ground truth sample and $\hat{y}_{i,q}$ its $q$-th quantile prediction for a dataset with $S$ samples. To evaluate the quality of distributional predictions, we use Continuous Ranked Probability Score (CRPS). The theoretical definition of CRPS for a predicted cumulative distribution function $F$ and observation $y$ is:

$$\mathrm{CRPS}(F, y) = \int_{\mathbb{R}} \left( F(z) - \mathbb{1}_{\{z \geq y\}} \right)^2 dz, \tag{2}$$

where $F : \mathbb{R} \to [0,1]$ is the predicted CDF derived from the quantile predictions, and $\mathbb{1}_{\{z \geq y\}}$ is the indicator function. For practical computation with finite samples $S$ and a discrete set of $Q$ quantiles, we approximate this using:

$$\mathrm{CRPS} = \frac{2}{SQ} \sum_{i=1}^{S} \sum_{j=1}^{Q} \rho(y_i, \hat{y}_{i,q_j}) \tag{3}$$

where $\rho(y, \hat{y}_q)$ is the quantile loss function defined as:

$$\rho(y, \hat{y}_q) = (y - \hat{y}_q)(q - \mathbb{1}_{\{y \leq \hat{y}_q\}}) \tag{4}$$

Coverage measures empirical calibration of predictive confidence intervals. For confidence level $\alpha \in (0, 1)$:

$$\text{COVERAGE}(\alpha) = \frac{1}{S} \sum_{i=1}^{S} \mathbb{1}[y_i > \hat{y}_{i,0.5-\alpha/2}] \mathbb{1}[y_i < \hat{y}_{i,0.5+\alpha/2}] \cdot 100\%$$

We employ the following metrics to evaluate the accuracy of point predictions: Symmetric Mean Absolute Percentage Error (sMAPE), Average Absolute Deviation (AAD), Root Mean Square Error (RMSE), Root Mean Square Logarithmic Error (RMSLE). Detailed mathematical definitions of the metrics are provided in Appendix A.

## 4 NIAQUE

In this section, we present NIAQUE (Neural Interpretable Any-Quantile Estimation), a probabilistic regression model. We first introduce the any-quantile learning approach as a general solution to the probabilistic regression problem defined in Section 3. We prove that this approach converges to the inverse cumulative distribution function of the conditional distribution, providing a theoretical foundation for our method. We then detail NIAQUE's neural architecture, demonstrate how it enables transfer learning across diverse tabular datasets, and present an approach to model interpretability based on probabilistic considerations.

### 4.1 Any-Quantile Learning

We formulate the any-quantile learning approach by augmenting the input space to include a quantile level $q \in (0, 1)$, allowing the neural network $f_\theta$ to learn mappings from $(\mathbf{x}, q)$ to the corresponding $q$-th conditional quantile of the target variable $y|\mathbf{x}$. Let $\hat{y}_q = f_\theta(\mathbf{x}, q)$ represent the predicted $q$-th quantile of the conditional distribution of $y|\mathbf{x}$. The objective is to learn parameters $\theta$ that minimize the expected quantile loss:

$$\min_\theta \mathbb{E}_{(\mathbf{x},y)\sim\mathcal{D}, q\sim\mathcal{U}(0,1)}[\rho(y, f_\theta(\mathbf{x}, q))], \tag{5}$$

where $\rho(\cdot, \cdot)$ is the quantile loss function defined in eq. (4).

We use gradient descent and mini-batch to learn the parameters. Precisely, the neural network is trained on dataset of $S$ samples, $(\mathbf{x}_i, y_i)$ drawn from the joint probability distribution $\mathcal{D}$. During training the quantile value $q$ is sampled from $\mathcal{U}(0, 1)$ and the loss is minimized using stochastic gradient descent (SGD). For a mini-batch of size $B$, the parameter update at iteration $k$ is:

$$\theta_{k+1} = \theta_k - \eta_k \nabla_\theta \frac{1}{B} \sum_{i=1}^{B} \rho(y_i, f_\theta(\mathbf{x}_i, q_i)). \tag{6}$$

As $k \to \infty$, the parameters converge to the solution of the following empirical risk minimization problem (Karimi et al., 2016):

$$\theta^* = \arg\min_{\theta\in\Theta} \frac{1}{S} \sum_{i=1}^{S} \rho(y_i, f_\theta(\mathbf{x}_i, q_i)). \tag{7}$$

By the strong law of large numbers, as $S$ grows, the empirical risk converges to the expected quantile loss:

$$\mathbb{E}_{\mathbf{x},y}\mathbb{E}_q \rho(y, f_\theta(\mathbf{x}, q)) = \mathbb{E}_{\mathbf{x},y} \int_0^1 \rho(y, f_\theta(\mathbf{x}, q))dq. \tag{8}$$

This expected loss has a direct connection to the Continuous Ranked Probability Score (CRPS), which can be expressed as an integral over quantile loss (Gneiting & Ranjan, 2011):

$$\text{CRPS}(F, y) = 2 \int_0^1 \rho(y, F^{-1}(q))dq. \tag{9}$$

Based on this fact, the following theorem proves that the expected pinball loss eq. (8) is minimized when $f_\theta(\mathbf{x}, q)$ corresponds to the inverse of the posterior CDF $P_{y|\mathbf{x}}$.

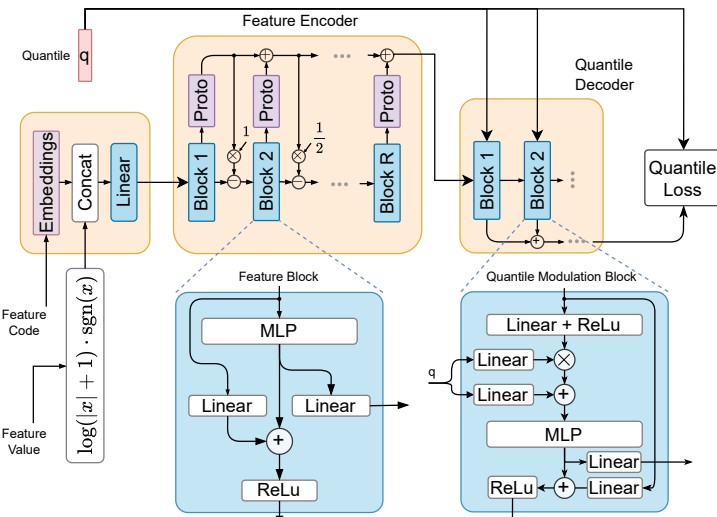

Figure 1: NIAQUE's encoder-decoder architecture transforms variable-dimensional inputs into fixed-size representations, enabling transfer learning and multi-task knowledge sharing across datasets.

**Theorem 1.** *Let $F$ be a probability measure over variable $y$ such that inverse $F^{-1}$ exists and let $P_{y,\mathbf{x}}$ be the joint probability measure of variables $\mathbf{x}, y$. Then the expected loss, $\mathbb{E}_{\mathbf{x},y,q}\,\rho(y, F^{-1}(q))$, is minimized if and only if $F = P_{y|\mathbf{x}}$.*

The following conclusions emerge. First, the quantile loss SGD update eq. (6) optimizes the empirical risk eq. (7) corresponding to the expected loss eq. (8). Based on (8,9) and Theorem 1, $f_{\theta^\star} = \arg\min_{f_\theta} \mathbb{E}_{\mathbf{x},y,q}\,\rho(y, f_\theta(\mathbf{x}, q))$ has a clear interpretation as the inverse CDF corresponding to $P_{y|\mathbf{x}}$. Second, as both the SGD iteration index $k$ and training sample size $S$ increase, and if $f_\theta$ is implemented as an MLP whose width and depth scale appropriately with sample size $S$, then (Farrell et al., 2021, Theorem 1) implies that the SGD solution converges to $f_{\theta^\star}(\mathbf{x}, q) \equiv P_{y|\mathbf{x}}^{-1}(q)$. Therefore, given uniform $q \sim \mathcal{U}(0,1)$, $\widehat{y}_q = f_{\theta^\star}(\mathbf{x}, q)$ has the interpretation of a sample from the posterior distribution $p(y|\mathbf{x})$, which follows from the proof of the inversion method (Devroye, 1986, Theorem 2.1).

### 4.2 Neural Encoder-Decoder Architecture

NIAQUE adopts a modular encoder-decoder design (Fig. 1) to process observation samples $\mathbf{x}_i$ with variable dimensionality $d_i$. The encoder maps each observation into a fixed-size latent embedding of dimension $E$, enabling downstream processing independent of input dimensionality. Feature values and associated codes (IDs) (dimension $1 \times d_i$) are embedded into a tensor of size $1 \times d_i \times E_{in}$, where $E_{in}$ is the embedding size per feature. These embeddings are aggregated using a prototype-based method to generate the latent observation representation. The decoder conditions this representation on arbitrary-length quantile vectors $\mathbf{q} \in \mathbb{R}^Q$, modulating the output using FiLM-based transformations (Perez et al., 2018). This separates input processing and quantile conditioning, achieving computational efficiency of $O(d_i + Q)$ per sample $\mathbf{x}_i$, compared to $O(d_i Q)$ complexity required to process quantiles and observations jointly. Note that our processing is linear if feature dimensionality, as opposed to quadratic scaling of attention-based approaches.

**Inputs:** NIAQUE processes both continuous and categorical features in a unified manner. Categorical features are first label-encoded to integers during preprocessing. For each feature in the observation vector $\mathbf{x}$, NIAQUE incorporates both its raw value and a learnable embedding based on its feature index (position in the feature vector). The feature index embedding learns feature-specific statistical properties, inter-feature dependencies, and their relationship with the target, enabling the model to distinguish between features even when they have similar numerical values. The embedded feature index is concatenated with the feature's

value after log-transformation:

$$z = \log(|x| + 1) \cdot \mathrm{sgn}(x), \tag{10}$$

which normalizes the features' dynamic range (including label-encoded categorical values), aligning it with that of index embeddings while preserving sign information, facilitating stable training, as validated by the ablation study in Appendix J.

**Feature Encoder:** The encoder employs a two-loop residual network architecture to efficiently handle variable-dimensional inputs. The following equations define the encoder's transformations, with the sample index $i$ omitted for brevity. Let the encoder input be $\mathbf{x}_{in} \in \mathbb{R}^{d \times E_{in}}$, where $d$ is the number of features and $E_{in}$ is the embedding size per feature. A fully-connected layer $\mathrm{FC}_{r,\ell}$ in residual block $r \in \{1, \ldots, R\}$, layer $\ell \in \{1, \ldots, L\}$, with weights $\mathbf{W}_{r,\ell}$ and biases $\mathbf{a}_{r,\ell}$, is defined as:

$$\mathrm{FC}_{r,\ell}(\mathbf{h}_{r,\ell-1}) \equiv \mathrm{RELU}(\mathbf{W}_{r,\ell}\mathbf{h}_{r,\ell-1} + \mathbf{a}_{r,\ell}).$$

We also define a prototype layer as: $\mathrm{PROTOTYPE}(\mathbf{x}) \equiv \frac{1}{d}\sum_{i=1}^{d} \mathbf{x}[i,:]$. The observation encoder is then described by the following equations:

$$\mathbf{x}_r = \mathrm{RELU}(\mathbf{b}_{r-1} - 1/(r-1) \cdot \mathbf{p}_{r-1}), \tag{11}$$

$$\mathbf{h}_{r,1} = \mathrm{FC}_{r,1}(\mathbf{x}_r), \ \ldots, \ \mathbf{h}_{r,L} = \mathrm{FC}_{r,L}(\mathbf{h}_{r,L-1}), \tag{12}$$

$$\mathbf{b}_r = \mathrm{RELU}(\mathbf{L}_r\mathbf{x}_r + \mathbf{h}_{r,L}), \ \mathbf{f}_r = \mathbf{F}_r\mathbf{h}_{r,L}, \tag{13}$$

$$\mathbf{p}_r = \mathbf{p}_{r-1} + \mathrm{PROTOTYPE}(\mathbf{f}_r). \tag{14}$$

These equations implement a dual-residual mechanism. First, Equations (12) and (13) form an MLP with a residual connection (see Feature Block in Fig. 1 bottom left). Second, Equations (11) and (14) form a second residual loop with the following key properties: a) Eq. (14) consolidates individual feature encodings into a prototype-based representation of the observation; b) Eq. (11) implements interactions between features (akin to attention, but with linear compute cost) and introduces an inductive bias by enforcing a delta-mode constraint, ensuring that feature contributions are only relevant when they deviate from the existing observation embedding, $\mathbf{p}_{r-1}$; c) The observation representation accumulates across residual blocks eq. (14), effectively implementing skip connections.

**Quantile Decoder:** The decoder implements a fully-connected conditioned residual architecture (Fig. 1, top-right). Its primary function is to implement the any-quantile functionality by injecting the quantile value inside the MLP block using FiLM modulation principle (Perez et al., 2018). Taking the observation embedding $\widetilde{\mathbf{b}}_0 = \mathbf{p}_R \in \mathbb{R}^E$ as input, it generates quantile-modulated representations $\widetilde{\mathbf{f}}_R \in \mathbb{R}^{Q \times E}$ for quantiles $\mathbf{q} \in \mathbb{R}^Q$ through:

$$\begin{aligned}
\mathbf{h}_{r,1} &= \mathrm{FC}_{r,1}^{\mathrm{QD}}(\widetilde{\mathbf{b}}_{r-1}), \quad \gamma_r, \beta_r = \mathrm{LINEAR}_r(\mathbf{q}), \\
\mathbf{h}_{r,2} &= \mathrm{FC}_{r,1}^{\mathrm{QD}}((1 + \gamma_r) \cdot \mathbf{h}_{r,1} + \beta_r), \\
&\ \ldots \\
\mathbf{h}_{r,L} &= \mathrm{FC}_{r,L}^{\mathrm{QD}}(\mathbf{h}_{r,L-1}), \\
\widetilde{\mathbf{b}}_r &= \mathrm{RELU}(\mathbf{L}_r^{\mathrm{QD}}\widetilde{\mathbf{b}}_{r-1} + \mathbf{h}_{r,L}), \ \ \widetilde{\mathbf{f}}_r = \widetilde{\mathbf{f}}_{r-1} + \mathbf{F}_r^{\mathrm{QD}}\mathbf{h}_{r,L}.
\end{aligned} \tag{15}$$

The final prediction $\widehat{\mathbf{y}}_q \in \mathbb{R}^Q$ is obtained via linear projection: $\widehat{\mathbf{y}}_q = \mathrm{LINEAR}[\widetilde{\mathbf{f}}_r]$.

### 4.3 Interpretability

NIAQUE's probabilistic framework facilitates interpretability via quantile predictions conditioned on individual features. Given $f_\theta(\mathbf{x}_s, q)$ as NIAQUE's estimate of quantile $q$ using only feature $\mathbf{x}_s$, the posterior confidence interval for this feature is defined as:

$$\mathrm{CI}_{\alpha,s} = f_\theta(\mathbf{x}_s, 1 - \alpha/2) - f_\theta(\mathbf{x}_s, \alpha/2), \tag{16}$$

where $1 - \alpha$ represents the probability that the target true value lies within the interval. Intuitively, more informative features produce narrower confidence intervals. We leverage this to quantify feature importance through normalized weights:

$$W_s = \frac{\overline{W}_s}{\sum_s \overline{W}_s}, \quad \overline{W}_s = \frac{1}{\overline{\text{CI}}_{0.95,s}}, \quad \overline{\text{CI}}_{\alpha,s} = \frac{1}{S} \sum_i f_\theta(\mathbf{x}_{s,i}, 1 - \alpha/2) - f_\theta(\mathbf{x}_{s,i}, \alpha/2). \tag{17}$$

where $\overline{\text{CI}}_{\alpha,s}$ is the average confidence interval width over validation samples $\{\mathbf{x}_i : \mathbf{x}_i \in \mathcal{D}_{\text{val}}\}$. To enhance marginal distribution modeling and support interpretability, we introduce single-feature samples during training, comprising approximately 5% of the dataset. An ablation study in Appendix J confirms the necessity of this augmentation for robust feature importance estimation.

## 4.4 Transfer Learning

NIAQUE facilitates effective transfer learning through two core mechanisms ensuring effective knowledge transfer across diverse tabular regression tasks with varying feature spaces. First, its feature ID embeddings learn both dataset-specific and cross-dataset relationships, as evidenced by the structured representation space observed in Fig. 3. Second, its prototype-based aggregation enables flexible processing of arbitrary feature combinations, inherently supporting cross-dataset learning.

The transfer learning process consists of two phases. During pretraining, NIAQUE learns from multiple heterogeneous datasets simultaneously, sampling rows uniformly at random from all datasets, with each dataset contributing only its relevant features. The model processes these diverse inputs through shared parameters, learning both task-specific characteristics and generalizable patterns. The learned embeddings capture statistical properties at multiple levels: individual feature distributions, feature interactions within datasets, and common patterns across different regression problems. This pretrained knowledge can then be leveraged in two ways. First, transfer across seen datasets, in which co-training on multiple datasets enables the learning of larger-capacity networks using data diversity as a regularizer, while also leveraging commonalities across classes of problems—implicitly extracted during training—to strengthen performance on related seen tasks. Second, few-shot transfer to unseen datasets through fine-tuning, where the pretrained feature representations provide a strong foundation for learning new tasks with minimal data via strong regression prior stored in pretrained model weights. In both cases, the model learns to distinguish and process dataset-specific features through their semantic embeddings, whereas shared model parameters enable knowledge transfer across related datasets via multi-task training.

The effectiveness of this approach stems from NIAQUE's ability to maintain dataset-specific information while learning transferable representations. Learnable feature ID embeddings act as task identifiers, allowing the model to adapt its processing based on the combination of input features, effectively serving as an implicit task ID. This capability is particularly valuable in tabular domains, where feature relationships and their predictive power can vary significantly across tasks. Our experimental results validate both the efficacy of cross-dataset pretraining and the model's adaptability to novel regression tasks via fine-tuning, establishing NIAQUE as a robust framework for transfer learning in tabular domains.

## 5 Empirical Results

We conduct extensive experiments to evaluate NIAQUE's effectiveness for transfer learning in tabular regression. Our evaluation addresses three key aspects: 1) the model's transfer learning capabilities across both seen and unseen tasks, 2) the quality of learned representations and interpretability, and 3) its practical effectiveness in a real-world competition setting.

### 5.1 Datasets and Experimental Setup

**Dataset and Evaluation:** We introduce TabRegSet-101 (Tabular Regression Set 101), a curated collection of 101 publicly available regression datasets gathered from UCI (Kelly et al., 2017), Kaggle (Kaggle, 2024), PMLB (Romano et al., 2021; Olson et al., 2017b), OpenML (Vanschoren et al., 2013), and KEEL (Alcalá-Fdez et al., 2011). These datasets span diverse domains, including *Housing and Real Estate, Energy and Efficiency,*

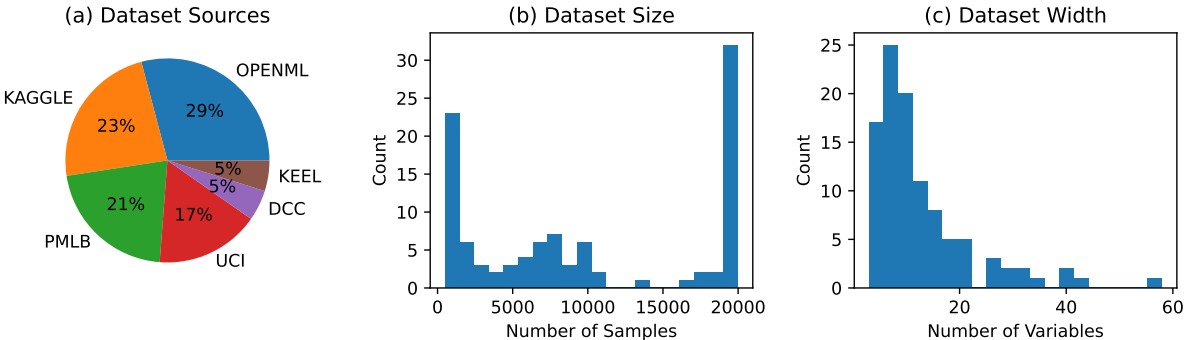

Figure 2: Statistics of the evaluation dataset: (a) distribution by source, (b) dataset sizes, and (c) feature counts.

*Retail and Sales*, *Computer Systems*, *Physics Models* and *Medicine* and exhibit different characteristics in terms of sample size and feature dimensionality. We focus specifically on the regression task in which the target variable is continuous or, if it has limited number of levels, these are ordered such as student exam scores or wine quality. The target variable in each dataset is normalized to the $[0, 10]$ range and the independent variables are used as is, raw. The target variable scaling is applied to equalize the contributions of the evaluation metrics from each dataset. Datasets have variable number of samples, the lowest being just below 1000. For very large datasets we limit the number of samples used in our benchmark to be 20,000 by subsampling uniformly at random. This allows us (i) to model imbalance, and at the same time (ii) avoid the situation in which a few large datasets could completely dominate the training and evaluation of the model. The distribution of datasets by source, number of samples and variables is shown in Figure 2. The datasets, along with their sample count, number of variables and source information are listed in Appendix C.

**Baselines:** We compare NIAQUE (total parameters ∼28M) against **Tree-based models**: XGBoost (Chen & Guestrin, 2016), LightGBM (Ke et al., 2017), and CatBoost (Prokhorenkova et al., 2019); and **Deep learning**: Transformer encoder-decoder with NIAQUE quantile decoder (total parameters ∼16M, details in Appendix H) and FT-Transformer (Gorishniy et al., 2021), using feature-wise tokenization with attention mechanism (total parameters ∼12M). The model is trained with multi-quantile loss. We evaluate three training scenarios: a) Global models (denoted by the -Global suffix): trained jointly on all datasets; b) Domain-specific models (suffix -Domain): trained on datasets from the same domain (e.g., housing, medical); c) Local models (suffix -Local): trained individually per dataset. XGBoost and CatBoost are trained using multi-quantile loss with fixed quantiles, with additional quantiles obtained through linear interpolation. LightGBM is trained with separate models per quantile. For training global tree-based models, we construct a unified table containing samples from all datasets, filling missing features with NULL values.

**Implementation Details:** NIAQUE (Total Parameters ∼ 27.6M) uses encoder and decoder containing 4 residual blocks, 2 layers each with latent dimension $E = 1024$ and input embedding size $E_{in} = 64$. Training uses Adam optimizer with initial learning rate $10^{-4}$ and batch size 512. The learning rate is reduced by $10\times$ at 500k, 600k, and 700k batches. We apply feature dropout with rate 0.2. Hyperparameters are selected using validation split and metrics are computed on the test split. Training requires approximately 24 hours for NIAQUE and 48 hours for Transformer on $4\times$V100 GPUs. In comparison, XGBoost training takes about 30 minutes on a one V100 for 3 quantiles, scaling linearly with the number of quantiles. All models are trained on the same train splits, with samples drawn uniformly at random across datasets. During training, quantile values are randomly generated for each instance in a batch.

## 5.2 Results

We evaluate NIAQUE's cross-dataset learning capabilities through following complementary analyses: (i) large-scale multi-dataset co-training, (ii) domain-specific co-training and (iii) transfer of pretrained model to unseen tasks via fine-tuning.

Table 1: Performance comparison, co-training across all 101 datasets. Lower values are better for all metrics except COVERAGE @ 95 (target: 95). For BIAS, lower absolute values are better.

| Model | SMAPE | AAD | RMSE | BIAS | COV@95 | CRPS |
|---|---|---|---|---|---|---|
| XGBoost-Global | 31.4 | 0.574 | 1.056 | -0.15 | 94.6 | 0.636 |
| XGBoost-Local | 25.6 | 0.433 | 0.883 | -0.03 | 90.8 | 0.334 |
| LightGBM-Global | 27.5 | 0.475 | 0.930 | -0.06 | 94.8 | 0.426 |
| LightGBM-Local | 25.7 | 0.427 | 0.865 | -0.03 | 91.5 | 0.327 |
| CatBoost-Global | 31.3 | 0.561 | 1.030 | -0.12 | **94.9** | 0.443 |
| CatBoost-Local | 24.3 | 0.408 | 0.840 | -0.03 | 92.7 | 0.315 |
| Transformer-Local | 26.9 | 0.462 | 0.904 | -0.05 | 93.6 | 0.329 |
| Transformer-Global | 23.1 | 0.383 | 0.806 | **-0.01** | 94.6 | 0.272 |
| FT-Transformer-Local | 25.1 | 0.420 | 0.858 | -0.04 | 94.2 | 0.298 |
| FT-Transformer-Global | 22.6 | 0.375 | 0.796 | -0.02 | 94.7 | 0.266 |
| NIAQUE-Local | 22.8 | 0.377 | 0.797 | -0.03 | **94.9** | 0.267 |
| NIAQUE-Global | **22.1** | **0.367** | **0.787** | -0.02 | 94.6 | **0.261** |

**Cross-Dataset Learning**. To evaluate NIAQUE's ability to handle large-scale multi-dataset learning, we conduct experiments across all 101 datasets simultaneously. Table 1 presents results, aggregated across datasets at sample level, comparing global and local training scenarios for various models. NIAQUE-Global achieves the best performance, outperforming both traditional tree-based methods and the Transformer baseline. Notably, while tree-based methods show better performance in local training compared to their global variants (e.g., CatBoost-Local SMAPE: 24.3 vs CatBoost-Global: 31.3), NIAQUE maintains superior performance in both scenarios, with its global model outperforming its local counterpart. Furthermore, NIAQUE maintains reliable uncertainty quantification across all scenarios, with coverage staying close to the target 95% level and consistently lower CRPS values compared to baselines. These results confirm NIAQUE's capacity to leverage cross-dataset learning effectively, maintaining or even improving performance on individual tasks through robust feature representations that generalize across diverse datasets and domains. This experiment also shows the principal inability of tree-based models to operate in cross-dataset learning scenarios, emphasizing their inability to develop joint representations across heterogeneous problems. For example, NIAQUE beats CatBoost-Global on more than 90% of the datasets, according to the detailed per-dataset performance breakdowns provided in Appendix D.1.

**Domain-Specific Results** (Tables 107 and 108 in Appendix D.2) focus on *House Price Prediction* and *Energy and Efficiency* domains, showing the ability of the model to effectively leverage additional information from domain-specific datasets to improve on target task.

**Adaptation to New Tasks on TabRegSet-101**. To evaluate NIAQUE's transfer learning capabilities on unseen tasks, we randomly split our collection of 101 datasets into 80 pretraining datasets and 21 held-out test datasets. We compare two scenarios: training from scratch (NIAQUE-Scratch) and fine-tuning a pretrained model (NIAQUE-Pretrain). The pretrained model is first trained on the 80 datasets and then fine-tuned on each held-out dataset using a 10 times smaller learning rate. To assess the impact of data scarcity, we evaluate both models by varying the fine-tuning data proportion ($p_s$) of the held-out datasets while maintaining constant test sets. Results in Table 2 demonstrate that: 1) The pretrained model consistently outperforms training from scratch across all metrics. 2) The performance gap widens as training data becomes scarcer (smaller $p_s$). 3) Both models maintain reliable uncertainty estimates, as evidenced by COVERAGE @ 95 values. These results validate that NIAQUE effectively transfers pretrained knowledge to novel regression tasks, with improvements particularly pronounced in low-data scenarios. Note that these results are not directly comparable with those in Table 1, as they are based on different dataset splits (21 vs. 101 datasets).

**Adaptation to New Tasks on Kaggle Competitions**. To validate NIAQUE's practical effectiveness in the wild, we evaluate its performance in recent Kaggle competitions: Regression with an Abalone Dataset (Read & Chow, 2024a), Regression with a Flood Prediction Dataset (Read & Chow, 2024b)

Table 2: Transfer learning results on held-out datasets. $p_s$ represents the proportion of training data used for fine-tuning, ranging from 0.05 (5%) to 1.0 (100%). Lower values are better for all metrics except COVERAGE @ 95 (target: 95). For BIAS, lower absolute values are better.

|  |  | NIAQUE | $p_s$=0.05 | 0.1 | 0.25 | 0.5 | 1.0 |
|---|---|---|---|---|---|---|---|
| SMAPE | Scratch | | 28.0 | 24.7 | 21.7 | 20.8 | 19.4 |
| | Pretrain | | 23.5 | 21.9 | 20.3 | 18.7 | 17.7 |
| AAD | Scratch | | 0.71 | 0.60 | 0.56 | 0.54 | 0.49 |
| | Pretrain | | 0.61 | 0.57 | 0.54 | 0.50 | 0.47 |
| RMSE | Scratch | | 1.23 | 1.10 | 1.06 | 1.04 | 0.96 |
| | Pretrain | | 1.11 | 1.08 | 1.04 | 0.97 | 0.94 |
| BIAS | Scratch | | -0.06 | -0.04 | -0.04 | 0.02 | -0.04 |
| | Pretrain | | -0.06 | -0.07 | -0.06 | -0.06 | -0.04 |
| CRPS | Scratch | | 0.488 | 0.423 | 0.392 | 0.383 | 0.351 |
| | Pretrain | | 0.427 | 0.404 | 0.380 | 0.354 | 0.334 |
| COV@95 | Scratch | | 93.3 | 93.0 | 94.4 | 93.1 | 94.4 |
| | Pretrain | | 95.3 | 94.4 | 94.2 | 93.9 | 94.6 |

Table 3: Performance comparison on Kaggle competition datasets: Abalone (RMSLE, lower is better), Flood Prediction (R2, higher is better).

| Model | Abalone RMLSE | Flood Prediction R2 Score |
|---|---|---|
| XGBoost (Broccoli Beef (siukeitin), 2024; Sayed, 2024) | 0.15019 | 0.842 |
| LightGBM (, dataWr3cker; Masoudi, 2024) | 0.14914 | 0.766 |
| CatBoost (Wate, 2024; Milind, 2024) | 0.14783 | 0.845 |
| TabNet (Broccoli Beef (siukeitin), 2024) | 0.15481 | 0.842 |
| TabDPT (Ma et al., 2024) | 0.15026 | 0.804 |
| TabPFN (Müller et al., 2025) | 0.15732 | 0.431 |
| NIAQUE-Scratch | 0.15047 | 0.865 |
| NIAQUE-Pretrain | **0.14808** | **0.867** |
| Winner (Heller, 2024; Aldparis, 2024) | **0.14374** | **0.869** |

(the private leaderboard competition results along with their ranks are presented in Table 3). Our approach involves two stages: pretraining and fine-tuning. First, we pretrain NIAQUE on TabRegSet-101 using our quantile loss framework. Then, we fine-tune the pretrained model on the competition's training data, optimizing for target metric. To systematically evaluate the impact of proposed pretraining strategy and architecture, we show NIAQUE-Scratch baseline (trained only on competition data, no pretraining), a number of tree-based baselines as well as TabDPT and TabPFN pretrained models. While TabDPT and TabPFN show promising results on small datasets, they face significant scalability challenges—TabPFN is limited to 10,000 samples and TabDPT requires substantial context size reduction for large datasets (details in Appendix 5.3). On the other hand, our approach shows very strong scalability and accuracy results on the competition datasets, outperforming vanilla tree-based models as well as TabDPT and TabPFN baselines. Additional results in Appendices D.3.1, D.3.2 show how our approach, without significant manual interventions, further benefits from advanced automatic feature engineering (OpenFE (Zhang et al., 2023)) and ensembling thereby rivalling results of human competitors. These results are particularly significant given that neural networks were generally considered ineffective for these competitions.

**Learned Representations** are studied qualitatively in Fig. 3 (left) showing UMAP projections (McInnes et al., 2018) of dataset row embeddings derived from NIAQUE's feature encoder. The encoder maps input

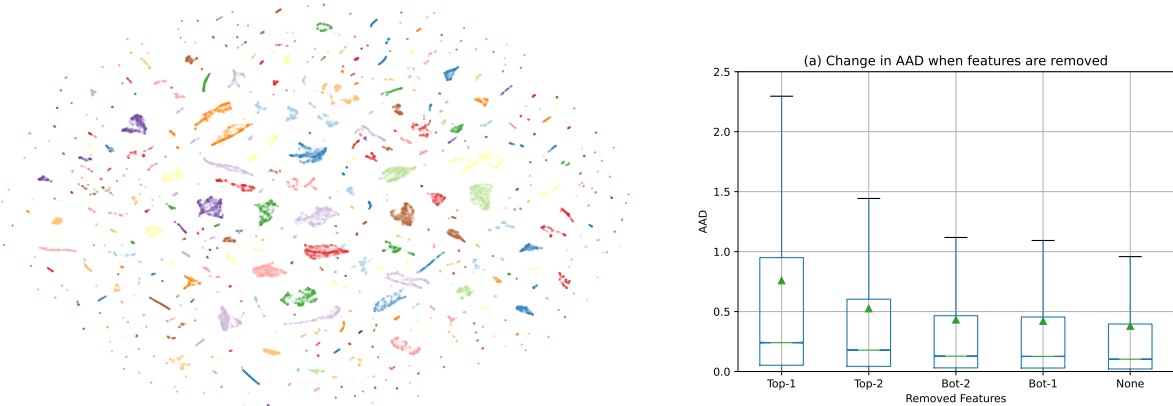

Figure 3: UMAP projections of embeddings derived from NIAQUE's feature encoder for each sample, colored by dataset (left). NIAQUE accuracy response to the removal of input features by importance (right). Top-rated features have the greatest impact on AAD degradation when removed.

features $\mathbf{x}_i$ to a fixed-dimensional latent space using a prototype-based aggregation mechanism. The resulting UMAP visualization reveals distinct dataset-specific clusters, indicating that NIAQUE learns representations that capture dataset-specific characteristics while maintaining a shared latent space that enables effective transfer learning.

**Feature Importance** is based on the inverse of the average confidence interval derived from feature's marginal distribution, as detailed in eq. (17). Qualitatively, Fig. 3 (right), indicates that features with higher weights (i.e., smaller average confidence intervals) are most critical to prediction accuracy—removing these features significantly increases the AAD metric, whereas eliminating features with lower weights has minimal impact. Quantitatively, two-sided t-test comparing the impact of removing the most vs. least important features (Top-1 vs. Bot-1 AAD values in Fig. 3) ($t = -50.24$, p-value $\approx 0$) along with the effect size (Cohen's $d = 0.22$) indicate significant and practically impactful effect across datasets. Additionally, we computed SHAP values with shap.SamplingExplainer across 101 datasets and used them as a reference ranking. For each dataset, we computed the NDCG score between the SHAP ranking and our model-native importance ranking, then averaged the results. The average NDCG was 0.899, while computation time was reduced from ~3.5h × 8 GPUs to ~20s × 1 GPU for all 101 datasets. The NDCG score of 0.9 generally demonstrates very strong ranking alignment with established attribution method at a fraction of the cost.

**Ablation Studies** (Appendices H–J) support NIAQUE's design choices. We observe: (1) our encoder outperforms attention-based alternatives; (2) log-transforming inputs (Eq. eq. (10)) enhances both stability and accuracy; (3) the model is robust to hyperparameter changes and benefits from greater depth; (4) incorporating single-feature samples enables interpretability without degrading performance. Finally, we performed the ablation of the prototype layer by disabling the prototype connection in eq. (14). We found that this drastically degrades the performance: SMAPE rises from 22.1 to 94.612, AAD from 0.367 to 2.585, and RMSE from 0.787 to 3.432, proving the effectiveness of the prototype layer.

### 5.3 Scalability Analysis of TabPFN and TabDPT

TabPFN and TabDPT mark key advances in tabular deep learning, showing strong results on small-scale datasets. However, our Kaggle experiments reveal major scalability bottlenecks, limiting their practicality in larger real-world settings. This highlights the need for scalable alternatives like NIAQUE.

**Limitations of TabPFN.** Despite TabPFN's impressive few-shot performance, its applicability is bounded by architectural and implementation constraints, as documented in the official repository (Müller et al., 2025): (i) maximum support for 10,000 training samples, (ii) limit of 500 features per instance, (iii) assumes all features are numerical, requiring preprocessing for categorical inputs. Furthermore, to adapt TabPFN

for our use cases, we employed several workarounds: (i) for large datasets (e.g., the flood prediction dataset with 1.12M samples), we applied Random Forest-based subsampling, following official guidelines (Müller et al., 2025), (ii) categorical features were ordinally encoded to conform to TabPFN's numerical input requirement. These constraints, especially the aggressive downsampling, likely contributed to TabPFN's suboptimal performance on large-scale tasks (e.g., R2 score of 0.431 on flood prediction).

**Scalability Challenges in TabDPT.** TabDPT, based on a transformer backbone, encounters scalability issues typical of attention-based models: (i) memory consumption grows quadratically with context size due to self-attention, (ii) inference time scales poorly with dataset size, (iii) performance is sensitive to reductions in context size, making trade-offs between scalability and accuracy non-trivial. In our experiments we had to reduce the context window for the flood prediction dataset to fit within memory constraints. We reduce the context size from 8,192 in powers of 2. Context size of 1024 finally works. This reduction correlated with a decline in model performance (R2 score dropped to 0.804). The compute cost of running extensive hyperparameter tuning on large datasets proved impractical. TabDPT takes 12 hours of inference time for each hyperparameter configuration for flood prediction datatset on a V100 GPU.

## 6 Discussion

We believe that our results applying NIAQUE to the 101 dataset benchmark lay out the stepping stone for the development of probabilistic meta-models eventually possessing the following key properties. **Scalability**: A unified model shares computational resources to address multiple regression tasks, optimizing resource utilization and reducing the operational costs of maintaining separate models. **Data Efficiency**: Training on diverse tasks introduces strong regularization effects, and we expect existing datasets to be repurposed to solve emerging problems, promoting data reuse and recycling. **Representation and Generalization**: A model trained across multiple datasets uncovers generalizable representations of regression tasks and ways of solving them, acquiring the ability to apply this knowledge across datasets. As an example, while TabPFN and TabDPT perform well on small datasets, they face significant scalability challenges with high-dimensional, large-scale data. NIAQUE demonstrates that scalable, accurate, and interpretable neural models are viable without relying on heavy preprocessing or memory-intensive components. These results position NIAQUE not just as a practical alternative, but also as evidence that our approach offers a fruitful research path beyond current paradigms.

**Limitations**. While we significantly expand the scope of cross-dataset probabilistic model training by applying our neural model to a 101-dataset benchmark, this remains a limited effort. It is still unclear how many datasets are required for a regression model to be considered foundational for solving, for instance, 80% of industry problems. What level of dataset diversity is necessary? Will millions or billions of unrelated datasets be required, or would 10,000 overlapping datasets suffice? Defining and evaluating global success in this context remains an open question, necessitating further research. Furthermore, our findings have implications for designing machine learning deployments based on unified models that address multiple regression tasks. We expect that this will eventually lead to improved operational efficiency and accuracy of the models. However, this could also contribute to the centralization of power among a few large entities. In this context, risk mitigation strategies include (i) improving model computational efficiency and (ii) publicly releasing data, model training code and pretrained models. Additionally, multi-task learning on multiple datasets may introduce new biases not present in locally trained models, making interpretability and fairness research critical. We explore some interpretability aspects in this paper, and further research on interpretability and fairness in large probabilistic regression models pretrained across multiple datasets seems to be an important area for future work.

## 7 Conclusion

Our study demonstrates that NIAQUE enables effective transfer learning for tabular probabilistic regression, providing additional evidence to further challenge the belief that tabular data resists neural modeling and generalization. Across 101 datasets and a Kaggle case studies, NIAQUE shows strong empirical performance, delivering scalability via a unified architecture, data efficiency through cross-dataset pretraining, and robust

generalization across tasks. Its probabilistic formulation further supports uncertainty quantification and feature importance estimation, useful for real-world deployment. While promising, open questions remain regarding the optimal scale of transfer, the design of universally effective feature preprocessing, and the theoretical principles underlying cross-domain generalization. Overall, our findings suggest a path forward for building interpretable tabular foundation models that scale well in large-scale downstream tasks.

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

## A Definitions of metrics

We employ the following metrics to evaluate the accuracy of point predictions: Symmetric Mean Absolute Percentage Error (sMAPE), Average Absolute Deviation (AAD), Root Mean Square Error (RMSE), Root Mean Square Logarithmic Error (RMSLE).

1. Symmetric Mean Absolute Percentage Error (sMAPE):

$$\text{sMAPE} = \frac{2}{S} \sum_{i=1}^{S} \frac{|y_i - \hat{y}_{i,0.5}|}{|y_i| + |\hat{y}_{i,0.5}|} \cdot 100\% \,. \tag{18}$$

2. Average Absolute Deviation (AAD):

$$\text{AAD} = \frac{1}{S} \sum_{i=1}^{S} |y_i - \hat{y}_{i,0.5}| \,. \tag{19}$$

3. Bias:

$$\text{BIAS} = \frac{1}{S} \sum_{i=1}^{S} \hat{y}_{i,0.5} - y_i \,. \tag{20}$$

4. Root Mean Square Error (RMSE):

$$\text{RMSE} = \sqrt{\frac{1}{S} \sum_{i=1}^{S} (y_i - \hat{y}_{i,0.5})^2} \,. \tag{21}$$

5. Root Mean Square Logarithmic Error (RMSLE):

$$\text{RMSLE} = \sqrt{\frac{1}{S} \sum_{i=1}^{S} (\log(y_i + 1) - \log(\hat{y}_{i,0.5} + 1))^2} \,. \tag{22}$$

# B Proof of Theorem 1

**Theorem.** *Let $F$ be a probability measure over variable $y$ such that inverse $F^{-1}$ exists and let $P_{y,\mathbf{x}}$ be the joint probability measure of variables $\mathbf{x}, y$. Then the expected loss, $\mathbb{E}\,\rho(y, F^{-1}(q))$, is minimized if and only if:*

$$F = P_{y|\mathbf{x}}. \tag{23}$$

*Additionally:*

$$\min_{F} \mathbb{E}\,\rho(y, F^{-1}(q)) = \mathbb{E}_{\mathbf{x}} \frac{1}{2} \int_{\mathbb{R}} P_{y|\mathbf{x}}(z)(1 - P_{y|\mathbf{x}}(z))\,dz. \tag{24}$$

*Proof.* First, combining (9) with the L2 representation of CRPS eq. (2) we can write:

$$\mathbb{E}\,\rho(y, F^{-1}(q)) = \mathbb{E}_{\mathbf{x},y} \frac{1}{2} \int_{\mathbb{R}} \left(F(z) - \mathbb{1}_{\{z \geq y\}}\right)^2 dz \tag{25}$$

$$= \mathbb{E}_{\mathbf{x}} \mathbb{E}_{y|\mathbf{x}} \frac{1}{2} \int_{\mathbb{R}} F^2(z) - 2F(z)\mathbb{1}_{\{z \geq y\}} + \mathbb{1}_{\{z \geq y\}} dz \tag{26}$$

$$= \mathbb{E}_{\mathbf{x}} \frac{1}{2} \int_{\mathbb{R}} F^2(z) - 2F(z)\mathbb{E}_{y|\mathbf{x}}\mathbb{1}_{\{z \geq y\}} + \mathbb{E}_{y|\mathbf{x}}\mathbb{1}_{\{z \geq y\}} dz \tag{27}$$

$$= \mathbb{E}_{\mathbf{x}} \frac{1}{2} \int_{\mathbb{R}} F^2(z) - 2F(z)P_{y|\mathbf{x}}(z) + P_{y|\mathbf{x}}(z) dz. \tag{28}$$

Here we used the law of total expectation and Fubini theorem to exchange the order of integration and then used the fact that $\mathbb{E}_{y|\mathbf{x}}\mathbb{1}_{\{z \geq y\}} = P_{y|\mathbf{x}}(z)$. Completing the square we further get:

$$\mathbb{E}\,\rho(y, F^{-1}(q)) = \mathbb{E}_{\mathbf{x}} \frac{1}{2} \int_{\mathbb{R}} F^2(z) - 2F(z)P_{y|\mathbf{x}}(z) + P_{y|\mathbf{x}}(z) + P_{y|\mathbf{x}}^2(z) - P_{y|\mathbf{x}}^2(z) dz \tag{29}$$

$$= \mathbb{E}_{\mathbf{x}} \frac{1}{2} \int_{\mathbb{R}} (F(z) - P_{y|\mathbf{x}}(z))^2 + P_{y|\mathbf{x}}(z) - P_{y|\mathbf{x}}^2(z) dz \tag{30}$$

$F = P_{y|\mathbf{x}}$ is clearly the unique minimizer of the last expression since $\int_{\mathbb{R}}(F(z) - P_{y|\mathbf{x}}(z))^2 dz > 0, \forall F \neq P_{y|\mathbf{x}}$. $\qquad \square$

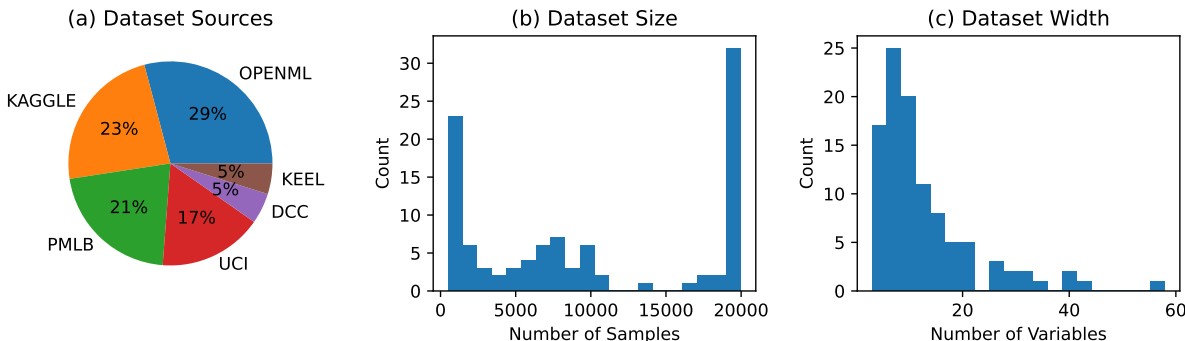

Figure 4: Statistics of the evaluation dataset: (a) distribution by source, (b) dataset sizes, and (c) feature counts.

Table 4: Performance comparison for top baselines, macro-average across all 101 datasets. Lower values are better for all metrics except COVERAGE @ 95 (target: 95). For BIAS, lower absolute values are better.

| Model | SMAPE | AAD | RMSE | BIAS | COV@95 | CRPS |
|---|---|---|---|---|---|---|
| CatBoost-Global | 42.9 | 0.982 | 1.301 | -0.34 | 92.2 | 0.709 |
| CatBoost-Local | 23.2 | 0.437 | 0.690 | -0.04 | 91.5 | 0.340 |
| FTTransformer-Local | 24.5 | 0.450 | 0.705 | -0.04 | 93.8 | 0.320 |
| FTTransformer-Global | 22.0 | 0.402 | 0.641 | **-0.02** | **94.3** | 0.285 |
| NIAQUE-Local | 22.2 | 0.405 | 0.650 | -0.03 | 94.2 | 0.287 |
| NIAQUE-Global | **21.6** | **0.394** | **0.632** | -0.03 | 93.6 | **0.283** |

## C   TabRegSet-101 Details

We introduce TabRegSet-101 (Tabular Regression Set 101), a curated collection of 101 publicly available regression datasets gathered from UCI (Kelly et al., 2017), Kaggle (Kaggle, 2024), PMLB (Romano et al., 2021; Olson et al., 2017b), OpenML (Vanschoren et al., 2013), and KEEL (Alcalá-Fdez et al., 2011). These datasets span diverse domains, including *Housing and Real Estate*, *Energy and Efficiency*, *Retail and Sales*, *Computer Systems*, *Physics Models* and *Medicine* and exhibit different characteristics in terms of sample size and feature dimensionality (Fig. 4). The datasets, along with their sample count, number of variables and source information are listed in Table 5.

We focus specifically on the regression task in which the target variable is continuous or, if it has limited number of levels, these are ordered such as student exam scores or wine quality. The target variable in each dataset is normalized to the [0, 10] range and the independent variables are used as is, raw. The target variable scaling is applied to equalize the contributions of the evaluation metrics from each dataset. Datasets have variable number of samples, the lowest being just below 1000. For very large datasets we limit the number of samples used in our benchmark to be 20,000 by subsampling uniformly at random. This allows us (i) to model imbalance, and at the same time (ii) avoid the situation in which a few large datasets could completely dominate the training and evaluation of the model. The distribution of datasets by source, number of samples and number of variables is shown in Figure 4.

Table 5: The list of datasets comprising TabRegSet-101 benchmark

| id | Name | samples | vars | source | URL |
|---|---|---|---|---|---|
| 0 | Abalone | 4177 | 7 | uci | `https://archive.ics.uci.edu/static/public/1/data.csv` |
| 1 | Student_Performance | 649 | 29 | uci | `https://archive.ics.uci.edu/static/public/320/data.csv` |
| 2 | Infrared_Thermography_Temperature | 1020 | 32 | uci | `https://archive.ics.uci.edu/static/public/925/data.csv` |
| 3 | Parkinsons_Telemonitoring | 5875 | 18 | uci | `https://archive.ics.uci.edu/static/public/189/data.csv` |
| 4 | Energy_Efficiency | 768 | 7 | uci | `https://archive.ics.uci.edu/static/public/242/data.csv` |
| 5 | 1027_ESL | 488 | 3 | pmlb | `https://github.com/EpistasisLab/penn-ml-benchmarks/raw/master/datasets/1027_ESL/1027_ESL.tsv.gz` |
| 6 | 1028_SWD | 1000 | 9 | pmlb | `https://github.com/EpistasisLab/penn-ml-benchmarks/raw/master/datasets/1028_SWD/1028_SWD.tsv.gz` |
| 7 | 1029_LEV | 1000 | 3 | pmlb | `https://github.com/EpistasisLab/penn-ml-benchmarks/raw/master/datasets/1029_LEV/1029_LEV.tsv.gz` |
| 8 | 1030_ERA | 1000 | 3 | pmlb | `https://github.com/EpistasisLab/penn-ml-benchmarks/raw/master/datasets/1030_ERA/1030_ERA.tsv.gz` |
| 9 | 1199_BNG_echoMonths | 17496 | 8 | pmlb | `https://github.com/EpistasisLab/penn-ml-benchmarks/raw/master/datasets/1199_BNG_echoMonths/1199_BNG_echoMonths.tsv.gz` |
| 10 | 197_cpu_act | 8192 | 20 | pmlb | `https://github.com/EpistasisLab/penn-ml-benchmarks/raw/master/datasets/197_cpu_act/197_cpu_act.tsv.gz` |
| 11 | 225_puma8NH | 8192 | 7 | pmlb | `https://github.com/EpistasisLab/penn-ml-benchmarks/raw/master/datasets/225_puma8NH/225_puma8NH.tsv.gz` |
| 12 | 227_cpu_small | 8192 | 11 | pmlb | `https://github.com/EpistasisLab/penn-ml-benchmarks/raw/master/datasets/227_cpu_small/227_cpu_small.tsv.gz` |
| 13 | 294_satellite_image | 6435 | 35 | pmlb | `https://github.com/EpistasisLab/penn-ml-benchmarks/raw/master/datasets/294_satellite_image/294_satellite_image.tsv.gz` |
| 14 | 344_mv | 20000 | 9 | pmlb | `https://github.com/EpistasisLab/penn-ml-benchmarks/raw/master/datasets/344_mv/344_mv.tsv.gz` |
| 15 | 503_wind | 6574 | 13 | pmlb | `https://github.com/EpistasisLab/penn-ml-benchmarks/raw/master/datasets/503_wind/503_wind.tsv.gz` |
| 16 | 529_pollen | 3848 | 3 | pmlb | `https://github.com/EpistasisLab/penn-ml-benchmarks/raw/master/datasets/529_pollen/529_pollen.tsv.gz` |
| 17 | 537_houses | 20000 | 7 | pmlb | `https://github.com/EpistasisLab/penn-ml-benchmarks/raw/master/datasets/537_houses/537_houses.tsv.gz` |

Table 5: The list of datasets comprising TabRegSet-101 benchmark

| | name | samples | vars | source | url |
|---|---|---|---|---|---|
| 18 | 547_no2 | 500 | 6 | pmlb | https://github.com/EpistasisLab/penn-ml-benchmarks/raw/master/datasets/547_no2/547_no2.tsv.gz |
| 19 | 564_fried | 20000 | 9 | pmlb | https://github.com/EpistasisLab/penn-ml-benchmarks/raw/master/datasets/564_fried/564_fried.tsv.gz |
| 20 | 595_fri_c0_1000_10 | 1000 | 9 | pmlb | https://github.com/EpistasisLab/penn-ml-benchmarks/raw/master/datasets/595_fri_c0_1000_10/595_fri_c0_1000_10.tsv.gz |
| 21 | 593_fri_c1_1000_10 | 1000 | 9 | pmlb | https://github.com/EpistasisLab/penn-ml-benchmarks/raw/master/datasets/593_fri_c1_1000_10/593_fri_c1_1000_10.tsv.gz |
| 22 | 1193_BNG_lowbwt | 20000 | 8 | pmlb | https://github.com/EpistasisLab/penn-ml-benchmarks/raw/master/datasets/1193_BNG_lowbwt/1193_BNG_lowbwt.tsv.gz |
| 23 | 1201_BNG_breastTumor | 20000 | 8 | pmlb | https://github.com/EpistasisLab/penn-ml-benchmarks/raw/master/datasets/1201_BNG_breastTumor/1201_BNG_breastTumor.tsv.gz |
| 24 | 1203_BNG_pwLinear | 20000 | 9 | pmlb | https://github.com/EpistasisLab/penn-ml-benchmarks/raw/master/datasets/1203_BNG_pwLinear/1203_BNG_pwLinear.tsv.gz |
| 25 | 215_2dplanes | 20000 | 9 | pmlb | https://github.com/EpistasisLab/penn-ml-benchmarks/raw/master/datasets/215_2dplanes/215_2dplanes.tsv.gz |
| 26 | 218_house_8L | 20000 | 7 | pmlb | https://github.com/EpistasisLab/penn-ml-benchmarks/raw/master/datasets/218_house_8L/218_house_8L.tsv.gz |
| 27 | QsarFishToxicity | 908 | 5 | uci | https://archive.ics.uci.edu/static/public/504/qsar+fish+toxicity.zip |
| 28 | concrete_compressive_strength | 1030 | 7 | uci | https://archive.ics.uci.edu/static/public/165/concrete+compressive+strength.zip |
| 29 | PRODUCTIVITY | 1197 | 12 | uci | https://archive.ics.uci.edu/static/public/597/productivity+prediction+of+garment+employees.zip |
| 30 | CCPP | 9568 | 3 | uci | https://archive.ics.uci.edu/static/public/294/combined+cycle+power+plant.zip |
| 31 | AIRFOIL | 1503 | 4 | uci | https://archive.ics.uci.edu/static/public/291/airfoil+self+noise.zip |
| 32 | TETOUAN | 20000 | 6 | uci | https://archive.ics.uci.edu/static/public/849/power+consumption+of+tetouan+city.zip |
| 33 | BIAS_CORRECTION | 7725 | 22 | uci | https://archive.ics.uci.edu/static/public/514/bias+correction+of+numerical+prediction+model+temperature+forecast.zip |

Table 5: The list of datasets comprising TabRegSet-101 benchmark

|  | name | samples | vars | source | url |
|---|---|---|---|---|---|
| 34 | APARTMENTS | 10000 | 10 | uci | `https://archive.ics.uci.edu/static/public/555/apartment+for+rent+classified.zip` |
| 35 | MedicalCost | 1338 | 5 | kaggle | kaggle datasets download -d miri-choi0218/insurance |
| 36 | Vehicle | 2059 | 18 | kaggle | kaggle datasets download -d nehalbirla/vehicle-dataset-from-cardekho |
| 37 | LifeExpectancy | 2928 | 18 | kaggle | kaggle datasets download -d kumarajarshi/life-expectancy-who |
| 38 | CalHousing | 20000 | 7 | dcc | `https://www.dcc.fc.up.pt/~ltorgo/Regression/cal_housing.tgz` |
| 39 | Ailerons | 7154 | 39 | dcc | `https://www.dcc.fc.up.pt/~ltorgo/Regression/ailerons.tgz` |
| 40 | DeltaElevators | 9517 | 5 | dcc | `https://www.dcc.fc.up.pt/~ltorgo/Regression/delta_elevators.tgz` |
| 41 | Pole | 10000 | 25 | dcc | `https://www.dcc.fc.up.pt/~ltorgo/Regression/pol.tgz` |
| 42 | Kinematics | 8192 | 7 | dcc | `https://www.dcc.fc.up.pt/~ltorgo/Regression/kinematics.tar.gz` |
| 43 | BigMartSales | 8523 | 10 | kaggle | kaggle datasets download -d brijbhushannanda1979/bigmart-sales-data |
| 44 | VideoGameSales | 16598 | 3 | kaggle | kaggle datasets download -d gregorut/videogamesales |
| 45 | NewsPopularity | 20000 | 58 | uci | `https://archive.ics.uci.edu/static/public/332/online+news+popularity.zip` |
| 46 | Wizmir | 1461 | 8 | keel | `https://sci2s.ugr.es/keel/dataset/data/regression/wizmir.zip` |
| 47 | Ele2 | 1056 | 3 | keel | `https://sci2s.ugr.es/keel/dataset/data/regression/ele-2.zip` |
| 48 | Treasury | 1049 | 14 | keel | `https://sci2s.ugr.es/keel/dataset/data/regression/treasury.zip` |
| 49 | Mortgage | 1049 | 14 | keel | `https://sci2s.ugr.es/keel/dataset/data/regression/mortgage.zip` |
| 50 | Laser | 993 | 3 | keel | `https://sci2s.ugr.es/keel/dataset/data/regression/laser.zip` |
| 51 | SpaceGa | 3107 | 5 | openml | `https://www.openml.org/data/download/52619/space_ga.arff` |
| 52 | VisualizingSoil | 8641 | 3 | openml | `https://www.openml.org/data/download/52988/visualizing_soil.arff` |
| 53 | Diamonds | 20000 | 8 | openml | `https://www.openml.org/data/download/21792853/dataset.arff` |
| 54 | TitanicFare | 1307 | 6 | openml | `https://www.openml.org/data/download/20649205/file277c5e2b70e8.arff` |
| 55 | Sulfur | 10081 | 5 | openml | `https://www.openml.org/data/download/2095629/phpBXEqg1.arff` |
| 56 | Debutanizer | 2394 | 6 | openml | `https://www.openml.org/data/download/2096280/phpWT77lf.arff` |

Table 5: The list of datasets comprising TabRegSet-101 benchmark

|  | name | samples | vars | source | url |
|---|---|---|---|---|---|
| 57 | Fardamento | 6277 | 5 | openml | https://www.openml.org/data/download/21854531/fardamento_saidas_19_20a20maio.arff |
| 58 | ProteinTertiary | 20000 | 8 | openml | https://api.openml.org/data/download/22111827/file22f167620a212.arff |
| 59 | BrazilianHouses | 10692 | 7 | openml | https://api.openml.org/data/download/22111854/file22f1627e4a960.arff |
| 60 | Cps88Wages | 20000 | 5 | openml | https://api.openml.org/data/download/22111848/file22f161d4b5556.arff |
| 61 | CPMP-2015 | 2108 | 25 | openml | https://www.openml.org/data/download/21377442/file16a868cf35f5.arff |
| 62 | NASA-PHM2008 | 20000 | 16 | openml | https://www.openml.org/data/download/22045221/dataset.arff |
| 63 | Wind | 6574 | 12 | openml | https://www.openml.org/data/download/52615/wind.arff |
| 64 | NewFuelCar | 20000 | 17 | openml | https://www.openml.org/data/download/21230500/pruebaconvonline.csv.arff |
| 65 | MiamiHousing | 13932 | 14 | openml | https://www.openml.org/data/download/22047757/miami2016.arff |
| 66 | BlackFriday | 20000 | 8 | openml | https://www.openml.org/data/download/21230845/file639340bd9ca9.arff |
| 67 | IEEE80211aaGATS | 5296 | 28 | openml | https://www.openml.org/data/download/22101884/dataset.arff |
| 68 | Yprop41 | 8885 | 41 | openml | https://api.openml.org/data/download/22111920/dataset.arff |
| 69 | Sarcos | 20000 | 20 | openml | https://api.openml.org/data/download/22111840/file22f166a1669bb.arff |
| 70 | ZurichDelays | 20000 | 16 | openml | https://www.openml.org/data/download/21854423/file86eb92864fd.arff |
| 71 | 1000-Cameras | 1015 | 13 | openml | https://www.openml.org/data/download/22102539/dataset.arff |
| 72 | GridStability | 10000 | 11 | openml | https://api.openml.org/data/download/22111837/file22f1652de1c8a.arff |
| 73 | PumaDyn32nh | 8192 | 31 | openml | https://api.openml.org/data/download/22111845/file22f161b261f3b.arff |
| 74 | Fifa | 19178 | 27 | openml | https://api.openml.org/data/download/22111894/file10aca711933d5.arff |
| 75 | WhiteWine | 4898 | 10 | openml | https://api.openml.org/data/download/22111835/file22f16150a82cd.arff |
| 76 | RedWine | 1599 | 10 | openml | https://api.openml.org/data/download/22111836/file22f162b311c38.arff |
| 77 | FpsBenchmark | 20000 | 42 | openml | https://api.openml.org/data/download/22111856/file22f1639d20997.arff |

Table 5: The list of datasets comprising TabRegSet-101 benchmark

|     | name | samples | vars | source | url |
| --- | --- | --- | --- | --- | --- |
| 78 | KingCountyHousing | 20000 | 20 | openml | `https://api.openml.org/data/download/22111853/file22f167bd414f1.arff` |
| 79 | AvocadoPrices | 18249 | 12 | kaggle | kaggle datasets download -d neuromusic/avocado-prices |
| 80 | Transcoding | 20000 | 18 | uci | `https://archive.ics.uci.edu/static/public/335/online+video+characteristics+and+transcoding+time+dataset.zip` |
| 81 | house__16H | 20000 | 15 | openml | `https://www.openml.org/data/download/52752/house_16H.arff` |
| 82 | Sales | 10738 | 13 | openml | `https://www.openml.org/data/download/21756753/dataset.arff` |
| 83 | WalmartSales | 6435 | 8 | kaggle | kaggle datasets download -d mikhail1681/walmart-sales |
| 84 | UsedCar | 6019 | 11 | kaggle | kaggle datasets download -d nitishjolly/used-car-price-prediction |
| 85 | HouseRent | 4746 | 11 | kaggle | kaggle datasets download -d iamsouravbanerjee/house-rent-prediction-dataset |
| 86 | LaptopPrice | 1273 | 15 | kaggle | kaggle datasets download -d ehtishamsadiq/uncleaned-laptop-price-dataset |
| 87 | UberFare | 20000 | 8 | kaggle | kaggle datasets download -d yasserh/uber-fares-dataset |
| 88 | Co2Emission | 7385 | 10 | kaggle | kaggle datasets download -d debajyotipodder/co2-emission-by-vehicles |
| 89 | SongPopularity | 18835 | 12 | kaggle | kaggle datasets download -d yasserh/song-popularity-dataset |
| 90 | Cars | 20000 | 8 | kaggle | kaggle datasets download -d aishwaryamuthukumar/cars-dataset-audi-bmw-ford-hyundai-skoda-vw |
| 91 | GemstonePrice | 20000 | 8 | kaggle | kaggle datasets download -d colearninglounge/gemstone-price-prediction |
| 92 | LoanAmount | 20000 | 20 | kaggle | kaggle datasets download -d phileinsophos/predict-loan-amount-data |
| 93 | SaudiArabiaCars | 5507 | 10 | kaggle | kaggle datasets download -d turkibintalib/saudi-arabia-used-cars-dataset |
| 94 | GpuKernelPerformance | 20000 | 13 | kaggle | kaggle datasets download -d rupals/gpu-runtime |
| 95 | AmericanHousePrices | 20000 | 10 | kaggle | kaggle datasets download -d jeremylarcher/american-house-prices-and-demographics-of-top-cities |
| 96 | KindleBooks | 20000 | 12 | kaggle | kaggle datasets download -d asaniczka/amazon-kindle-books-dataset-2023-130k-books |
| 97 | BookSales | 1070 | 8 | kaggle | kaggle datasets download -d thedevastator/books-sales-and-ratings |
| 98 | CapitalGain | 20000 | 12 | kaggle | kaggle datasets download -d minnieliang/adult-data |
| 99 | MarketingCampaign | 2976 | 14 | kaggle | kaggle datasets download -d ahmadazari/marketing-campaign-data |

Table 5: The list of datasets comprising TabRegSet-101 benchmark

|     | name          | samples | vars | source | url                                                                                  |
|-----|---------------|---------|------|--------|--------------------------------------------------------------------------------------|
| 100 | CampaignUplift | 2000    | 9    | kaggle | kaggle datasets download -d hwwang98/software-usage-promotion-campaign-uplift-model |

Table 6: Performance comparison on 1000-Cameras dataset. Lower values are better for all metrics except CI_Coverage (target: 95%). For BIAS, lower absolute values are better.

| Model | SMAPE | AAD | RMSE | BIAS | COV@95 | CRPS |
|---|---|---|---|---|---|---|
| CatBoost-Global | 130.371 | 1.058 | 1.250 | 0.59 | 66.7 | 0.788 |
| CatBoost-Local | 54.894 | 0.379 | 0.996 | -0.27 | 86.3 | 0.285 |
| FTTransformer-Local | 48.959 | 0.337 | 0.905 | -0.09 | 86.0 | 0.221 |
| FTTransformer-Global | 46.265 | 0.261 | 0.778 | -0.08 | 86.6 | 0.208 |
| NIAQUE-Local | 42.195 | 0.245 | 0.782 | -0.10 | 87.0 | 0.189 |
| NIAQUE-Global | 41.757 | 0.218 | 0.722 | -0.10 | 87.3 | 0.178 |

Table 7: Performance comparison on 1027_ESL dataset. Lower values are better for all metrics except CI_Coverage (target: 95%). For BIAS, lower absolute values are better.

| Model | SMAPE | AAD | RMSE | BIAS | COV@95 | CRPS |
|---|---|---|---|---|---|---|
| CatBoost-Global | 57.151 | 2.638 | 3.034 | -2.58 | 95.9 | 1.585 |
| CatBoost-Local | 10.353 | 0.531 | 0.696 | -0.05 | 93.9 | 0.394 |
| FTTransformer-Local | 12.201 | 0.561 | 0.713 | -0.05 | 86.6 | 0.406 |
| FTTransformer-Global | 10.431 | 0.520 | 0.661 | -0.04 | 90.5 | 0.374 |
| NIAQUE-Local | 10.634 | 0.539 | 0.721 | -0.05 | 87.5 | 0.404 |
| NIAQUE-Global | 10.293 | 0.537 | 0.704 | -0.05 | 85.7 | 0.394 |

# D    Supplementary Results

## D.1    Per-Dataset Performance Results

This sub-section provides detailed performance metrics for each of the 101 datasets in our benchmark. Tables are organized alphabetically by dataset name. Each table shows the performance of top model variants (CatBoost-Global, CatBoost-Local, FTTransformer-Local, FTTransformer-Global, NIAQUE-Local, and NIAQUE-Global) across all evaluation metrics.

Table 8: Performance comparison on 1028_SWD dataset. Lower values are better for all metrics except CI_Coverage (target: 95%). For BIAS, lower absolute values are better.

| Model | SMAPE | AAD | RMSE | BIAS | COV@95 | CRPS |
|---|---|---|---|---|---|---|
| CatBoost-Global | 46.203 | 2.557 | 3.526 | -2.43 | 80.0 | 1.866 |
| CatBoost-Local | 28.035 | 1.534 | 2.120 | 0.15 | 88.0 | 1.189 |
| FTTransformer-Local | 30.224 | 1.489 | 2.182 | -0.09 | 95.7 | 1.141 |
| FTTransformer-Global | 26.172 | 1.468 | 2.158 | -0.07 | 93.4 | 1.013 |
| NIAQUE-Local | 27.206 | 1.407 | 2.162 | -0.10 | 98.3 | 1.034 |
| NIAQUE-Global | 26.771 | 1.413 | 2.145 | -0.10 | 96.0 | 1.024 |

Table 9: Performance comparison on 1029_LEV dataset. Lower values are better for all metrics except CI_Coverage (target: 95%). For BIAS, lower absolute values are better.

| Model | SMAPE | AAD | RMSE | BIAS | COV@95 | CRPS |
|---|---|---|---|---|---|---|
| CatBoost-Global | 63.046 | 2.509 | 2.978 | -1.97 | 90.0 | 1.569 |
| CatBoost-Local | 36.867 | 1.362 | 1.846 | -0.14 | 93.0 | 0.991 |
| FTTransformer-Local | 35.764 | 1.301 | 1.828 | -0.27 | 90.8 | 0.919 |
| FTTransformer-Global | 34.588 | 1.230 | 1.787 | -0.27 | 92.0 | 0.878 |
| NIAQUE-Local | 33.430 | 1.154 | 1.708 | -0.34 | 92.6 | 0.888 |
| NIAQUE-Global | 32.545 | 1.130 | 1.734 | -0.30 | 88.0 | 0.889 |

Table 10: Performance comparison on 1030_ERA dataset. Lower values are better for all metrics except CI_Coverage (target: 95%). For BIAS, lower absolute values are better.

| Model | SMAPE | AAD | RMSE | BIAS | COV@95 | CRPS |
|---|---|---|---|---|---|---|
| CatBoost-Global | 61.934 | 2.098 | 2.617 | -0.92 | 91.0 | 1.390 |
| CatBoost-Local | 48.810 | 1.452 | 1.802 | -0.10 | 91.0 | 1.043 |
| FTTransformer-Local | 54.164 | 1.511 | 1.884 | -0.17 | 95.1 | 1.051 |
| FTTransformer-Global | 50.872 | 1.337 | 1.797 | -0.15 | 92.9 | 0.987 |
| NIAQUE-Local | 50.580 | 1.435 | 1.870 | -0.18 | 96.6 | 1.006 |
| NIAQUE-Global | 51.829 | 1.457 | 1.848 | -0.15 | 97.0 | 1.019 |

Table 11: Performance comparison on 1193_BNG_lowbwt dataset. Lower values are better for all metrics except CI_Coverage (target: 95%). For BIAS, lower absolute values are better.

| Model | SMAPE | AAD | RMSE | BIAS | COV@95 | CRPS |
|---|---|---|---|---|---|---|
| CatBoost-Global | 17.093 | 0.889 | 1.141 | -0.18 | 95.9 | 0.658 |
| CatBoost-Local | 16.586 | 0.867 | 1.111 | 0.00 | 93.5 | 0.619 |
| FTTransformer-Local | 18.315 | 0.852 | 1.082 | 0.00 | 95.7 | 0.622 |
| FTTransformer-Global | 16.620 | 0.844 | 1.063 | 0.03 | 95.1 | 0.594 |
| NIAQUE-Local | 17.399 | 0.895 | 1.138 | 0.02 | 99.0 | 0.593 |
| NIAQUE-Global | 16.278 | 0.850 | 1.114 | 0.02 | 94.3 | 0.588 |

Table 12: Performance comparison on 1199_BNG_echoMonths dataset. Lower values are better for all metrics except CI_Coverage (target: 95%). For BIAS, lower absolute values are better.

| Model | SMAPE | AAD | RMSE | BIAS | COV@95 | CRPS |
|---|---|---|---|---|---|---|
| CatBoost-Global | 46.037 | 1.261 | 1.585 | -0.27 | 98.3 | 0.912 |
| CatBoost-Local | 36.889 | 1.109 | 1.513 | -0.11 | 93.4 | 0.810 |
| FTTransformer-Local | 39.263 | 1.122 | 1.500 | -0.10 | 96.3 | 0.798 |
| FTTransformer-Global | 34.530 | 1.017 | 1.542 | -0.09 | 94.7 | 0.734 |
| NIAQUE-Local | 35.032 | 1.078 | 1.576 | -0.09 | 97.0 | 0.787 |
| NIAQUE-Global | 35.024 | 1.089 | 1.536 | -0.10 | 93.3 | 0.760 |

Table 13: Performance comparison on 1201_BNG_breastTumor dataset. Lower values are better for all metrics except CI_Coverage (target: 95%). For BIAS, lower absolute values are better.

| Model | SMAPE | AAD | RMSE | BIAS | COV@95 | CRPS |
|---|---|---|---|---|---|---|
| CatBoost-Global | 24.650 | 1.101 | 1.406 | -0.11 | 95.9 | 0.822 |
| CatBoost-Local | 26.422 | 1.191 | 1.557 | 0.04 | 94.2 | 0.883 |
| FTTransformer-Local | 30.854 | 1.162 | 1.645 | 0.02 | 90.5 | 0.894 |
| FTTransformer-Global | 26.612 | 1.176 | 1.533 | 0.03 | 99.0 | 0.789 |
| NIAQUE-Local | 27.202 | 1.223 | 1.588 | 0.03 | 95.2 | 0.803 |
| NIAQUE-Global | 26.360 | 1.184 | 1.583 | 0.03 | 94.0 | 0.836 |

Table 14: Performance comparison on 1203_BNG_pwLinear dataset. Lower values are better for all metrics except CI_Coverage (target: 95%). For BIAS, lower absolute values are better.

| Model | SMAPE | AAD | RMSE | BIAS | COV@95 | CRPS |
|---|---|---|---|---|---|---|
| CatBoost-Global | 17.269 | 0.743 | 1.012 | -0.07 | 95.7 | 0.586 |
| CatBoost-Local | 19.800 | 0.719 | 0.999 | -0.02 | 95.1 | 0.562 |
| FTTransformer-Local | 21.944 | 0.735 | 1.084 | -0.03 | 93.4 | 0.539 |
| FTTransformer-Global | 19.310 | 0.751 | 0.992 | -0.01 | 98.4 | 0.522 |
| NIAQUE-Local | 20.350 | 0.741 | 1.040 | -0.02 | 94.8 | 0.534 |
| NIAQUE-Global | 19.698 | 0.710 | 1.007 | -0.02 | 94.7 | 0.509 |

Table 15: Performance comparison on 197_cpu_act dataset. Lower values are better for all metrics except CI_Coverage (target: 95%). For BIAS, lower absolute values are better.

| Model | SMAPE | AAD | RMSE | BIAS | COV@95 | CRPS |
|---|---|---|---|---|---|---|
| CatBoost-Global | 12.227 | 0.502 | 0.692 | -0.03 | 87.7 | 0.398 |
| CatBoost-Local | 9.134 | 0.190 | 0.282 | 0.01 | 91.3 | 0.174 |
| FTTransformer-Local | 11.919 | 0.218 | 0.321 | 0.00 | 90.4 | 0.152 |
| FTTransformer-Global | 9.138 | 0.179 | 0.259 | 0.02 | 96.1 | 0.126 |
| NIAQUE-Local | 9.722 | 0.186 | 0.273 | 0.01 | 95.8 | 0.133 |
| NIAQUE-Global | 9.395 | 0.178 | 0.262 | 0.01 | 93.5 | 0.130 |

Table 16: Performance comparison on 215_2dplanes dataset. Lower values are better for all metrics except CI_Coverage (target: 95%). For BIAS, lower absolute values are better.

| Model | SMAPE | AAD | RMSE | BIAS | COV@95 | CRPS |
|---|---|---|---|---|---|---|
| CatBoost-Global | 12.621 | 0.547 | 0.714 | -0.02 | 98.4 | 0.442 |
| CatBoost-Local | 7.430 | 0.334 | 0.422 | 0.00 | 93.0 | 0.242 |
| FTTransformer-Local | 9.219 | 0.351 | 0.472 | -0.00 | 97.4 | 0.250 |
| FTTransformer-Global | 7.024 | 0.336 | 0.409 | 0.01 | 92.0 | 0.219 |
| NIAQUE-Local | 7.817 | 0.340 | 0.441 | 0.01 | 92.2 | 0.237 |
| NIAQUE-Global | 7.459 | 0.334 | 0.423 | 0.01 | 93.2 | 0.236 |

Table 17: Performance comparison on 218_house_8L dataset. Lower values are better for all metrics except CI_Coverage (target: 95%). For BIAS, lower absolute values are better.

| Model | SMAPE | AAD | RMSE | BIAS | COV@95 | CRPS |
|---|---|---|---|---|---|---|
| CatBoost-Global | 31.438 | 0.367 | 0.810 | -0.17 | 86.1 | 0.275 |
| CatBoost-Local | 26.765 | 0.299 | 0.623 | -0.10 | 93.4 | 0.229 |
| FTTransformer-Local | 28.623 | 0.342 | 0.659 | -0.07 | 96.5 | 0.245 |
| FTTransformer-Global | 25.992 | 0.272 | 0.588 | -0.07 | 94.2 | 0.195 |
| NIAQUE-Local | 26.473 | 0.298 | 0.621 | -0.06 | 95.8 | 0.212 |
| NIAQUE-Global | 25.339 | 0.286 | 0.602 | -0.07 | 94.4 | 0.205 |

Table 18: Performance comparison on 225_puma8NH dataset. Lower values are better for all metrics except CI_Coverage (target: 95%). For BIAS, lower absolute values are better.

| Model | SMAPE | AAD | RMSE | BIAS | COV@95 | CRPS |
|---|---|---|---|---|---|---|
| CatBoost-Global | 32.042 | 1.669 | 1.965 | -0.72 | 91.7 | 1.173 |
| CatBoost-Local | 21.536 | 1.059 | 1.389 | -0.01 | 92.6 | 0.776 |
| FTTransformer-Local | 24.253 | 1.156 | 1.591 | -0.02 | 89.2 | 0.804 |
| FTTransformer-Global | 24.234 | 1.136 | 1.436 | -0.00 | 93.5 | 0.787 |
| NIAQUE-Local | 23.464 | 1.130 | 1.513 | -0.01 | 91.0 | 0.749 |
| NIAQUE-Global | 23.232 | 1.114 | 1.474 | -0.02 | 89.8 | 0.782 |

Table 19: Performance comparison on 227_cpu_small dataset. Lower values are better for all metrics except CI_Coverage (target: 95%). For BIAS, lower absolute values are better.

| Model | SMAPE | AAD | RMSE | BIAS | COV@95 | CRPS |
|---|---|---|---|---|---|---|
| CatBoost-Global | 12.680 | 0.510 | 0.644 | -0.07 | 85.5 | 0.405 |
| CatBoost-Local | 9.630 | 0.221 | 0.325 | 0.02 | 91.3 | 0.197 |
| FTTransformer-Local | 12.332 | 0.253 | 0.361 | 0.00 | 92.5 | 0.183 |
| FTTransformer-Global | 9.393 | 0.215 | 0.313 | 0.02 | 93.4 | 0.154 |
| NIAQUE-Local | 10.314 | 0.219 | 0.322 | 0.01 | 99.0 | 0.153 |
| NIAQUE-Global | 9.818 | 0.212 | 0.307 | 0.01 | 94.9 | 0.153 |

Table 20: Performance comparison on 294_satellite_image dataset. Lower values are better for all metrics except CI_Coverage (target: 95%). For BIAS, lower absolute values are better.

| Model | SMAPE | AAD | RMSE | BIAS | COV@95 | CRPS |
|---|---|---|---|---|---|---|
| CatBoost-Global | 66.647 | 1.312 | 1.917 | -0.30 | 80.0 | 1.046 |
| CatBoost-Local | 58.329 | 0.902 | 1.592 | -0.20 | 93.5 | 0.743 |
| FTTransformer-Local | 57.452 | 0.688 | 1.418 | -0.04 | 97.9 | 0.502 |
| FTTransformer-Global | 55.783 | 0.553 | 1.408 | -0.02 | 99.0 | 0.434 |
| NIAQUE-Local | 53.936 | 0.492 | 1.311 | -0.02 | 96.0 | 0.353 |
| NIAQUE-Global | 52.053 | 0.434 | 1.290 | -0.03 | 97.8 | 0.316 |

Table 21: Performance comparison on 344_mv dataset. Lower values are better for all metrics except CI_Coverage (target: 95%). For BIAS, lower absolute values are better.

| Model | SMAPE | AAD | RMSE | BIAS | COV@95 | CRPS |
|---|---|---|---|---|---|---|
| CatBoost-Global | 5.877 | 0.292 | 0.548 | -0.02 | 96.9 | 0.283 |
| CatBoost-Local | 1.268 | 0.051 | 0.097 | -0.00 | 94.8 | 0.111 |
| FTTransformer-Local | 3.056 | 0.069 | 0.101 | -0.01 | 98.0 | 0.072 |
| FTTransformer-Global | 0.428 | 0.020 | 0.031 | 0.01 | 97.2 | 0.028 |
| NIAQUE-Local | 0.737 | 0.024 | 0.037 | -0.00 | 99.0 | 0.022 |
| NIAQUE-Global | 0.219 | 0.013 | 0.017 | -0.00 | 100.0 | 0.012 |

Table 22: Performance comparison on 503_wind dataset. Lower values are better for all metrics except CI_Coverage (target: 95%). For BIAS, lower absolute values are better.

| Model | SMAPE | AAD | RMSE | BIAS | COV@95 | CRPS |
|-------|-------|-----|------|------|--------|------|
| CatBoost-Global | 29.173 | 0.956 | 1.248 | -0.11 | 98.6 | 0.795 |
| CatBoost-Local | 17.484 | 0.544 | 0.702 | 0.02 | 93.2 | 0.394 |
| FTTransformer-Local | 21.583 | 0.584 | 0.765 | 0.02 | 94.0 | 0.413 |
| FTTransformer-Global | 18.283 | 0.521 | 0.722 | 0.04 | 95.5 | 0.370 |
| NIAQUE-Local | 17.955 | 0.545 | 0.710 | 0.03 | 96.7 | 0.379 |
| NIAQUE-Global | 17.398 | 0.544 | 0.712 | 0.03 | 93.8 | 0.384 |

Table 23: Performance comparison on 529_pollen dataset. Lower values are better for all metrics except CI_Coverage (target: 95%). For BIAS, lower absolute values are better.

| Model | SMAPE | AAD | RMSE | BIAS | COV@95 | CRPS |
|-------|-------|-----|------|------|--------|------|
| CatBoost-Global | 21.431 | 1.082 | 1.372 | -0.66 | 98.4 | 0.777 |
| CatBoost-Local | 11.251 | 0.548 | 0.698 | -0.02 | 90.4 | 0.396 |
| FTTransformer-Local | 13.877 | 0.577 | 0.698 | -0.03 | 90.3 | 0.396 |
| FTTransformer-Global | 10.601 | 0.523 | 0.677 | -0.01 | 91.2 | 0.361 |
| NIAQUE-Local | 10.993 | 0.526 | 0.665 | -0.01 | 87.3 | 0.390 |
| NIAQUE-Global | 10.937 | 0.525 | 0.679 | -0.02 | 90.4 | 0.377 |

Table 24: Performance comparison on 537_houses dataset. Lower values are better for all metrics except CI_Coverage (target: 95%). For BIAS, lower absolute values are better.

| Model | SMAPE | AAD | RMSE | BIAS | COV@95 | CRPS |
|-------|-------|-----|------|------|--------|------|
| CatBoost-Global | 35.823 | 1.278 | 1.809 | -0.30 | 96.0 | 0.977 |
| CatBoost-Local | 19.621 | 0.756 | 1.188 | -0.19 | 94.2 | 0.583 |
| FTTransformer-Local | 23.501 | 0.771 | 1.285 | -0.16 | 91.3 | 0.582 |
| FTTransformer-Global | 21.238 | 0.800 | 1.219 | -0.12 | 94.1 | 0.585 |
| NIAQUE-Local | 22.344 | 0.823 | 1.231 | -0.13 | 94.9 | 0.600 |
| NIAQUE-Global | 22.198 | 0.819 | 1.263 | -0.13 | 93.2 | 0.586 |

Table 25: Performance comparison on 547_no2 dataset. Lower values are better for all metrics except CI_Coverage (target: 95%). For BIAS, lower absolute values are better.

| Model | SMAPE | AAD | RMSE | BIAS | COV@95 | CRPS |
|-------|-------|-----|------|------|--------|------|
| CatBoost-Global | 59.636 | 2.671 | 2.992 | -2.51 | 100.0 | 1.573 |
| CatBoost-Local | 18.142 | 0.851 | 1.144 | 0.12 | 86.0 | 0.634 |
| FTTransformer-Local | 20.602 | 0.931 | 1.282 | -0.01 | 83.8 | 0.731 |
| FTTransformer-Global | 17.903 | 0.873 | 1.162 | 0.00 | 81.1 | 0.656 |
| NIAQUE-Local | 21.115 | 0.924 | 1.213 | -0.00 | 85.2 | 0.665 |
| NIAQUE-Global | 20.253 | 0.925 | 1.189 | -0.01 | 80.0 | 0.687 |

Table 26: Performance comparison on 564_fried dataset. Lower values are better for all metrics except CI_Coverage (target: 95%). For BIAS, lower absolute values are better.

| Model | SMAPE | AAD | RMSE | BIAS | COV@95 | CRPS |
|---|---|---|---|---|---|---|
| CatBoost-Global | 11.919 | 0.522 | 0.677 | 0.01 | 97.9 | 0.421 |
| CatBoost-Local | 6.382 | 0.282 | 0.354 | -0.01 | 93.5 | 0.204 |
| FTTransformer-Local | 8.052 | 0.281 | 0.380 | -0.01 | 96.5 | 0.199 |
| FTTransformer-Global | 5.930 | 0.261 | 0.324 | 0.01 | 95.2 | 0.170 |
| NIAQUE-Local | 6.489 | 0.271 | 0.343 | -0.00 | 93.6 | 0.192 |
| NIAQUE-Global | 5.978 | 0.262 | 0.330 | -0.00 | 94.2 | 0.184 |

Table 27: Performance comparison on 593_fri_c1_1000_10 dataset. Lower values are better for all metrics except CI_Coverage (target: 95%). For BIAS, lower absolute values are better.

| Model | SMAPE | AAD | RMSE | BIAS | COV@95 | CRPS |
|---|---|---|---|---|---|---|
| CatBoost-Global | 61.047 | 2.633 | 3.026 | -2.20 | 95.0 | 1.589 |
| CatBoost-Local | 8.834 | 0.343 | 0.443 | 0.08 | 84.0 | 0.282 |
| FTTransformer-Local | 10.481 | 0.352 | 0.464 | 0.05 | 91.9 | 0.245 |
| FTTransformer-Global | 8.592 | 0.301 | 0.398 | 0.06 | 84.0 | 0.230 |
| NIAQUE-Local | 9.019 | 0.322 | 0.421 | 0.07 | 84.6 | 0.225 |
| NIAQUE-Global | 8.447 | 0.297 | 0.398 | 0.05 | 89.0 | 0.215 |

Table 28: Performance comparison on 595_fri_c0_1000_10 dataset. Lower values are better for all metrics except CI_Coverage (target: 95%). For BIAS, lower absolute values are better.

| Model | SMAPE | AAD | RMSE | BIAS | COV@95 | CRPS |
|---|---|---|---|---|---|---|
| CatBoost-Global | 56.409 | 2.543 | 3.044 | -2.26 | 94.0 | 1.579 |
| CatBoost-Local | 10.569 | 0.456 | 0.569 | -0.08 | 81.0 | 0.338 |
| FTTransformer-Local | 12.767 | 0.458 | 0.577 | 0.01 | 81.3 | 0.332 |
| FTTransformer-Global | 9.962 | 0.413 | 0.489 | 0.03 | 81.1 | 0.282 |
| NIAQUE-Local | 10.247 | 0.433 | 0.531 | 0.02 | 81.1 | 0.309 |
| NIAQUE-Global | 9.801 | 0.412 | 0.510 | 0.02 | 81.0 | 0.297 |

Table 29: Performance comparison on AIRFOIL dataset. Lower values are better for all metrics except CI_Coverage (target: 95%). For BIAS, lower absolute values are better.

| Model | SMAPE | AAD | RMSE | BIAS | COV@95 | CRPS |
|---|---|---|---|---|---|---|
| CatBoost-Global | 49.769 | 2.340 | 2.733 | -1.86 | 94.0 | 1.485 |
| CatBoost-Local | 13.736 | 0.583 | 0.786 | 0.04 | 90.7 | 0.439 |
| FTTransformer-Local | 12.358 | 0.500 | 0.728 | 0.07 | 97.6 | 0.324 |
| FTTransformer-Global | 10.709 | 0.453 | 0.603 | 0.10 | 96.6 | 0.296 |
| NIAQUE-Local | 9.590 | 0.389 | 0.592 | 0.09 | 95.6 | 0.282 |
| NIAQUE-Global | 8.722 | 0.378 | 0.542 | 0.09 | 95.4 | 0.265 |

Table 30: Performance comparison on APARTMENTS dataset. Lower values are better for all metrics except CI_Coverage (target: 95%). For BIAS, lower absolute values are better.

| Model | SMAPE | AAD | RMSE | BIAS | COV@95 | CRPS |
|---|---|---|---|---|---|---|
| CatBoost-Global | 37.753 | 0.099 | 0.177 | 0.01 | 93.9 | 0.076 |
| CatBoost-Local | 22.317 | 0.056 | 0.111 | -0.02 | 93.7 | 0.042 |
| FTTransformer-Local | 24.986 | 0.096 | 0.167 | -0.02 | 94.3 | 0.053 |
| FTTransformer-Global | 22.516 | 0.055 | 0.118 | 0.00 | 98.4 | 0.028 |
| NIAQUE-Local | 25.530 | 0.070 | 0.149 | -0.01 | 96.7 | 0.044 |
| NIAQUE-Global | 23.637 | 0.060 | 0.131 | -0.01 | 94.7 | 0.042 |

Table 31: Performance comparison on Abalone dataset. Lower values are better for all metrics except CI_Coverage (target: 95%). For BIAS, lower absolute values are better.

| Model | SMAPE | AAD | RMSE | BIAS | COV@95 | CRPS |
|---|---|---|---|---|---|---|
| CatBoost-Global | 2.569 | 0.057 | 0.099 | -0.00 | 94.9 | 0.054 |
| CatBoost-Local | 16.587 | 0.559 | 0.848 | -0.15 | 90.2 | 0.416 |
| FTTransformer-Local | 17.623 | 0.531 | 0.845 | -0.12 | 93.1 | 0.391 |
| FTTransformer-Global | 15.269 | 0.546 | 0.789 | -0.14 | 93.3 | 0.399 |
| NIAQUE-Local | 17.429 | 0.554 | 0.835 | -0.12 | 95.6 | 0.386 |
| NIAQUE-Global | 15.842 | 0.536 | 0.817 | -0.14 | 93.1 | 0.383 |

Table 32: Performance comparison on Ailerons dataset. Lower values are better for all metrics except CI_Coverage (target: 95%). For BIAS, lower absolute values are better.

| Model | SMAPE | AAD | RMSE | BIAS | COV@95 | CRPS |
|---|---|---|---|---|---|---|
| CatBoost-Global | 9.310 | 0.678 | 0.931 | -0.15 | 95.1 | 0.524 |
| CatBoost-Local | 5.222 | 0.367 | 0.519 | 0.02 | 94.3 | 0.274 |
| FTTransformer-Local | 7.749 | 0.412 | 0.586 | 0.03 | 95.0 | 0.282 |
| FTTransformer-Global | 4.741 | 0.365 | 0.504 | 0.05 | 92.1 | 0.256 |
| NIAQUE-Local | 5.845 | 0.384 | 0.535 | 0.05 | 93.7 | 0.268 |
| NIAQUE-Global | 5.315 | 0.377 | 0.519 | 0.05 | 93.3 | 0.266 |

Table 33: Performance comparison on AmericanHousePrices dataset. Lower values are better for all metrics except CI_Coverage (target: 95%). For BIAS, lower absolute values are better.

| Model | SMAPE | AAD | RMSE | BIAS | COV@95 | CRPS |
|---|---|---|---|---|---|---|
| CatBoost-Global | 34.982 | 0.070 | 0.274 | -0.03 | 92.0 | 0.053 |
| CatBoost-Local | 22.551 | 0.046 | 0.134 | -0.01 | 94.2 | 0.039 |
| FTTransformer-Local | 24.524 | 0.085 | 0.182 | -0.02 | 95.7 | 0.047 |
| FTTransformer-Global | 22.340 | 0.041 | 0.128 | 0.00 | 96.9 | 0.019 |
| NIAQUE-Local | 22.585 | 0.051 | 0.154 | -0.00 | 91.6 | 0.033 |
| NIAQUE-Global | 21.578 | 0.044 | 0.140 | -0.01 | 93.3 | 0.032 |

Table 34: Performance comparison on AvocadoPrices dataset. Lower values are better for all metrics except CI_Coverage (target: 95%). For BIAS, lower absolute values are better.

| Model | SMAPE | AAD | RMSE | BIAS | COV@95 | CRPS |
|---|---|---|---|---|---|---|
| CatBoost-Global | 25.051 | 0.829 | 1.081 | -0.16 | 99.1 | 0.689 |
| CatBoost-Local | 16.905 | 0.559 | 0.771 | -0.05 | 94.5 | 0.416 |
| FTTransformer-Local | 14.009 | 0.494 | 0.657 | 0.03 | 90.2 | 0.332 |
| FTTransformer-Global | 12.382 | 0.418 | 0.555 | 0.05 | 88.7 | 0.267 |
| NIAQUE-Local | 10.722 | 0.344 | 0.538 | 0.03 | 89.6 | 0.247 |
| NIAQUE-Global | 10.054 | 0.331 | 0.492 | 0.03 | 88.1 | 0.241 |

Table 35: Performance comparison on BIAS_CORRECTION dataset. Lower values are better for all metrics except CI_Coverage (target: 95%). For BIAS, lower absolute values are better.

| Model | SMAPE | AAD | RMSE | BIAS | COV@95 | CRPS |
|---|---|---|---|---|---|---|
| CatBoost-Global | 12.196 | 0.686 | 0.942 | -0.28 | 98.1 | 0.590 |
| CatBoost-Local | 6.949 | 0.379 | 0.513 | 0.01 | 93.8 | 0.284 |
| FTTransformer-Local | 8.451 | 0.380 | 0.518 | -0.00 | 93.2 | 0.269 |
| FTTransformer-Global | 6.301 | 0.346 | 0.439 | 0.02 | 95.6 | 0.238 |
| NIAQUE-Local | 6.460 | 0.341 | 0.463 | 0.01 | 94.3 | 0.242 |
| NIAQUE-Global | 5.814 | 0.328 | 0.436 | 0.01 | 95.3 | 0.232 |

Table 36: Performance comparison on BigMartSales dataset. Lower values are better for all metrics except CI_Coverage (target: 95%). For BIAS, lower absolute values are better.

| Model | SMAPE | AAD | RMSE | BIAS | COV@95 | CRPS |
|---|---|---|---|---|---|---|
| CatBoost-Global | 59.011 | 0.741 | 0.925 | 0.16 | 93.9 | 0.554 |
| CatBoost-Local | 41.202 | 0.555 | 0.794 | -0.03 | 94.4 | 0.394 |
| FTTransformer-Local | 44.637 | 0.602 | 0.901 | -0.04 | 95.4 | 0.393 |
| FTTransformer-Global | 40.186 | 0.568 | 0.876 | -0.03 | 99.0 | 0.387 |
| NIAQUE-Local | 39.697 | 0.569 | 0.834 | -0.03 | 97.3 | 0.397 |
| NIAQUE-Global | 40.960 | 0.562 | 0.823 | -0.03 | 95.3 | 0.391 |

Table 37: Performance comparison on BlackFriday dataset. Lower values are better for all metrics except CI_Coverage (target: 95%). For BIAS, lower absolute values are better.

| Model | SMAPE | AAD | RMSE | BIAS | COV@95 | CRPS |
|---|---|---|---|---|---|---|
| CatBoost-Global | 27.779 | 1.217 | 1.578 | 0.23 | 92.4 | 0.876 |
| CatBoost-Local | 27.003 | 1.187 | 1.567 | 0.26 | 94.0 | 0.858 |
| FTTransformer-Local | 27.843 | 1.243 | 1.553 | 0.18 | 95.9 | 0.814 |
| FTTransformer-Global | 27.469 | 1.179 | 1.661 | 0.22 | 90.6 | 0.834 |
| NIAQUE-Local | 27.271 | 1.213 | 1.562 | 0.26 | 96.7 | 0.787 |
| NIAQUE-Global | 27.051 | 1.182 | 1.574 | 0.23 | 95.7 | 0.823 |

Table 38: Performance comparison on BookSales dataset. Lower values are better for all metrics except CI_Coverage (target: 95%). For BIAS, lower absolute values are better.

| Model | SMAPE | AAD | RMSE | BIAS | COV@95 | CRPS |
|---|---|---|---|---|---|---|
| CatBoost-Global | 131.764 | 0.928 | 1.367 | 0.45 | 74.8 | 0.752 |
| CatBoost-Local | 41.686 | 0.239 | 0.938 | -0.14 | 84.1 | 0.213 |
| FTTransformer-Local | 38.638 | 0.282 | 1.006 | -0.12 | 94.6 | 0.201 |
| FTTransformer-Global | 38.462 | 0.241 | 0.869 | -0.08 | 92.4 | 0.179 |
| NIAQUE-Local | 38.616 | 0.247 | 0.894 | -0.12 | 94.9 | 0.174 |
| NIAQUE-Global | 37.963 | 0.245 | 0.846 | -0.11 | 95.3 | 0.173 |

Table 39: Performance comparison on BrazilianHouses dataset. Lower values are better for all metrics except CI_Coverage (target: 95%). For BIAS, lower absolute values are better.

| Model | SMAPE | AAD | RMSE | BIAS | COV@95 | CRPS |
|---|---|---|---|---|---|---|
| CatBoost-Global | 47.270 | 0.019 | 0.031 | -0.00 | 87.6 | 0.018 |
| CatBoost-Local | 32.904 | 0.014 | 0.024 | -0.00 | 93.9 | 0.010 |
| FTTransformer-Local | 33.447 | 0.053 | 0.069 | -0.01 | 93.9 | 0.021 |
| FTTransformer-Global | 33.888 | 0.011 | 0.014 | 0.00 | 93.8 | 0.000 |
| NIAQUE-Local | 33.455 | 0.022 | 0.038 | -0.00 | 95.7 | 0.010 |
| NIAQUE-Global | 33.171 | 0.014 | 0.024 | -0.01 | 94.7 | 0.010 |

Table 40: Performance comparison on CCPP dataset. Lower values are better for all metrics except CI_Coverage (target: 95%). For BIAS, lower absolute values are better.

| Model | SMAPE | AAD | RMSE | BIAS | COV@95 | CRPS |
|---|---|---|---|---|---|---|
| CatBoost-Global | 20.671 | 0.876 | 1.252 | -0.43 | 97.7 | 0.672 |
| CatBoost-Local | 10.508 | 0.364 | 0.497 | 0.03 | 95.8 | 0.271 |
| FTTransformer-Local | 13.392 | 0.423 | 0.576 | 0.01 | 98.3 | 0.278 |
| FTTransformer-Global | 11.360 | 0.384 | 0.531 | 0.04 | 98.6 | 0.256 |
| NIAQUE-Local | 11.885 | 0.395 | 0.571 | 0.02 | 98.5 | 0.279 |
| NIAQUE-Global | 11.568 | 0.402 | 0.542 | 0.02 | 96.0 | 0.284 |

Table 41: Performance comparison on CONCRETE_COMPRESSIVE_STRENGTH dataset. Lower values are better for all metrics except CI_Coverage (target: 95%). For BIAS, lower absolute values are better.

| Model | SMAPE | AAD | RMSE | BIAS | COV@95 | CRPS |
|---|---|---|---|---|---|---|
| CatBoost-Global | 53.356 | 2.066 | 2.489 | -1.52 | 97.1 | 1.317 |
| CatBoost-Local | 13.376 | 0.531 | 0.706 | -0.22 | 84.5 | 0.390 |
| FTTransformer-Local | 16.743 | 0.610 | 0.808 | -0.25 | 91.9 | 0.402 |
| FTTransformer-Global | 14.757 | 0.587 | 0.734 | -0.27 | 93.0 | 0.402 |
| NIAQUE-Local | 15.572 | 0.600 | 0.760 | -0.22 | 85.6 | 0.412 |
| NIAQUE-Global | 15.097 | 0.602 | 0.773 | -0.24 | 90.3 | 0.418 |

Table 42: Performance comparison on CPMP-2015 dataset. Lower values are better for all metrics except CI_Coverage (target: 95%). For BIAS, lower absolute values are better.

| Model | SMAPE | AAD | RMSE | BIAS | COV@95 | CRPS |
|---|---|---|---|---|---|---|
| CatBoost-Global | 140.648 | 2.600 | 3.943 | -2.10 | 73.9 | 1.809 |
| CatBoost-Local | 93.345 | 0.684 | 1.827 | -0.53 | 87.2 | 0.602 |
| FTTransformer-Local | 90.263 | 0.643 | 1.852 | -0.47 | 92.4 | 0.522 |
| FTTransformer-Global | 89.302 | 0.608 | 1.711 | -0.46 | 95.2 | 0.468 |
| NIAQUE-Local | 86.994 | 0.582 | 1.677 | -0.40 | 94.1 | 0.463 |
| NIAQUE-Global | 84.264 | 0.564 | 1.650 | -0.42 | 94.3 | 0.421 |

Table 43: Performance comparison on CalHousing dataset. Lower values are better for all metrics except CI_Coverage (target: 95%). For BIAS, lower absolute values are better.

| Model | SMAPE | AAD | RMSE | BIAS | COV@95 | CRPS |
|---|---|---|---|---|---|---|
| CatBoost-Global | 35.828 | 1.279 | 1.809 | -0.31 | 96.1 | 0.976 |
| CatBoost-Local | 19.653 | 0.757 | 1.192 | -0.19 | 94.5 | 0.587 |
| FTTransformer-Local | 22.615 | 0.819 | 1.225 | -0.14 | 93.3 | 0.595 |
| FTTransformer-Global | 21.469 | 0.782 | 1.207 | -0.10 | 93.2 | 0.586 |
| NIAQUE-Local | 23.100 | 0.819 | 1.229 | -0.11 | 92.2 | 0.597 |
| NIAQUE-Global | 22.268 | 0.820 | 1.258 | -0.14 | 93.1 | 0.586 |

Table 44: Performance comparison on CampaignUplift dataset. Lower values are better for all metrics except CI_Coverage (target: 95%). For BIAS, lower absolute values are better.

| Model | SMAPE | AAD | RMSE | BIAS | COV@95 | CRPS |
|---|---|---|---|---|---|---|
| CatBoost-Global | 64.266 | 1.163 | 1.343 | 0.82 | 97.5 | 0.862 |
| CatBoost-Local | 11.578 | 0.127 | 0.204 | 0.03 | 92.0 | 0.100 |
| FTTransformer-Local | 13.735 | 0.145 | 0.212 | 0.01 | 92.8 | 0.100 |
| FTTransformer-Global | 10.392 | 0.105 | 0.143 | 0.03 | 95.0 | 0.067 |
| NIAQUE-Local | 10.560 | 0.117 | 0.148 | 0.02 | 96.8 | 0.074 |
| NIAQUE-Global | 10.280 | 0.105 | 0.132 | 0.02 | 96.0 | 0.073 |

Table 45: Performance comparison on CapitalGain dataset. Lower values are better for all metrics except CI_Coverage (target: 95%). For BIAS, lower absolute values are better.

| Model | SMAPE | AAD | RMSE | BIAS | COV@95 | CRPS |
|---|---|---|---|---|---|---|
| CatBoost-Global | 10.193 | 0.116 | 0.749 | -0.09 | 92.5 | 0.131 |
| CatBoost-Local | 9.952 | 0.131 | 0.862 | -0.11 | 94.0 | 0.156 |
| FTTransformer-Local | 12.274 | 0.168 | 0.911 | -0.13 | 95.6 | 0.141 |
| FTTransformer-Global | 9.373 | 0.136 | 0.866 | -0.11 | 98.5 | 0.105 |
| NIAQUE-Local | 9.938 | 0.141 | 0.859 | -0.11 | 95.8 | 0.113 |
| NIAQUE-Global | 9.749 | 0.131 | 0.863 | -0.11 | 94.5 | 0.111 |

Table 46: Performance comparison on Cars dataset. Lower values are better for all metrics except CI_Coverage (target: 95%). For BIAS, lower absolute values are better.

| Model | SMAPE | AAD | RMSE | BIAS | COV@95 | CRPS |
|---|---|---|---|---|---|---|
| CatBoost-Global | 15.195 | 0.160 | 0.272 | -0.01 | 97.9 | 0.129 |
| CatBoost-Local | 10.225 | 0.132 | 0.218 | -0.02 | 94.2 | 0.099 |
| FTTransformer-Local | 12.543 | 0.168 | 0.238 | -0.02 | 95.6 | 0.114 |
| FTTransformer-Global | 10.181 | 0.123 | 0.179 | 0.00 | 93.4 | 0.077 |
| NIAQUE-Local | 10.467 | 0.127 | 0.195 | -0.01 | 94.8 | 0.090 |
| NIAQUE-Global | 10.028 | 0.120 | 0.180 | -0.01 | 94.1 | 0.085 |

Table 47: Performance comparison on Co2Emission dataset. Lower values are better for all metrics except CI_Coverage (target: 95%). For BIAS, lower absolute values are better.

| Model | SMAPE | AAD | RMSE | BIAS | COV@95 | CRPS |
|---|---|---|---|---|---|---|
| CatBoost-Global | 8.610 | 0.281 | 0.556 | -0.07 | 98.6 | 0.446 |
| CatBoost-Local | 2.043 | 0.068 | 0.140 | 0.02 | 94.6 | 0.058 |
| FTTransformer-Local | 4.113 | 0.098 | 0.173 | 0.00 | 99.0 | 0.059 |
| FTTransformer-Global | 1.568 | 0.056 | 0.101 | 0.02 | 97.7 | 0.031 |
| NIAQUE-Local | 2.106 | 0.065 | 0.122 | 0.02 | 97.6 | 0.043 |
| NIAQUE-Global | 1.628 | 0.056 | 0.104 | 0.01 | 98.1 | 0.041 |

Table 48: Performance comparison on Cps88Wages dataset. Lower values are better for all metrics except CI_Coverage (target: 95%). For BIAS, lower absolute values are better.

| Model | SMAPE | AAD | RMSE | BIAS | COV@95 | CRPS |
|---|---|---|---|---|---|---|
| CatBoost-Global | 59.548 | 0.208 | 0.364 | 0.06 | 94.5 | 0.160 |
| CatBoost-Local | 42.185 | 0.205 | 0.322 | -0.05 | 95.0 | 0.155 |
| FTTransformer-Local | 43.132 | 0.260 | 0.386 | -0.05 | 93.7 | 0.161 |
| FTTransformer-Global | 43.363 | 0.203 | 0.319 | -0.04 | 97.1 | 0.133 |
| NIAQUE-Local | 42.586 | 0.212 | 0.336 | -0.05 | 97.6 | 0.146 |
| NIAQUE-Global | 42.222 | 0.205 | 0.323 | -0.05 | 95.0 | 0.146 |

Table 49: Performance comparison on Debutanizer dataset. Lower values are better for all metrics except CI_Coverage (target: 95%). For BIAS, lower absolute values are better.

| Model | SMAPE | AAD | RMSE | BIAS | COV@95 | CRPS |
|---|---|---|---|---|---|---|
| CatBoost-Global | 41.923 | 1.125 | 1.596 | 0.27 | 96.2 | 0.946 |
| CatBoost-Local | 23.651 | 0.646 | 1.183 | -0.26 | 92.5 | 0.511 |
| FTTransformer-Local | 24.355 | 0.601 | 1.075 | -0.05 | 93.6 | 0.444 |
| FTTransformer-Global | 22.909 | 0.564 | 0.990 | -0.04 | 97.8 | 0.411 |
| NIAQUE-Local | 21.878 | 0.512 | 0.904 | -0.05 | 94.8 | 0.378 |
| NIAQUE-Global | 21.048 | 0.515 | 0.878 | -0.04 | 93.8 | 0.381 |

Table 50: Performance comparison on DeltaElevators dataset. Lower values are better for all metrics except CI_Coverage (target: 95%). For BIAS, lower absolute values are better.

| Model | SMAPE | AAD | RMSE | BIAS | COV@95 | CRPS |
|---|---|---|---|---|---|---|
| CatBoost-Global | 8.697 | 0.434 | 0.593 | -0.20 | 97.7 | 0.410 |
| CatBoost-Local | 7.759 | 0.384 | 0.519 | -0.00 | 94.3 | 0.291 |
| FTTransformer-Local | 9.916 | 0.420 | 0.556 | -0.01 | 93.6 | 0.294 |
| FTTransformer-Global | 8.046 | 0.393 | 0.502 | 0.01 | 97.4 | 0.273 |
| NIAQUE-Local | 8.362 | 0.406 | 0.543 | -0.00 | 94.3 | 0.270 |
| NIAQUE-Global | 7.965 | 0.393 | 0.533 | -0.01 | 94.6 | 0.280 |

Table 51: Performance comparison on Diamonds dataset. Lower values are better for all metrics except CI_Coverage (target: 95%). For BIAS, lower absolute values are better.

| Model | SMAPE | AAD | RMSE | BIAS | COV@95 | CRPS |
|---|---|---|---|---|---|---|
| CatBoost-Global | 6.633 | 0.038 | 0.116 | -0.00 | 94.7 | 0.037 |
| CatBoost-Local | 2.052 | 0.021 | 0.045 | -0.00 | 94.5 | 0.020 |
| FTTransformer-Local | 4.179 | 0.059 | 0.084 | -0.01 | 98.5 | 0.027 |
| FTTransformer-Global | 1.739 | 0.015 | 0.025 | 0.01 | 99.0 | 0.001 |
| NIAQUE-Local | 2.275 | 0.026 | 0.048 | 0.00 | 94.3 | 0.013 |
| NIAQUE-Global | 1.807 | 0.017 | 0.033 | 0.00 | 96.8 | 0.013 |

Table 52: Performance comparison on Ele2 dataset. Lower values are better for all metrics except CI_Coverage (target: 95%). For BIAS, lower absolute values are better.

| Model | SMAPE | AAD | RMSE | BIAS | COV@95 | CRPS |
|---|---|---|---|---|---|---|
| CatBoost-Global | 102.764 | 2.074 | 2.331 | 1.28 | 85.8 | 1.448 |
| CatBoost-Local | 11.076 | 0.098 | 0.162 | -0.01 | 86.8 | 0.088 |
| FTTransformer-Local | 10.716 | 0.117 | 0.173 | -0.01 | 95.0 | 0.075 |
| FTTransformer-Global | 8.807 | 0.067 | 0.102 | 0.01 | 97.6 | 0.041 |
| NIAQUE-Local | 8.329 | 0.070 | 0.109 | 0.00 | 98.5 | 0.045 |
| NIAQUE-Global | 7.524 | 0.059 | 0.091 | 0.00 | 98.1 | 0.041 |

Table 53: Performance comparison on Energy_Efficiency dataset. Lower values are better for all metrics except CI_Coverage (target: 95%). For BIAS, lower absolute values are better.

| Model | SMAPE | AAD | RMSE | BIAS | COV@95 | CRPS |
|---|---|---|---|---|---|---|
| CatBoost-Global | 64.138 | 2.377 | 2.902 | -1.16 | 84.4 | 1.520 |
| CatBoost-Local | 4.014 | 0.095 | 0.144 | 0.03 | 96.1 | 0.083 |
| FTTransformer-Local | 9.404 | 0.184 | 0.256 | -0.05 | 99.0 | 0.133 |
| FTTransformer-Global | 8.063 | 0.176 | 0.222 | -0.04 | 97.8 | 0.110 |
| NIAQUE-Local | 9.406 | 0.201 | 0.271 | -0.04 | 99.0 | 0.144 |
| NIAQUE-Global | 9.353 | 0.199 | 0.270 | -0.04 | 97.4 | 0.145 |

Table 54: Performance comparison on Fardamento dataset. Lower values are better for all metrics except CI_Coverage (target: 95%). For BIAS, lower absolute values are better.

| Model | SMAPE | AAD | RMSE | BIAS | COV@95 | CRPS |
|---|---|---|---|---|---|---|
| CatBoost-Global | 102.512 | 0.053 | 0.164 | -0.02 | 77.2 | 0.050 |
| CatBoost-Local | 95.195 | 0.047 | 0.161 | -0.03 | 79.3 | 0.042 |
| FTTransformer-Local | 92.805 | 0.091 | 0.300 | -0.03 | 92.4 | 0.051 |
| FTTransformer-Global | 92.013 | 0.054 | 0.230 | -0.01 | 95.4 | 0.028 |
| NIAQUE-Local | 101.712 | 0.062 | 0.306 | -0.02 | 94.0 | 0.042 |
| NIAQUE-Global | 97.806 | 0.059 | 0.294 | -0.02 | 98.7 | 0.041 |

Table 55: Performance comparison on Fifa dataset. Lower values are better for all metrics except CI_Coverage (target: 95%). For BIAS, lower absolute values are better.

| Model | SMAPE | AAD | RMSE | BIAS | COV@95 | CRPS |
|---|---|---|---|---|---|---|
| CatBoost-Global | 77.475 | 0.130 | 0.324 | -0.05 | 82.5 | 0.096 |
| CatBoost-Local | 72.034 | 0.114 | 0.276 | -0.02 | 89.3 | 0.086 |
| FTTransformer-Local | 73.861 | 0.151 | 0.319 | -0.03 | 95.0 | 0.099 |
| FTTransformer-Global | 76.096 | 0.109 | 0.254 | -0.00 | 93.7 | 0.070 |
| NIAQUE-Local | 74.066 | 0.122 | 0.293 | -0.01 | 96.3 | 0.084 |
| NIAQUE-Global | 74.935 | 0.115 | 0.278 | -0.02 | 94.9 | 0.081 |

Table 56: Performance comparison on FpsBenchmark dataset. Lower values are better for all metrics except CI_Coverage (target: 95%). For BIAS, lower absolute values are better.

| Model | SMAPE | AAD | RMSE | BIAS | COV@95 | CRPS |
|---|---|---|---|---|---|---|
| CatBoost-Global | 30.554 | 0.785 | 1.090 | -0.04 | 99.0 | 0.642 |
| CatBoost-Local | 6.420 | 0.149 | 0.308 | -0.03 | 95.0 | 0.151 |
| FTTransformer-Local | 11.348 | 0.229 | 0.468 | -0.00 | 99.0 | 0.163 |
| FTTransformer-Global | 9.100 | 0.212 | 0.423 | 0.02 | 99.0 | 0.136 |
| NIAQUE-Local | 10.185 | 0.240 | 0.507 | 0.01 | 99.0 | 0.148 |
| NIAQUE-Global | 10.501 | 0.242 | 0.496 | 0.01 | 98.6 | 0.148 |

Table 57: Performance comparison on GemstonePrice dataset. Lower values are better for all metrics except CI_Coverage (target: 95%). For BIAS, lower absolute values are better.

| Model | SMAPE | AAD | RMSE | BIAS | COV@95 | CRPS |
|---|---|---|---|---|---|---|
| CatBoost-Global | 24.207 | 0.272 | 0.512 | -0.02 | 96.2 | 0.210 |
| CatBoost-Local | 11.621 | 0.184 | 0.376 | 0.00 | 95.4 | 0.154 |
| FTTransformer-Local | 13.563 | 0.204 | 0.362 | 0.00 | 95.7 | 0.137 |
| FTTransformer-Global | 11.080 | 0.159 | 0.313 | 0.02 | 92.0 | 0.107 |
| NIAQUE-Local | 11.018 | 0.163 | 0.316 | 0.02 | 94.7 | 0.110 |
| NIAQUE-Global | 10.824 | 0.153 | 0.297 | 0.01 | 94.5 | 0.109 |

Table 58: Performance comparison on GpuKernelPerformance dataset. Lower values are better for all metrics except CI_Coverage (target: 95%). For BIAS, lower absolute values are better.

| Model | SMAPE | AAD | RMSE | BIAS | COV@95 | CRPS |
|---|---|---|---|---|---|---|
| CatBoost-Global | 43.784 | 0.221 | 0.668 | -0.14 | 93.3 | 0.156 |
| CatBoost-Local | 51.879 | 0.255 | 0.757 | -0.15 | 94.1 | 0.199 |
| FTTransformer-Local | 29.646 | 0.168 | 0.398 | -0.01 | 94.1 | 0.114 |
| FTTransformer-Global | 22.349 | 0.089 | 0.249 | 0.01 | 96.2 | 0.061 |
| NIAQUE-Local | 12.550 | 0.052 | 0.129 | 0.00 | 97.6 | 0.033 |
| NIAQUE-Global | 8.881 | 0.026 | 0.064 | 0.00 | 94.2 | 0.019 |

Table 59: Performance comparison on GridStability dataset. Lower values are better for all metrics except CI_Coverage (target: 95%). For BIAS, lower absolute values are better.

| Model | SMAPE | AAD | RMSE | BIAS | COV@95 | CRPS |
|---|---|---|---|---|---|---|
| CatBoost-Global | 27.938 | 1.315 | 1.553 | -0.42 | 98.0 | 0.938 |
| CatBoost-Local | 9.966 | 0.370 | 0.518 | -0.01 | 92.7 | 0.279 |
| FTTransformer-Local | 9.250 | 0.274 | 0.424 | 0.00 | 93.9 | 0.189 |
| FTTransformer-Global | 5.975 | 0.212 | 0.311 | 0.02 | 97.1 | 0.137 |
| NIAQUE-Local | 5.272 | 0.167 | 0.271 | 0.01 | 92.0 | 0.121 |
| NIAQUE-Global | 4.564 | 0.144 | 0.230 | 0.01 | 92.9 | 0.104 |

Table 60: Performance comparison on HouseRent dataset. Lower values are better for all metrics except CI_Coverage (target: 95%). For BIAS, lower absolute values are better.

| Model | SMAPE | AAD | RMSE | BIAS | COV@95 | CRPS |
|---|---|---|---|---|---|---|
| CatBoost-Global | 66.354 | 0.059 | 0.127 | -0.01 | 87.8 | 0.052 |
| CatBoost-Local | 31.500 | 0.028 | 0.068 | -0.01 | 92.8 | 0.022 |
| FTTransformer-Local | 31.390 | 0.066 | 0.113 | -0.01 | 95.5 | 0.031 |
| FTTransformer-Global | 29.643 | 0.024 | 0.057 | 0.00 | 94.5 | 0.006 |
| NIAQUE-Local | 30.469 | 0.035 | 0.082 | -0.00 | 93.6 | 0.019 |
| NIAQUE-Global | 30.273 | 0.026 | 0.067 | -0.00 | 93.3 | 0.019 |

Table 61: Performance comparison on IEEE80211aaGATS dataset. Lower values are better for all metrics except CI_Coverage (target: 95%). For BIAS, lower absolute values are better.

| Model | SMAPE | AAD | RMSE | BIAS | COV@95 | CRPS |
|---|---|---|---|---|---|---|
| CatBoost-Global | 26.743 | 1.000 | 1.585 | 0.36 | 84.2 | 0.866 |
| CatBoost-Local | 12.939 | 0.261 | 0.542 | -0.05 | 94.0 | 0.338 |
| FTTransformer-Local | 10.248 | 0.250 | 0.452 | -0.01 | 96.4 | 0.227 |
| FTTransformer-Global | 7.344 | 0.190 | 0.392 | 0.01 | 96.4 | 0.181 |
| NIAQUE-Local | 5.817 | 0.182 | 0.367 | 0.00 | 99.0 | 0.139 |
| NIAQUE-Global | 4.736 | 0.167 | 0.319 | 0.00 | 97.2 | 0.122 |

Table 62: Performance comparison on Infrared_Thermography_Temperature dataset. Lower values are better for all metrics except CI_Coverage (target: 95%). For BIAS, lower absolute values are better.

| Model | SMAPE | AAD | RMSE | BIAS | COV@95 | CRPS |
|---|---|---|---|---|---|---|
| CatBoost-Global | 23.255 | 0.743 | 1.103 | -0.20 | 100.0 | 0.780 |
| CatBoost-Local | 14.479 | 0.416 | 0.529 | 0.02 | 90.2 | 0.292 |
| FTTransformer-Local | 16.384 | 0.473 | 0.599 | 0.02 | 98.5 | 0.303 |
| FTTransformer-Global | 14.692 | 0.428 | 0.524 | 0.04 | 96.4 | 0.277 |
| NIAQUE-Local | 15.061 | 0.424 | 0.560 | 0.03 | 94.6 | 0.291 |
| NIAQUE-Global | 14.471 | 0.420 | 0.542 | 0.03 | 95.1 | 0.295 |

Table 63: Performance comparison on KindleBooks dataset. Lower values are better for all metrics except CI_Coverage (target: 95%). For BIAS, lower absolute values are better.

| Model | SMAPE | AAD | RMSE | BIAS | COV@95 | CRPS |
|---|---|---|---|---|---|---|
| CatBoost-Global | 33.753 | 0.055 | 0.165 | -0.02 | 89.5 | 0.046 |
| CatBoost-Local | 29.304 | 0.046 | 0.159 | -0.02 | 94.5 | 0.042 |
| FTTransformer-Local | 32.212 | 0.082 | 0.198 | -0.02 | 98.1 | 0.050 |
| FTTransformer-Global | 28.591 | 0.042 | 0.123 | -0.00 | 96.6 | 0.020 |
| NIAQUE-Local | 30.017 | 0.052 | 0.156 | -0.01 | 97.7 | 0.034 |
| NIAQUE-Global | 28.834 | 0.043 | 0.143 | -0.01 | 94.8 | 0.032 |

Table 64: Performance comparison on Kinematics dataset. Lower values are better for all metrics except CI_Coverage (target: 95%). For BIAS, lower absolute values are better.

| Model | SMAPE | AAD | RMSE | BIAS | COV@95 | CRPS |
|---|---|---|---|---|---|---|
| CatBoost-Global | 30.256 | 1.334 | 1.687 | -0.58 | 96.8 | 0.969 |
| CatBoost-Local | 20.933 | 0.832 | 1.102 | 0.11 | 92.4 | 0.611 |
| FTTransformer-Local | 17.407 | 0.660 | 0.780 | -0.01 | 87.3 | 0.453 |
| FTTransformer-Global | 12.528 | 0.522 | 0.645 | 0.01 | 86.5 | 0.356 |
| NIAQUE-Local | 11.584 | 0.414 | 0.542 | 0.00 | 86.9 | 0.282 |
| NIAQUE-Global | 10.113 | 0.371 | 0.474 | 0.00 | 82.2 | 0.267 |

Table 65: Performance comparison on KingCountyHousing dataset. Lower values are better for all metrics except CI_Coverage (target: 95%). For BIAS, lower absolute values are better.

| Model | SMAPE | AAD | RMSE | BIAS | COV@95 | CRPS |
|---|---|---|---|---|---|---|
| CatBoost-Global | 26.667 | 0.172 | 0.387 | -0.05 | 96.2 | 0.129 |
| CatBoost-Local | 15.617 | 0.095 | 0.176 | -0.02 | 93.8 | 0.075 |
| FTTransformer-Local | 25.275 | 0.166 | 0.253 | -0.03 | 96.6 | 0.108 |
| FTTransformer-Global | 23.652 | 0.128 | 0.198 | -0.01 | 98.6 | 0.081 |
| NIAQUE-Local | 26.563 | 0.160 | 0.245 | -0.02 | 95.3 | 0.104 |
| NIAQUE-Global | 27.154 | 0.152 | 0.236 | -0.02 | 95.2 | 0.106 |

Table 66: Performance comparison on LaptopPrice dataset. Lower values are better for all metrics except CI_Coverage (target: 95%). For BIAS, lower absolute values are better.

| Model | SMAPE | AAD | RMSE | BIAS | COV@95 | CRPS |
|---|---|---|---|---|---|---|
| CatBoost-Global | 67.065 | 1.348 | 1.531 | 0.98 | 97.7 | 0.995 |
| CatBoost-Local | 20.932 | 0.372 | 0.598 | -0.12 | 91.4 | 0.271 |
| FTTransformer-Local | 22.567 | 0.402 | 0.621 | -0.11 | 87.6 | 0.274 |
| FTTransformer-Global | 19.003 | 0.358 | 0.572 | -0.09 | 86.4 | 0.235 |
| NIAQUE-Local | 19.326 | 0.367 | 0.582 | -0.11 | 84.0 | 0.261 |
| NIAQUE-Global | 18.936 | 0.354 | 0.558 | -0.10 | 84.4 | 0.257 |

Table 67: Performance comparison on Laser dataset. Lower values are better for all metrics except CI_Coverage (target: 95%). For BIAS, lower absolute values are better.

| Model | SMAPE | AAD | RMSE | BIAS | COV@95 | CRPS |
|---|---|---|---|---|---|---|
| CatBoost-Global | 66.495 | 1.731 | 2.052 | 0.32 | 99.0 | 1.185 |
| CatBoost-Local | 6.804 | 0.138 | 0.397 | 0.05 | 91.0 | 0.171 |
| FTTransformer-Local | 7.275 | 0.142 | 0.289 | -0.00 | 93.3 | 0.114 |
| FTTransformer-Global | 4.282 | 0.082 | 0.193 | 0.02 | 98.1 | 0.072 |
| NIAQUE-Local | 4.375 | 0.075 | 0.151 | 0.01 | 97.7 | 0.056 |
| NIAQUE-Global | 3.770 | 0.064 | 0.116 | 0.01 | 98.0 | 0.047 |

Table 68: Performance comparison on LifeExpectancy dataset. Lower values are better for all metrics except CI_Coverage (target: 95%). For BIAS, lower absolute values are better.

| Model | SMAPE | AAD | RMSE | BIAS | COV@95 | CRPS |
|---|---|---|---|---|---|---|
| CatBoost-Global | 15.881 | 0.890 | 1.184 | -0.52 | 99.3 | 0.687 |
| CatBoost-Local | 5.229 | 0.269 | 0.412 | 0.01 | 94.2 | 0.228 |
| FTTransformer-Local | 6.853 | 0.269 | 0.407 | -0.03 | 95.9 | 0.198 |
| FTTransformer-Global | 3.824 | 0.211 | 0.347 | -0.01 | 98.7 | 0.149 |
| NIAQUE-Local | 4.072 | 0.203 | 0.337 | -0.02 | 99.0 | 0.146 |
| NIAQUE-Global | 3.494 | 0.190 | 0.313 | -0.02 | 93.9 | 0.142 |

Table 69: Performance comparison on LoanAmount dataset. Lower values are better for all metrics except CI_Coverage (target: 95%). For BIAS, lower absolute values are better.

| Model | SMAPE | AAD | RMSE | BIAS | COV@95 | CRPS |
|---|---|---|---|---|---|---|
| CatBoost-Global | 49.617 | 0.232 | 0.529 | 0.06 | 81.8 | 0.231 |
| CatBoost-Local | 46.025 | 0.213 | 0.531 | 0.12 | 78.3 | 0.213 |
| FTTransformer-Local | 38.888 | 0.250 | 0.587 | 0.09 | 93.3 | 0.197 |
| FTTransformer-Global | 32.407 | 0.189 | 0.513 | 0.11 | 92.0 | 0.167 |
| NIAQUE-Local | 27.394 | 0.205 | 0.543 | 0.10 | 92.0 | 0.152 |
| NIAQUE-Global | 26.088 | 0.191 | 0.518 | 0.11 | 97.4 | 0.153 |

Table 70: Performance comparison on MarketingCampaign dataset. Lower values are better for all metrics except CI_Coverage (target: 95%). For BIAS, lower absolute values are better.

| Model | SMAPE | AAD | RMSE | BIAS | COV@95 | CRPS |
|---|---|---|---|---|---|---|
| CatBoost-Global | 34.478 | 0.594 | 0.859 | 0.18 | 99.3 | 0.452 |
| CatBoost-Local | 17.050 | 0.353 | 0.719 | -0.09 | 66.8 | 0.272 |
| FTTransformer-Local | 18.104 | 0.393 | 0.791 | -0.07 | 88.4 | 0.273 |
| FTTransformer-Global | 15.907 | 0.298 | 0.743 | -0.04 | 90.5 | 0.219 |
| NIAQUE-Local | 15.505 | 0.334 | 0.775 | -0.05 | 87.2 | 0.234 |
| NIAQUE-Global | 14.484 | 0.321 | 0.731 | -0.05 | 93.0 | 0.230 |

Table 71: Performance comparison on MedicalCost dataset. Lower values are better for all metrics except CI_Coverage (target: 95%). For BIAS, lower absolute values are better.

| Model | SMAPE | AAD | RMSE | BIAS | COV@95 | CRPS |
|---|---|---|---|---|---|---|
| CatBoost-Global | 96.062 | 2.061 | 2.317 | 0.97 | 86.6 | 1.412 |
| CatBoost-Local | 15.806 | 0.364 | 0.845 | -0.12 | 94.8 | 0.369 |
| FTTransformer-Local | 17.090 | 0.346 | 0.823 | -0.16 | 92.9 | 0.291 |
| FTTransformer-Global | 14.432 | 0.281 | 0.764 | -0.15 | 93.7 | 0.234 |
| NIAQUE-Local | 14.276 | 0.271 | 0.753 | -0.12 | 94.7 | 0.220 |
| NIAQUE-Global | 13.666 | 0.260 | 0.757 | -0.15 | 94.0 | 0.208 |

Table 72: Performance comparison on MiamiHousing dataset. Lower values are better for all metrics except CI_Coverage (target: 95%). For BIAS, lower absolute values are better.

| Model | SMAPE | AAD | RMSE | BIAS | COV@95 | CRPS |
|---|---|---|---|---|---|---|
| CatBoost-Global | 29.994 | 0.390 | 0.779 | -0.06 | 97.3 | 0.299 |
| CatBoost-Local | 15.589 | 0.194 | 0.381 | -0.03 | 95.2 | 0.157 |
| FTTransformer-Local | 17.107 | 0.212 | 0.375 | -0.02 | 92.1 | 0.145 |
| FTTransformer-Global | 13.926 | 0.155 | 0.312 | -0.00 | 93.9 | 0.116 |
| NIAQUE-Local | 13.441 | 0.169 | 0.313 | -0.01 | 97.8 | 0.116 |
| NIAQUE-Global | 13.173 | 0.155 | 0.292 | -0.01 | 92.8 | 0.110 |

Table 73: Performance comparison on Mortgage dataset. Lower values are better for all metrics except CI_Coverage (target: 95%). For BIAS, lower absolute values are better.

| Model | SMAPE | AAD | RMSE | BIAS | COV@95 | CRPS |
|---|---|---|---|---|---|---|
| CatBoost-Global | 64.387 | 1.814 | 2.337 | -0.02 | 85.7 | 1.276 |
| CatBoost-Local | 6.670 | 0.077 | 0.113 | 0.00 | 92.4 | 0.082 |
| FTTransformer-Local | 9.029 | 0.115 | 0.162 | -0.01 | 99.0 | 0.075 |
| FTTransformer-Global | 6.582 | 0.076 | 0.110 | 0.00 | 98.6 | 0.048 |
| NIAQUE-Local | 6.918 | 0.085 | 0.133 | -0.00 | 97.5 | 0.056 |
| NIAQUE-Global | 6.398 | 0.077 | 0.119 | -0.00 | 97.1 | 0.053 |

Table 74: Performance comparison on NASA-PHM2008 dataset. Lower values are better for all metrics except CI_Coverage (target: 95%). For BIAS, lower absolute values are better.

| Model | SMAPE | AAD | RMSE | BIAS | COV@95 | CRPS |
|---|---|---|---|---|---|---|
| CatBoost-Global | 37.375 | 0.987 | 1.326 | -0.31 | 97.3 | 0.718 |
| CatBoost-Local | 31.464 | 0.834 | 1.143 | -0.17 | 93.8 | 0.596 |
| FTTransformer-Local | 33.765 | 0.788 | 1.232 | -0.15 | 92.1 | 0.615 |
| FTTransformer-Global | 28.671 | 0.781 | 1.130 | -0.14 | 93.5 | 0.566 |
| NIAQUE-Local | 30.348 | 0.854 | 1.162 | -0.14 | 95.4 | 0.555 |
| NIAQUE-Global | 29.978 | 0.813 | 1.127 | -0.13 | 94.0 | 0.564 |

Table 75: Performance comparison on NewFuelCar dataset. Lower values are better for all metrics except CI_Coverage (target: 95%). For BIAS, lower absolute values are better.

| Model | SMAPE | AAD | RMSE | BIAS | COV@95 | CRPS |
|---|---|---|---|---|---|---|
| CatBoost-Global | 18.349 | 0.278 | 0.461 | 0.05 | 97.2 | 0.310 |
| CatBoost-Local | 2.649 | 0.061 | 0.107 | 0.02 | 93.3 | 0.057 |
| FTTransformer-Local | 4.391 | 0.092 | 0.138 | -0.00 | 96.3 | 0.056 |
| FTTransformer-Global | 1.779 | 0.047 | 0.077 | 0.02 | 98.5 | 0.028 |
| NIAQUE-Local | 2.034 | 0.055 | 0.094 | 0.01 | 98.5 | 0.036 |
| NIAQUE-Global | 1.550 | 0.046 | 0.081 | 0.01 | 95.5 | 0.034 |

Table 76: Performance comparison on NewsPopularity dataset. Lower values are better for all metrics except CI_Coverage (target: 95%). For BIAS, lower absolute values are better.

| Model | SMAPE | AAD | RMSE | BIAS | COV@95 | CRPS |
|---|---|---|---|---|---|---|
| CatBoost-Global | 53.910 | 0.028 | 0.157 | -0.02 | 87.0 | 0.025 |
| CatBoost-Local | 51.060 | 0.035 | 0.194 | -0.03 | 94.2 | 0.034 |
| FTTransformer-Local | 54.237 | 0.073 | 0.237 | -0.04 | 99.0 | 0.042 |
| FTTransformer-Global | 48.657 | 0.032 | 0.185 | -0.02 | 96.9 | 0.016 |
| NIAQUE-Local | 51.937 | 0.043 | 0.209 | -0.02 | 95.9 | 0.030 |
| NIAQUE-Global | 51.099 | 0.035 | 0.194 | -0.03 | 95.7 | 0.028 |

Table 77: Performance comparison on PRODUCTIVITY dataset. Lower values are better for all metrics except CI_Coverage (target: 95%). For BIAS, lower absolute values are better.

| Model | SMAPE | AAD | RMSE | BIAS | COV@95 | CRPS |
|---|---|---|---|---|---|---|
| CatBoost-Global | 63.361 | 2.803 | 3.161 | -2.52 | 93.3 | 1.674 |
| CatBoost-Local | 20.267 | 0.840 | 1.418 | 0.16 | 91.7 | 0.677 |
| FTTransformer-Local | 22.318 | 0.816 | 1.361 | -0.03 | 91.0 | 0.598 |
| FTTransformer-Global | 18.342 | 0.761 | 1.317 | -0.02 | 91.9 | 0.579 |
| NIAQUE-Local | 17.589 | 0.733 | 1.311 | -0.03 | 92.1 | 0.574 |
| NIAQUE-Global | 17.487 | 0.727 | 1.317 | -0.03 | 90.8 | 0.555 |

Table 78: Performance comparison on Parkinsons_Telemonitoring dataset. Lower values are better for all metrics except CI_Coverage (target: 95%). For BIAS, lower absolute values are better.

| Model | SMAPE | AAD | RMSE | BIAS | COV@95 | CRPS |
|---|---|---|---|---|---|---|
| CatBoost-Global | 42.713 | 1.699 | 2.220 | -0.98 | 94.7 | 1.189 |
| CatBoost-Local | 19.506 | 0.667 | 0.980 | 0.03 | 94.9 | 0.495 |
| FTTransformer-Local | 17.967 | 0.573 | 0.871 | -0.12 | 95.8 | 0.409 |
| FTTransformer-Global | 15.604 | 0.474 | 0.885 | -0.09 | 95.3 | 0.339 |
| NIAQUE-Local | 14.407 | 0.428 | 0.836 | -0.12 | 95.9 | 0.303 |
| NIAQUE-Global | 13.267 | 0.404 | 0.818 | -0.12 | 94.7 | 0.287 |

Table 79: Performance comparison on Pole dataset. Lower values are better for all metrics except CI_Coverage (target: 95%). For BIAS, lower absolute values are better.

| Model | SMAPE | AAD | RMSE | BIAS | COV@95 | CRPS |
|---|---|---|---|---|---|---|
| CatBoost-Global | 133.955 | 0.281 | 0.678 | -0.01 | 58.5 | 0.393 |
| CatBoost-Local | 139.171 | 0.719 | 1.462 | 0.11 | 60.5 | 0.699 |
| FTTransformer-Local | 131.589 | 0.418 | 0.921 | -0.03 | 87.1 | 0.400 |
| FTTransformer-Global | 135.896 | 0.301 | 0.706 | -0.01 | 88.7 | 0.264 |
| NIAQUE-Local | 133.800 | 0.182 | 0.460 | -0.02 | 90.0 | 0.144 |
| NIAQUE-Global | 133.544 | 0.125 | 0.368 | -0.02 | 98.5 | 0.091 |

Table 80: Performance comparison on ProteinTertiary dataset. Lower values are better for all metrics except CI_Coverage (target: 95%). For BIAS, lower absolute values are better.

| Model | SMAPE | AAD | RMSE | BIAS | COV@95 | CRPS |
|---|---|---|---|---|---|---|
| CatBoost-Global | 61.606 | 1.996 | 2.515 | -0.48 | 93.9 | 1.374 |
| CatBoost-Local | 47.676 | 1.575 | 2.244 | -0.29 | 93.7 | 1.165 |
| FTTransformer-Local | 42.046 | 1.288 | 2.124 | -0.11 | 94.6 | 0.969 |
| FTTransformer-Global | 38.191 | 1.248 | 1.990 | -0.07 | 95.1 | 0.892 |
| NIAQUE-Local | 33.479 | 1.148 | 1.914 | -0.10 | 97.0 | 0.804 |
| NIAQUE-Global | 32.642 | 1.136 | 1.874 | -0.10 | 93.8 | 0.807 |

Table 81: Performance comparison on PumaDyn32nh dataset. Lower values are better for all metrics except CI_Coverage (target: 95%). For BIAS, lower absolute values are better.

| Model | SMAPE | AAD | RMSE | BIAS | COV@95 | CRPS |
|---|---|---|---|---|---|---|
| CatBoost-Global | 23.296 | 1.045 | 1.338 | -0.56 | 97.3 | 0.755 |
| CatBoost-Local | 17.175 | 0.735 | 0.925 | -0.05 | 92.1 | 0.530 |
| FTTransformer-Local | 21.163 | 0.746 | 0.964 | -0.02 | 86.4 | 0.562 |
| FTTransformer-Global | 17.833 | 0.737 | 0.978 | -0.01 | 88.3 | 0.536 |
| NIAQUE-Local | 18.709 | 0.773 | 0.939 | -0.01 | 88.9 | 0.545 |
| NIAQUE-Global | 18.663 | 0.762 | 0.953 | -0.01 | 86.2 | 0.545 |

Table 82: Performance comparison on QsarFishToxicity dataset. Lower values are better for all metrics except CI_Coverage (target: 95%). For BIAS, lower absolute values are better.

| Model | SMAPE | AAD | RMSE | BIAS | COV@95 | CRPS |
|---|---|---|---|---|---|---|
| CatBoost-Global | 45.093 | 1.680 | 2.043 | -1.27 | 98.9 | 1.123 |
| CatBoost-Local | 20.804 | 0.709 | 1.004 | 0.10 | 84.6 | 0.546 |
| FTTransformer-Local | 23.345 | 0.757 | 1.093 | 0.02 | 87.6 | 0.521 |
| FTTransformer-Global | 20.162 | 0.720 | 1.005 | 0.05 | 84.7 | 0.509 |
| NIAQUE-Local | 20.736 | 0.707 | 0.981 | 0.04 | 85.9 | 0.535 |
| NIAQUE-Global | 20.313 | 0.704 | 0.963 | 0.03 | 84.6 | 0.515 |

Table 83: Performance comparison on RedWine dataset. Lower values are better for all metrics except CI_Coverage (target: 95%). For BIAS, lower absolute values are better.

| Model | SMAPE | AAD | RMSE | BIAS | COV@95 | CRPS |
|---|---|---|---|---|---|---|
| CatBoost-Global | 28.508 | 1.486 | 2.075 | -1.32 | 98.1 | 1.147 |
| CatBoost-Local | 17.303 | 0.909 | 1.238 | -0.07 | 93.8 | 0.683 |
| FTTransformer-Local | 18.635 | 0.914 | 1.236 | -0.19 | 97.6 | 0.637 |
| FTTransformer-Global | 16.686 | 0.863 | 1.248 | -0.20 | 97.9 | 0.599 |
| NIAQUE-Local | 16.889 | 0.874 | 1.339 | -0.16 | 97.2 | 0.608 |
| NIAQUE-Global | 16.112 | 0.844 | 1.302 | -0.18 | 94.4 | 0.582 |

Table 84: Performance comparison on Sales dataset. Lower values are better for all metrics except CI_Coverage (target: 95%). For BIAS, lower absolute values are better.

| Model | SMAPE | AAD | RMSE | BIAS | COV@95 | CRPS |
|---|---|---|---|---|---|---|
| CatBoost-Global | 63.524 | 0.277 | 0.471 | -0.04 | 87.0 | 0.221 |
| CatBoost-Local | 58.466 | 0.227 | 0.415 | -0.09 | 84.9 | 0.163 |
| FTTransformer-Local | 61.860 | 0.253 | 0.414 | -0.06 | 99.0 | 0.167 |
| FTTransformer-Global | 57.553 | 0.212 | 0.330 | -0.04 | 95.3 | 0.141 |
| NIAQUE-Local | 58.734 | 0.214 | 0.348 | -0.06 | 93.4 | 0.146 |
| NIAQUE-Global | 57.967 | 0.205 | 0.334 | -0.05 | 96.4 | 0.143 |

Table 85: Performance comparison on Sarcos dataset. Lower values are better for all metrics except CI_Coverage (target: 95%). For BIAS, lower absolute values are better.

| Model | SMAPE | AAD | RMSE | BIAS | COV@95 | CRPS |
|---|---|---|---|---|---|---|
| CatBoost-Global | 5.244 | 0.252 | 0.408 | -0.03 | 98.4 | 0.245 |
| CatBoost-Local | 3.257 | 0.151 | 0.230 | 0.00 | 94.2 | 0.122 |
| FTTransformer-Local | 4.846 | 0.166 | 0.226 | -0.01 | 95.4 | 0.098 |
| FTTransformer-Global | 2.112 | 0.106 | 0.152 | 0.01 | 97.2 | 0.071 |
| NIAQUE-Local | 2.572 | 0.109 | 0.162 | 0.00 | 96.0 | 0.073 |
| NIAQUE-Global | 1.985 | 0.094 | 0.141 | 0.00 | 94.0 | 0.068 |

Table 86: Performance comparison on SaudiArabiaCars dataset. Lower values are better for all metrics except CI_Coverage (target: 95%). For BIAS, lower absolute values are better.

| Model | SMAPE | AAD | RMSE | BIAS | COV@95 | CRPS |
|---|---|---|---|---|---|---|
| CatBoost-Global | 50.282 | 0.504 | 0.986 | -0.05 | 96.0 | 0.398 |
| CatBoost-Local | 23.424 | 0.246 | 0.706 | -0.10 | 93.8 | 0.208 |
| FTTransformer-Local | 28.061 | 0.329 | 0.747 | -0.09 | 91.3 | 0.227 |
| FTTransformer-Global | 27.649 | 0.278 | 0.662 | -0.05 | 93.2 | 0.213 |
| NIAQUE-Local | 28.392 | 0.287 | 0.692 | -0.07 | 89.7 | 0.207 |
| NIAQUE-Global | 29.542 | 0.285 | 0.672 | -0.06 | 89.3 | 0.215 |

Table 87: Performance comparison on SongPopularity dataset. Lower values are better for all metrics except CI_Coverage (target: 95%). For BIAS, lower absolute values are better.

| Model | SMAPE | AAD | RMSE | BIAS | COV@95 | CRPS |
|---|---|---|---|---|---|---|
| CatBoost-Global | 38.240 | 1.738 | 2.164 | -0.19 | 94.2 | 1.285 |
| CatBoost-Local | 35.011 | 1.570 | 2.060 | 0.38 | 94.3 | 1.170 |
| FTTransformer-Local | 38.622 | 1.704 | 1.963 | 0.36 | 94.6 | 1.130 |
| FTTransformer-Global | 35.259 | 1.563 | 2.100 | 0.33 | 94.9 | 1.079 |
| NIAQUE-Local | 37.494 | 1.533 | 2.059 | 0.33 | 96.2 | 1.102 |
| NIAQUE-Global | 35.045 | 1.542 | 2.044 | 0.35 | 94.1 | 1.101 |

Table 88: Performance comparison on SpaceGa dataset. Lower values are better for all metrics except CI_Coverage (target: 95%). For BIAS, lower absolute values are better.

| Model | SMAPE | AAD | RMSE | BIAS | COV@95 | CRPS |
|---|---|---|---|---|---|---|
| CatBoost-Global | 6.846 | 0.553 | 0.703 | -0.30 | 94.5 | 0.412 |
| CatBoost-Local | 3.350 | 0.265 | 0.400 | 0.03 | 92.6 | 0.202 |
| FTTransformer-Local | 6.763 | 0.392 | 0.534 | 0.04 | 96.4 | 0.283 |
| FTTransformer-Global | 4.684 | 0.370 | 0.519 | 0.06 | 94.0 | 0.261 |
| NIAQUE-Local | 5.636 | 0.434 | 0.581 | 0.05 | 95.7 | 0.312 |
| NIAQUE-Global | 5.574 | 0.448 | 0.588 | 0.06 | 95.8 | 0.315 |

Table 89: Performance comparison on Student_Performance dataset. Lower values are better for all metrics except CI_Coverage (target: 95%). For BIAS, lower absolute values are better.

| Model | SMAPE | AAD | RMSE | BIAS | COV@95 | CRPS |
|---|---|---|---|---|---|---|
| CatBoost-Global | 78.337 | 3.903 | 4.176 | -3.81 | 72.3 | 2.336 |
| CatBoost-Local | 17.487 | 1.052 | 1.421 | -0.32 | 87.7 | 0.837 |
| FTTransformer-Local | 20.403 | 1.120 | 1.427 | -0.11 | 86.2 | 0.809 |
| FTTransformer-Global | 17.268 | 1.011 | 1.434 | -0.08 | 87.3 | 0.787 |
| NIAQUE-Local | 17.589 | 1.050 | 1.410 | -0.10 | 82.8 | 0.754 |
| NIAQUE-Global | 16.656 | 1.015 | 1.424 | -0.11 | 87.7 | 0.768 |

Table 90: Performance comparison on Sulfur dataset. Lower values are better for all metrics except CI_Coverage (target: 95%). For BIAS, lower absolute values are better.

| Model | SMAPE | AAD | RMSE | BIAS | COV@95 | CRPS |
|---|---|---|---|---|---|---|
| CatBoost-Global | 20.719 | 0.161 | 0.289 | 0.04 | 98.4 | 0.133 |
| CatBoost-Local | 14.665 | 0.116 | 0.203 | -0.01 | 93.6 | 0.086 |
| FTTransformer-Local | 15.814 | 0.151 | 0.221 | -0.01 | 94.1 | 0.090 |
| FTTransformer-Global | 13.591 | 0.108 | 0.166 | 0.01 | 92.8 | 0.065 |
| NIAQUE-Local | 14.458 | 0.108 | 0.178 | 0.01 | 94.1 | 0.075 |
| NIAQUE-Global | 13.703 | 0.104 | 0.158 | 0.00 | 95.8 | 0.073 |

Table 91: Performance comparison on TETOUAN dataset. Lower values are better for all metrics except CI_Coverage (target: 95%). For BIAS, lower absolute values are better.

| Model | SMAPE | AAD | RMSE | BIAS | COV@95 | CRPS |
|---|---|---|---|---|---|---|
| CatBoost-Global | 14.156 | 0.625 | 0.824 | -0.05 | 98.8 | 0.534 |
| CatBoost-Local | 10.438 | 0.448 | 0.629 | -0.01 | 94.9 | 0.334 |
| FTTransformer-Local | 11.680 | 0.416 | 0.583 | -0.01 | 95.7 | 0.313 |
| FTTransformer-Global | 9.089 | 0.369 | 0.543 | 0.01 | 99.0 | 0.271 |
| NIAQUE-Local | 9.059 | 0.360 | 0.546 | 0.00 | 92.9 | 0.260 |
| NIAQUE-Global | 8.316 | 0.359 | 0.515 | -0.00 | 93.5 | 0.255 |

Table 92: Performance comparison on TitanicFare dataset. Lower values are better for all metrics except CI_Coverage (target: 95%). For BIAS, lower absolute values are better.

| Model | SMAPE | AAD | RMSE | BIAS | COV@95 | CRPS |
|---|---|---|---|---|---|---|
| CatBoost-Global | 92.115 | 0.590 | 0.759 | 0.45 | 78.6 | 0.441 |
| CatBoost-Local | 22.778 | 0.175 | 0.511 | -0.04 | 93.9 | 0.173 |
| FTTransformer-Local | 25.910 | 0.208 | 0.530 | -0.05 | 94.1 | 0.153 |
| FTTransformer-Global | 21.883 | 0.156 | 0.439 | -0.03 | 90.7 | 0.114 |
| NIAQUE-Local | 22.496 | 0.164 | 0.464 | -0.04 | 90.9 | 0.122 |
| NIAQUE-Global | 21.963 | 0.155 | 0.450 | -0.04 | 90.8 | 0.117 |

Table 93: Performance comparison on Transcoding dataset. Lower values are better for all metrics except CI_Coverage (target: 95%). For BIAS, lower absolute values are better.

| Model | SMAPE | AAD | RMSE | BIAS | COV@95 | CRPS |
|---|---|---|---|---|---|---|
| CatBoost-Global | 28.703 | 0.120 | 0.385 | -0.08 | 93.8 | 0.086 |
| CatBoost-Local | 21.052 | 0.119 | 0.367 | -0.06 | 94.3 | 0.093 |
| FTTransformer-Local | 15.672 | 0.112 | 0.248 | -0.00 | 95.0 | 0.062 |
| FTTransformer-Global | 11.171 | 0.052 | 0.149 | 0.01 | 90.7 | 0.028 |
| NIAQUE-Local | 9.435 | 0.044 | 0.109 | 0.01 | 91.9 | 0.027 |
| NIAQUE-Global | 7.671 | 0.029 | 0.072 | 0.00 | 95.1 | 0.020 |

Table 94: Performance comparison on Treasury dataset. Lower values are better for all metrics except CI_Coverage (target: 95%). For BIAS, lower absolute values are better.

| Model | SMAPE | AAD | RMSE | BIAS | COV@95 | CRPS |
|---|---|---|---|---|---|---|
| CatBoost-Global | 70.550 | 1.693 | 2.103 | 0.41 | 83.8 | 1.225 |
| CatBoost-Local | 7.815 | 0.101 | 0.211 | -0.00 | 84.8 | 0.091 |
| FTTransformer-Local | 9.984 | 0.129 | 0.229 | -0.04 | 91.9 | 0.087 |
| FTTransformer-Global | 7.103 | 0.086 | 0.161 | -0.01 | 95.7 | 0.055 |
| NIAQUE-Local | 7.784 | 0.091 | 0.163 | -0.02 | 95.8 | 0.066 |
| NIAQUE-Global | 7.227 | 0.082 | 0.145 | -0.03 | 95.2 | 0.062 |

Table 95: Performance comparison on UberFare dataset. Lower values are better for all metrics except CI_Coverage (target: 95%). For BIAS, lower absolute values are better.

| Model | SMAPE | AAD | RMSE | BIAS | COV@95 | CRPS |
|---|---|---|---|---|---|---|
| CatBoost-Global | 4.803 | 0.080 | 0.180 | -0.02 | 94.2 | 0.061 |
| CatBoost-Local | 9.736 | 0.122 | 0.304 | -0.04 | 94.2 | 0.096 |
| FTTransformer-Local | 17.967 | 0.238 | 0.498 | -0.16 | 96.0 | 0.168 |
| FTTransformer-Global | 18.255 | 0.226 | 0.451 | -0.12 | 94.5 | 0.158 |
| NIAQUE-Local | 21.310 | 0.266 | 0.528 | -0.13 | 96.9 | 0.202 |
| NIAQUE-Global | 21.892 | 0.274 | 0.541 | -0.16 | 94.1 | 0.202 |

Table 96: Performance comparison on UsedCar dataset. Lower values are better for all metrics except CI_Coverage (target: 95%). For BIAS, lower absolute values are better.

| Model | SMAPE | AAD | RMSE | BIAS | COV@95 | CRPS |
|---|---|---|---|---|---|---|
| CatBoost-Global | 47.910 | 0.277 | 0.469 | 0.03 | 97.0 | 0.215 |
| CatBoost-Local | 19.392 | 0.111 | 0.237 | -0.02 | 93.4 | 0.088 |
| FTTransformer-Local | 23.181 | 0.152 | 0.287 | -0.01 | 90.2 | 0.102 |
| FTTransformer-Global | 21.556 | 0.116 | 0.235 | 0.01 | 94.2 | 0.073 |
| NIAQUE-Local | 23.584 | 0.129 | 0.266 | 0.00 | 93.7 | 0.089 |
| NIAQUE-Global | 22.921 | 0.122 | 0.249 | 0.00 | 91.5 | 0.088 |

Table 97: Performance comparison on Vehicle dataset. Lower values are better for all metrics except CI_Coverage (target: 95%). For BIAS, lower absolute values are better.

| Model | SMAPE | AAD | RMSE | BIAS | COV@95 | CRPS |
|---|---|---|---|---|---|---|
| CatBoost-Global | 69.582 | 0.378 | 0.584 | 0.13 | 93.7 | 0.310 |
| CatBoost-Local | 15.964 | 0.110 | 0.299 | -0.02 | 94.2 | 0.091 |
| FTTransformer-Local | 21.671 | 0.166 | 0.355 | -0.01 | 98.2 | 0.107 |
| FTTransformer-Global | 21.950 | 0.125 | 0.282 | 0.01 | 96.3 | 0.079 |
| NIAQUE-Local | 23.717 | 0.137 | 0.308 | 0.00 | 97.9 | 0.096 |
| NIAQUE-Global | 23.686 | 0.134 | 0.295 | -0.00 | 94.2 | 0.095 |

Table 98: Performance comparison on VideoGameSales dataset. Lower values are better for all metrics except CI_Coverage (target: 95%). For BIAS, lower absolute values are better.

| Model | SMAPE | AAD | RMSE | BIAS | COV@95 | CRPS |
|---|---|---|---|---|---|---|
| CatBoost-Global | 98.925 | 0.133 | 0.627 | -0.10 | 80.2 | 0.122 |
| CatBoost-Local | 89.060 | 0.121 | 0.615 | -0.09 | 93.1 | 0.105 |
| FTTransformer-Local | 96.390 | 0.173 | 0.680 | -0.10 | 95.2 | 0.108 |
| FTTransformer-Global | 89.584 | 0.121 | 0.581 | -0.09 | 94.3 | 0.086 |
| NIAQUE-Local | 93.144 | 0.128 | 0.625 | -0.09 | 98.5 | 0.097 |
| NIAQUE-Global | 91.507 | 0.125 | 0.618 | -0.09 | 95.3 | 0.096 |

Table 99: Performance comparison on VisualizingSoil dataset. Lower values are better for all metrics except CI_Coverage (target: 95%). For BIAS, lower absolute values are better.

| Model | SMAPE | AAD | RMSE | BIAS | COV@95 | CRPS |
|---|---|---|---|---|---|---|
| CatBoost-Global | 38.068 | 0.684 | 1.060 | 0.24 | 93.2 | 0.620 |
| CatBoost-Local | 14.906 | 0.058 | 0.105 | 0.00 | 95.5 | 0.112 |
| FTTransformer-Local | 15.108 | 0.075 | 0.113 | -0.01 | 97.3 | 0.068 |
| FTTransformer-Global | 11.674 | 0.024 | 0.048 | 0.01 | 98.6 | 0.030 |
| NIAQUE-Local | 10.672 | 0.027 | 0.052 | 0.00 | 99.0 | 0.021 |
| NIAQUE-Global | 9.617 | 0.015 | 0.033 | -0.00 | 97.9 | 0.013 |

Table 100: Performance comparison on WalmartSales dataset. Lower values are better for all metrics except CI_Coverage (target: 95%). For BIAS, lower absolute values are better.

| Model | SMAPE | AAD | RMSE | BIAS | COV@95 | CRPS |
|---|---|---|---|---|---|---|
| CatBoost-Global | 61.504 | 1.320 | 1.575 | 0.38 | 96.1 | 0.982 |
| CatBoost-Local | 14.493 | 0.235 | 0.448 | -0.06 | 94.3 | 0.189 |
| FTTransformer-Local | 16.095 | 0.276 | 0.562 | -0.09 | 92.9 | 0.189 |
| FTTransformer-Global | 13.604 | 0.240 | 0.501 | -0.06 | 94.8 | 0.160 |
| NIAQUE-Local | 13.611 | 0.255 | 0.555 | -0.05 | 90.8 | 0.173 |
| NIAQUE-Global | 12.916 | 0.249 | 0.549 | -0.07 | 92.7 | 0.164 |

Table 101: Performance comparison on WhiteWine dataset. Lower values are better for all metrics except CI_Coverage (target: 95%). For BIAS, lower absolute values are better.

| Model | SMAPE | AAD | RMSE | BIAS | COV@95 | CRPS |
|---|---|---|---|---|---|---|
| CatBoost-Global | 22.573 | 1.064 | 1.494 | -0.70 | 95.5 | 0.892 |
| CatBoost-Local | 19.482 | 0.898 | 1.213 | -0.16 | 94.3 | 0.679 |
| FTTransformer-Local | 20.628 | 0.925 | 1.340 | -0.20 | 98.0 | 0.655 |
| FTTransformer-Global | 19.436 | 0.879 | 1.206 | -0.19 | 94.7 | 0.619 |
| NIAQUE-Local | 19.824 | 0.877 | 1.243 | -0.17 | 92.7 | 0.634 |
| NIAQUE-Global | 19.445 | 0.890 | 1.245 | -0.18 | 92.9 | 0.615 |

Table 102: Performance comparison on Wind dataset. Lower values are better for all metrics except CI_Coverage (target: 95%). For BIAS, lower absolute values are better.

| Model | SMAPE | AAD | RMSE | BIAS | COV@95 | CRPS |
|---|---|---|---|---|---|---|
| CatBoost-Global | 29.371 | 0.964 | 1.259 | -0.11 | 98.8 | 0.801 |
| CatBoost-Local | 18.869 | 0.590 | 0.754 | 0.05 | 93.6 | 0.424 |
| FTTransformer-Local | 21.169 | 0.617 | 0.863 | 0.03 | 91.9 | 0.425 |
| FTTransformer-Global | 18.744 | 0.589 | 0.708 | 0.04 | 92.6 | 0.397 |
| NIAQUE-Local | 18.820 | 0.611 | 0.759 | 0.04 | 94.3 | 0.412 |
| NIAQUE-Global | 19.038 | 0.587 | 0.755 | 0.03 | 93.0 | 0.412 |

Table 103: Performance comparison on Wizmir dataset. Lower values are better for all metrics except CI_Coverage (target: 95%). For BIAS, lower absolute values are better.

| Model | SMAPE | AAD | RMSE | BIAS | COV@95 | CRPS |
|---|---|---|---|---|---|---|
| CatBoost-Global | 27.798 | 1.467 | 1.982 | -1.09 | 91.2 | 1.101 |
| CatBoost-Local | 3.709 | 0.150 | 0.207 | 0.01 | 89.1 | 0.124 |
| FTTransformer-Local | 5.719 | 0.189 | 0.239 | -0.02 | 93.8 | 0.116 |
| FTTransformer-Global | 3.115 | 0.145 | 0.176 | -0.00 | 99.0 | 0.095 |
| NIAQUE-Local | 3.446 | 0.142 | 0.216 | -0.01 | 96.1 | 0.102 |
| NIAQUE-Global | 2.983 | 0.135 | 0.195 | -0.01 | 96.6 | 0.100 |

Table 104: Performance comparison on Yprop41 dataset. Lower values are better for all metrics except CI_Coverage (target: 95%). For BIAS, lower absolute values are better.

| Model | SMAPE | AAD | RMSE | BIAS | COV@95 | CRPS |
|---|---|---|---|---|---|---|
| CatBoost-Global | 2.218 | 0.206 | 0.317 | -0.12 | 91.6 | 0.161 |
| CatBoost-Local | 2.125 | 0.197 | 0.295 | -0.07 | 92.8 | 0.154 |
| FTTransformer-Local | 4.460 | 0.251 | 0.338 | -0.10 | 90.3 | 0.150 |
| FTTransformer-Global | 2.063 | 0.190 | 0.286 | -0.09 | 92.9 | 0.133 |
| NIAQUE-Local | 2.584 | 0.209 | 0.312 | -0.09 | 93.4 | 0.146 |
| NIAQUE-Global | 2.136 | 0.198 | 0.299 | -0.09 | 92.7 | 0.144 |

Table 105: Performance comparison on ZurichDelays dataset. Lower values are better for all metrics except CI_Coverage (target: 95%). For BIAS, lower absolute values are better.

| Model | SMAPE | AAD | RMSE | BIAS | COV@95 | CRPS |
|---|---|---|---|---|---|---|
| CatBoost-Global | 2.343 | 0.132 | 0.254 | -0.08 | 96.2 | 0.108 |
| CatBoost-Local | 2.304 | 0.129 | 0.240 | -0.05 | 94.6 | 0.099 |
| FTTransformer-Local | 4.493 | 0.173 | 0.276 | -0.06 | 93.9 | 0.109 |
| FTTransformer-Global | 2.342 | 0.131 | 0.234 | -0.05 | 94.9 | 0.080 |
| NIAQUE-Local | 2.795 | 0.141 | 0.262 | -0.06 | 97.3 | 0.093 |
| NIAQUE-Global | 2.327 | 0.130 | 0.242 | -0.06 | 95.7 | 0.093 |

Table 106: Performance comparison on house_16H dataset. Lower values are better for all metrics except CI_Coverage (target: 95%). For BIAS, lower absolute values are better.

| Model | SMAPE | AAD | RMSE | BIAS | COV@95 | CRPS |
|---|---|---|---|---|---|---|
| CatBoost-Global | 32.984 | 0.407 | 0.930 | -0.19 | 87.5 | 0.313 |
| CatBoost-Local | 27.666 | 0.316 | 0.682 | -0.12 | 93.2 | 0.246 |
| FTTransformer-Local | 28.408 | 0.331 | 0.704 | -0.07 | 93.7 | 0.230 |
| FTTransformer-Global | 25.411 | 0.293 | 0.655 | -0.05 | 98.3 | 0.203 |
| NIAQUE-Local | 24.935 | 0.300 | 0.620 | -0.05 | 96.3 | 0.209 |
| NIAQUE-Global | 24.701 | 0.283 | 0.612 | -0.06 | 95.4 | 0.203 |

Table 107: Performance comparison on Wind dataset across point prediction accuracy (SMAPE, AAD, RMSE, BIAS) and distributional accuracy (COV@95, CRPS) metrics. Lower values are better for all metrics except COV@95, where values closer to 95 are optimal. The results with 95% confidence intervals derived from multiple random seed runs.

| Model | SMAPE | AAD | RMSE | BIAS | COV@95 | CRPS |
|---|---|---|---|---|---|---|
| XGBoost-Local | $19.4 \pm 0.1$ | $0.607 \pm 0.001$ | $0.779 \pm 0.004$ | $0.044 \pm 0.01$ | $64.9 \pm 0.2$ | $0.473 \pm 0.001$ |
| XGBoost-Domain | $19.3 \pm 0.2$ | $0.603 \pm 0.002$ | $0.779 \pm 0.005$ | $0.016 \pm 0.01$ | $96.0 \pm 0.2$ | $0.462 \pm 0.002$ |
| XGBoost-Global | $20.0 \pm 4.4$ | $0.627 \pm 0.100$ | $0.811 \pm 0.143$ | $0.010 \pm 0.05$ | $97.0 \pm 0.3$ | $0.527 \pm 0.165$ |
| LightGBM-Local | $18.8 \pm 0.2$ | $0.589 \pm 0.01$ | $0.753 \pm 0.007$ | $0.04 \pm 0.01$ | $91.5 \pm 1.0$ | $0.429 \pm 0.003$ |
| LightGBM-Domain | $18.9 \pm 0.1$ | $0.592 \pm 0.004$ | $0.766 \pm 0.006$ | $0.05 \pm 0.01$ | $93.8 \pm 0.6$ | $0.446 \pm 0.002$ |
| LightGBM-Global | $20.8 \pm 0.1$ | $0.657 \pm 0.002$ | $0.854 \pm 0.001$ | $0.02 \pm 0.01$ | $97.8 \pm 0.3$ | $0.599 \pm 0.032$ |
| CatBoost-Local | $18.9 \pm 0.1$ | $0.590 \pm 0.001$ | $0.754 \pm 0.003$ | $0.046 \pm 0.01$ | $93.6 \pm 0.1$ | $0.424 \pm 0.001$ |
| CatBoost-Domain | $19.8 \pm 0.2$ | $0.617 \pm 0.002$ | $0.796 \pm 0.004$ | $0.038 \pm 0.01$ | $94.8 \pm 0.2$ | $0.454 \pm 0.002$ |
| CatBoost-Global | $26.3 \pm 0.2$ | $0.832 \pm 0.006$ | $1.068 \pm 0.009$ | $0.027 \pm 0.02$ | $98.3 \pm 0.2$ | $0.722 \pm 0.004$ |
| Transformer-Local | $18.5 \pm 0.3$ | $0.575 \pm 0.008$ | $0.740 \pm 0.015$ | $0.025 \pm 0.01$ | $94.2 \pm 0.3$ | $0.410 \pm 0.005$ |
| Transformer-Domain | $18.5 \pm 0.3$ | $0.573 \pm 0.008$ | $0.736 \pm 0.015$ | $0.022 \pm 0.01$ | $94.3 \pm 0.3$ | $0.405 \pm 0.005$ |
| Transformer-Global | $18.4 \pm 0.3$ | $0.572 \pm 0.008$ | $0.733 \pm 0.015$ | $0.020 \pm 0.01$ | $94.5 \pm 0.3$ | $0.401 \pm 0.005$ |
| NIAQUE-Local | $18.8 \pm 0.4$ | $0.582 \pm 0.012$ | $0.747 \pm 0.019$ | $0.045 \pm 0.01$ | $95.7 \pm 0.4$ | $0.407 \pm 0.011$ |
| NIAQUE-Domain | $18.7 \pm 0.2$ | $0.585 \pm 0.006$ | $0.760 \pm 0.010$ | $0.027 \pm 0.01$ | $95.4 \pm 0.3$ | $0.415 \pm 0.006$ |
| NIAQUE-Global | $19.0 \pm 0.1$ | $0.587 \pm 0.002$ | $0.755 \pm 0.005$ | $0.031 \pm 0.01$ | $93.0 \pm 0.2$ | $0.412 \pm 0.002$ |

## D.2 Domain Adaptation

For domain adaptation, we first focus on *House Price Prediction*, selecting HouseRent (Banerjee, 2022) (4.7K samples, 11 features) as our representative dataset. This dataset offers a balanced evaluation ground with its moderate sample size and limited feature dimensionality. Next, we focus on *Energy and Efficiency Domain*. We analyze the Wind dataset (OpenML, n.d.) (6.5K samples, 12 features) as our representative case study. Similar to HouseRent, this dataset was chosen for its moderate sample size and limited feature dimensionality, providing a balanced evaluation ground. Tables 107, 108 show the performance comparison between NIAQUE and baselines across different training scenarios. The results reinforce our main findings: NIAQUE maintains consistent performance across local, domain, and global training settings (RMSE: 0.747-0.760), while traditional models like CatBoost show significant degradation in global settings (RMSE increases from 0.754 to 1.068). Notably, NIAQUE's performance in the Energy and Efficiency domain exhibits similar patterns to those observed in the House Price Prediction domain. The model maintains reliable uncertainty quantification (coverage near 95%) and demonstrates effective knowledge transfer in domain-specific training, achieving comparable or better performance than local training, showing the ability of the model to effectively leverage additional information from domain-specific datasets to improve on target task.

Table 108: Performance comparison on HouseRent dataset across point prediction accuracy (SMAPE, AAD, RMSE, BIAS) and distributional accuracy (COV@95, CRPS) metrics. Lower values are better for all metrics except COV@95, where values closer to 95 are optimal. The results with 95% confidence intervals derived from multiple random seed runs.

| Model | SMAPE | AAD | RMSE | BIAS | COV@95 | CRPS |
|---|---|---|---|---|---|---|
| XGBoost-Local | 34.1 | 0.028 | 0.066 | −0.002 | 86.5 | 0.022 |
| XGBoost-Domain | 32.9 | 0.031 | 0.073 | 0.004 | 88.8 | 0.026 |
| XGBoost-Global | 35.4 | 0.033 | 0.075 | 0.011 | 62.5 | 0.033 |
| LightGBM-Local | 31.4 | 0.028 | 0.068 | −0.01 | 92.5 | 0.021 |
| LightGBM-Domain | 33.6 | 0.029 | 0.07 | −0.01 | 91.8 | 0.024 |
| LightGBM-Global | 37.1 | 0.036 | 0.085 | −0.01 | 94.4 | 0.028 |
| CatBoost-Local | 31.5 | 0.028 | 0.068 | −0.007 | 92.8 | 0.022 |
| CatBoost-Domain | 32.4 | 0.028 | 0.072 | −0.011 | 94.1 | 0.022 |
| CatBoost-Global | 50.3 | 0.044 | 0.104 | −0.015 | 88.4 | 0.032 |
| Transformer-Local | 33.5 | 0.030 | 0.070 | −0.008 | 93.6 | 0.025 |
| Transformer-Domain | 32.8 | 0.029 | 0.069 | −0.007 | 94.0 | 0.023 |
| Transformer-Global | 32.3 | 0.028 | 0.068 | −0.007 | 93.5 | 0.020 |
| NIAQUE-Local | 32.0 | 0.028 | 0.067 | −0.007 | 96.0 | 0.019 |
| NIAQUE-Domain | 30.4 | 0.026 | 0.063 | −0.002 | 93.5 | 0.018 |
| NIAQUE-Global | 30.3 | 0.026 | 0.067 | −0.005 | 93.3 | 0.019 |

Table 109: Performance comparison on Kaggle competition dataset. Lower values are better. Baseline results are adopted from publicly shared notebooks and discussion forums in the competition (Reade & Chow, 2024a). Rank represents the position of the various methods on private leaderboard.

| Model | Feature Engineering | RMLSE | Rank |
|---|---|---|---|
| XGBoost (Broccoli Beef (siukeitin), 2024) | None | 0.15019 | 1615 |
| LightGBM (, dataWr3cker) | None | 0.14914 | 1464 |
| CatBoost (Wate, 2024) | None | 0.14783 | 1064 |
| TabNet (Broccoli Beef (siukeitin), 2024) | None | 0.15481 | 2047 |
| TabDPT | None | 0.15026 | 1623 |
| TabDPT | OpenFE | 0.14751 | 924 |
| TabPFN | None | 0.15732 | 2132 |
| TabPFN | OpenFE | 0.14922 | 1478 |
| NIAQUE-Scratch | None | 0.15047 | 1646 |
| NIAQUE-Pretrain-100 | None | 0.14823 | 1232 |
| NIAQUE-Pretrain-full | None | 0.14808 | 1178 |
| NIAQUE-Pretrain-full | OpenFE | 0.14556 | 304 |
| NIAQUE-Ensemble | OpenFE | 0.14423 | 8 |
| Winner (Heller, 2024) | Manual | 0.14374 | 1 |

## D.3 Adaptation to New Tasks: Kaggle Competitions

### D.3.1 Abalone Competition

To validate NIAQUE's practical effectiveness, we evaluate its performance in recent Kaggle competitions. Abalone (Reade & Chow, 2024a), focuses on the Abalone age prediction task. This competition, with 2,700 participants and over 20,000 submissions, provides an excellent real-world benchmark, particularly as neural networks are widely reported to underperform in it, compared to traditional boosted tree methods.

**Methodology:** Our approach involves two stages: pretraining and fine-tuning. First, we pretrain NIAQUE on TabRegSet-101 (which includes the original UCI Abalone dataset (Dua & Graff, 2019)) using our quantile loss framework. Then, we fine-tune the pretrained model on the competition's training data, optimizing for RMSLE metric as defined in Section 3. The dataset size is $90,614$ training samples and $60,410$ test samples (Reade & Chow, 2024a). To systematically evaluate the impact of transfer learning, we implement three variants: NIAQUE-Scratch (trained only on competition data, no pretraining), NIAQUE-Pretrain-100 (pretrained on TabRegSet-101, excluding Abalone dataset), and NIAQUE-Pretrain-full (pretrained on full TabRegSet-101). Note that the competition dataset is synthetic, generated using a deep learning model trained on the original Abalone dataset, and the competition explicitly encourages the use of the original dataset. Finally, we measure the effect of automated feature engineering on our model using OpenFE (Zhang et al., 2023), an automated feature engineering framework, and explore ensemble strategies leveraging our model's probabilistic nature (NIAQUE-Ensemble).

**Results and Analysis:** The private leaderboard competition results along with their ranks are presented in Table 109, highlighting NIAQUE's capabilities against a wide range of well-established representative approaches. The effectiveness of transfer learning is evident in the performance progression: NIAQUE-Scratch (RMSLE: 0.15047), trained only on competition data, performs similarly to traditional baselines like XGBoost (0.15019). NIAQUE-Pretrain-100 (0.14823), pretrained on TabRegSet-101 excluding the Abalone dataset, shows significant improvement, demonstrating that knowledge from unrelated regression tasks can enhance performance. NIAQUE-Pretrain-full (0.14808), leveraging the complete TabRegSet-101, further improves performance by incorporating original Abalone information in the training mix and matching Cat-Boost (0.14783) without any feature engineering. The addition of automated feature engineering (OpenFE) into the competition dataset further improves NIAQUE's performance to 0.14556, significantly outperforming all baseline models and approaching the competition's winning score. Our best result comes from NIAQUE-Ensemble (RMSLE: 0.14423), which leverages the probabilistic nature of our model by combining predictions from different quantiles and model variants (Scratch and Pretrain-full). The ensemble benefits from quantile predictions (in addition to the median) that help correct for data imbalance, predicting higher values where models might underestimate and vice versa, achieving the 8th position on the private leaderboard, remarkably close to the winning score of 0.14374. This result is attained without extensive manual intervention—eschewing hand-engineered features in favor of transfer learning, standard ensemble techniques and automated feature extraction. Moreover, the model's probabilistic formulation is leveraged to further enhance point prediction accuracy, consistent with Bayesian estimation theory, where optimal estimators are typically derived as functions of the posterior distribution (e.g., the variance-optimal estimator is the posterior mean).

These results are particularly significant given that neural networks were generally considered ineffective for this task. Multiple participants reported neural approaches failing to achieve scores better than 0.15, with the competition's winner noting: *"I tried different neural network architectures only to observe that none of them is competitive"* (Heller, 2024), echoed by other participants: *"yes, I also tried different neural network architectures only to observe that they could not reach beyond .15xx"* (Heller, 2024). This real-world validation highlights NIAQUE's competitiveness against heavily engineered solutions and underscores the efficacy of integrating transfer learning with probabilistic modeling. Whereas the winning solution relied on an ensemble of 49 models with extensive manual tuning, our approach achieves comparable performance through principled transfer learning and uncertainty quantification, maintaining interpretability and necessitating minimal task-specific modifications.

### D.3.2 FloodPrediction Competition

The flood prediction competition (Reade & Chow, 2024b) presents a challenging regression task aimed at predicting flood event probabilities based on environmental features including MonsoonIntensity, TopographyDrainage, RiverManagement, and Deforestation. With 2,932 participants, this competition addresses a critical real-world problem where labeled data consists of flood probabilities rather than binary outcomes, making regression models more suitable than classification approaches.

**Methodology:** We adapted NIAQUE for this flood probability prediction task while maintaining its core probabilistic framework. The competition dataset comprises 1.12M training samples and 745K test samples,

Table 110: Performance comparison on Flood Prediction competition. Higher R2-scores are better. Baseline results are from competition forums (Reade & Chow, 2024b). Rank represents the position on private leaderboard.

| Model | R2 Score | Rank |
| --- | --- | --- |
| XGBoost (Broccoli Beef (siukeitin), 2024; Sayed, 2024) | 0.842 | 2304 |
| LightGBM (, dataWr3cker; Masoudi, 2024) | 0.766 | 2557 |
| CatBoost (Wate, 2024; Milind, 2024) | 0.845 | 1700 |
| TabNet (Broccoli Beef (siukeitin), 2024) | 0.842 | 2304 |
| TabDPT () | 0.804 | 2529 |
| TabPFN () | 0.431 | - |
| NIAQUE-Scratch | 0.865 | 1099 |
| NIAQUE-Pretrain | 0.867 | 935 |
| Winner (Heller, 2024; Aldparis, 2024) | 0.869 | 1 |

each containing relevant features derived from Flood Prediction Factors (Dhankour, 2024). Similar to our Abalone experiment, we evaluate transfer learning effectiveness through two variants: NIAQUE-Scratch (trained directly on competition data) and NIAQUE-Pretrain (pretrained on TabRegSet-101). The model is pretrained using our quantile loss framework and then fine-tuned with MSE Loss on the competition dataset. Notably, we found that automated feature engineering not only failed to improve performance but also significantly increased the feature space dimensionality, making several baseline methods (TabDPT and TabPFN) computationally intractable. Therefore, we focus on comparing the fundamental algorithmic capabilities without feature engineering enhancements.

**Results and Analysis:** The competition results (Table 110) demonstrate NIAQUE's strong performance in flood prediction. NIAQUE-Scratch achieves an R2-score of 0.865, significantly outperforming both traditional methods like XGBoost (0.842), CatBoost (0.845), and LightGBM (0.766), as well as newer deep learning approaches such as TabNet (0.842), TabDPT (0.804), and TabPFN (0.431). NIAQUE-Pretrain further improves performance to 0.867, approaching the winning score of 0.869, and securing rank 935 on the private leaderboard.

These results are particularly noteworthy given the scale and complexity of the dataset. Unlike the Abalone competition, where feature engineering played a crucial role, this competition highlights NIAQUE's ability to learn effective representations directly from raw features. The improvement from pretraining, though modest in absolute terms (0.002 increase in R2-score), represents a significant advancement in ranking (from 1099 to 935), demonstrating the value of transfer learning even in specialized environmental prediction tasks. The strong performance of our base model without extensive modifications or feature engineering suggests that NIAQUE's probabilistic framework and architecture are well-suited for large-scale regression tasks where the target variable represents underlying probabilities.

The relatively poor performance of some specialized tabular models (particularly TabPFN with R2-score of 0.431) on this large-scale dataset underscores the importance of scalability in real-world applications. NIAQUE maintains its computational efficiency while handling over a million samples, making it practical for deployment in real-world flood prediction systems where both accuracy and computational resources are critical considerations.

### D.4 Statistical Significance Analysis

To save space, we present benchmarking results with confidence intervals here. All confidence intervals are obtained by aggregating the evaluation results over 4 runs with different random seeds.

Table 111: Performance comparison across all metrics, with point prediction accuracy (SMAPE, AAD, RMSE, BIAS) and distributional accuracy (COV@95, CRPS). Lower values are better for all metrics except COV@95, where values closer to 95 are optimal. The results with 95% confidence intervals derived from 4 random seed runs.

| | SMAPE | AAD | RMSE | BIAS | COV@95 | CRPS |
|---|---|---|---|---|---|---|
| XGBoost-global | $31.4 \pm 4.4$ | $0.574 \pm 0.100$ | $1.056 \pm 0.143$ | $-0.15 \pm 0.05$ | $94.6 \pm 0.3$ | $0.636 \pm 0.165$ |
| XGBoost-local | $25.6 \pm 0.1$ | $0.433 \pm 0.001$ | $0.883 \pm 0.004$ | $-0.03 \pm 0.01$ | $90.8 \pm 0.2$ | $0.334 \pm 0.001$ |
| LightGBM-global | $27.5 \pm 0.1$ | $0.475 \pm 0.001$ | $0.930 \pm 0.003$ | $-0.06 \pm 0.01$ | $94.8 \pm 0.1$ | $0.426 \pm 0.017$ |
| LightGBM-local | $25.7 \pm 0.1$ | $0.427 \pm 0.003$ | $0.865 \pm 0.012$ | $-0.03 \pm 0.01$ | $91.5 \pm 0.2$ | $0.327 \pm 0.001$ |
| CATBOOST-global | $31.3 \pm 0.2$ | $0.561 \pm 0.006$ | $1.030 \pm 0.009$ | $-0.12 \pm 0.02$ | $94.9 \pm 0.2$ | $0.443 \pm 0.004$ |
| CATBOOST-local | $24.3 \pm 0.1$ | $0.408 \pm 0.001$ | $0.840 \pm 0.003$ | $-0.03 \pm 0.01$ | $92.7 \pm 0.1$ | $0.315 \pm 0.001$ |
| Transformer-global | $23.1 \pm 0.3$ | $0.383 \pm 0.008$ | $0.806 \pm 0.015$ | $-0.01 \pm 0.01$ | $94.6 \pm 0.3$ | $0.272 \pm 0.005$ |
| NIAQUE-local | $22.8 \pm 0.4$ | $0.377 \pm 0.012$ | $0.797 \pm 0.019$ | $-0.03 \pm 0.01$ | $94.9 \pm 0.4$ | $0.267 \pm 0.011$ |
| NIAQUE-global | $22.1 \pm 0.1$ | $0.367 \pm 0.002$ | $0.787 \pm 0.005$ | $-0.02 \pm 0.01$ | $94.6 \pm 0.2$ | $0.261 \pm 0.002$ |

Table 112: Ablation study of the XGBoost model.

| type | max depth | learning rate | SMAPE | AAD | BIAS | RMSE | CRPS | COVERAGE @ 95 |
|---|---|---|---|---|---|---|---|---|
| global | 8 | 0.02 | 31.4 | 0.574 | -0.15 | 1.056 | 0.636 | 94.6 |
| global | 16 | 0.02 | 25.7 | 0.441 | -0.07 | 0.864 | 0.484 | 91.5 |
| global | 32 | 0.02 | 24.1 | 0.402 | -0.05 | 0.800 | 0.353 | 80.0 |
| global | 40 | 0.02 | 24.6 | 0.414 | -0.05 | 0.815 | 0.378 | 78.2 |
| global | 48 | 0.02 | 24.1 | 0.397 | -0.04 | 0.785 | 0.362 | 74.8 |
| global | 96 | 0.02 | 23.8 | 0.384 | -0.03 | 0.769 | 0.346 | 64.9 |
| local | 16 | 0.02 | 23.0 | 0.367 | -0.00 | 0.753 | 0.317 | 52.0 |
| local | 12 | 0.02 | 22.7 | 0.369 | -0.01 | 0.756 | 0.304 | 66.0 |
| local | 8 | 0.02 | 22.4 | 0.372 | -0.02 | 0.773 | 0.294 | 82.3 |
| local | 8 | 0.05 | 22.5 | 0.373 | -0.02 | 0.773 | 0.291 | 82.4 |
| local | 6 | 0.02 | 22.7 | 0.382 | -0.02 | 0.795 | 0.298 | 87.3 |
| local | 4 | 0.02 | 24.1 | 0.412 | -0.03 | 0.847 | 0.318 | 90.2 |
| local | 3 | 0.02 | 25.6 | 0.433 | -0.03 | 0.883 | 0.334 | 90.8 |

# E    XGBoost Baseline

Table 113: Ablation study of the CATBoost model.

| type | depth | min data in leaf | SMAPE | AAD | BIAS | RMSE | CRPS | COVERAGE @ 95 |
|---|---|---|---|---|---|---|---|---|
| global | 16 | 50 | 31.4 | 0.565 | -0.12 | 1.036 | 0.442 | 94.2 |
| global | 16 | 100 | 31.3 | 0.561 | -0.12 | 1.030 | 0.443 | 94.9 |
| global | 16 | 200 | 31.6 | 0.569 | -0.13 | 1.041 | 0.445 | 94.2 |
| global | 8 | 100 | 41.1 | 0.785 | -0.26 | 1.324 | 0.602 | 94.3 |
| local | 3 | 50 | 24.3 | 0.409 | -0.03 | 0.841 | 0.316 | 92.7 |
| local | 3 | 100 | 24.3 | 0.407 | -0.03 | 0.843 | 0.317 | 92.7 |
| local | 3 | 200 | 24.3 | 0.408 | -0.03 | 0.840 | 0.315 | 92.7 |
| local | 5 | 50 | 22.2 | 0.373 | -0.02 | 0.785 | 0.285 | 90.7 |
| local | 5 | 100 | 22.3 | 0.374 | -0.02 | 0.786 | 0.285 | 91.3 |
| local | 5 | 200 | 22.4 | 0.378 | -0.02 | 0.791 | 0.288 | 91.6 |
| local | 7 | 50 | 21.5 | 0.359 | -0.02 | 0.761 | 0.272 | 87.2 |
| local | 7 | 100 | 21.6 | 0.362 | -0.02 | 0.765 | 0.273 | 88.6 |
| local | 7 | 200 | 21.8 | 0.366 | -0.02 | 0.772 | 0.277 | 89.9 |

Table 114: CATBoost accuracy as a function of the number of quantiles.

| type | depth | min data in leaf | num quantiles | SMAPE | AAD | BIAS | RMSE | CRPS | COVERAGE @ 95 |
|---|---|---|---|---|---|---|---|---|---|
| global | 16 | 100 | 3 | 31.3 | 0.561 | -0.12 | 1.030 | 0.443 | 94.9 |
| global | 16 | 100 | 5 | 35.0 | 0.665 | -0.13 | 1.183 | 0.482 | 96.2 |
| global | 16 | 100 | 7 | 38.5 | 0.746 | -0.18 | 1.265 | 0.533 | 96.2 |
| global | 16 | 100 | 9 | 43.7 | 0.879 | -0.25 | 1.437 | 0.622 | 96.2 |
| global | 16 | 100 | 51 | 68.9 | 1.538 | -0.53 | 2.132 | 1.036 | 95.5 |
| local | 7 | 100 | 3 | 21.5 | 0.359 | -0.02 | 0.761 | 0.272 | 87.2 |
| local | 7 | 100 | 9 | 23.9 | 0.399 | -0.03 | 0.823 | 0.284 | 92.4 |
| local | 7 | 100 | 51 | 30.3 | 0.525 | -0.09 | 1.079 | 0.369 | 92.1 |
| local | 16 | 100 | 51 | 30.2 | 0.514 | -0.09 | 1.055 | 0.362 | 92.4 |

# F  CATBoost Baseline

The CATBoost is trained using the standard package via `pip install catboost` using `grow_policy = Depthwise`. The explored hyper-parqameter grid appears in Table 113.

Table 114 shows CATBoost accuracy as a function of the number of quantiles. Quantiles are generated using linspace grid `np.linspace(0.01, 0.99, num_quantiles)`. We recover the best overall result for the case of 3 quantiles, and increasing the number of quantiles leads to quickly deteriorating metrics. It appears that CATBoost is unfit to solve complex multi-quantile problems.

Table 115: Ablation study of the LightGBM global model.

| type | max_depth | num leaves | learning rate | SMAPE | AAD | BIAS | RMSE | CRPS | COVERAGE @ 95 |
|---|---|---|---|---|---|---|---|---|---|
| global | -1 | 10 | 0.05 | 35.6 | 0.661 | -0.17 | 1.199 | 0.804 | 95.2 |
| global | -1 | 20 | 0.05 | 30.9 | 0.554 | -0.11 | 1.034 | 0.566 | 95.4 |
| global | -1 | 40 | 0.05 | 27.5 | 0.475 | -0.06 | 0.930 | 0.426 | 94.8 |
| global | -1 | 100 | 0.05 | 24.6 | 0.417 | -0.03 | 0.852 | 0.342 | 93.3 |
| global | -1 | 200 | 0.05 | 23.4 | 0.393 | -0.02 | 0.813 | 0.32 | 92.3 |
| global | -1 | 400 | 0.05 | 23.6 | 0.379 | -0.02 | 0.786 | 0.305 | 90.9 |
| global | 3 | 10 | 0.05 | 50.7 | 1.084 | -0.49 | 1.763 | 1.013 | 94.1 |
| global | 3 | 20 | 0.05 | 50.7 | 1.084 | -0.49 | 1.763 | 1.013 | 94.1 |
| global | 3 | 40 | 0.05 | 50.7 | 1.084 | -0.49 | 1.763 | 1.013 | 94.1 |
| global | 3 | 100 | 0.05 | 50.7 | 1.084 | -0.49 | 1.763 | 1.013 | 94.1 |
| global | 3 | 200 | 0.05 | 50.7 | 1.084 | -0.49 | 1.763 | 1.013 | 94.1 |
| global | 3 | 400 | 0.05 | 50.7 | 1.084 | -0.49 | 1.763 | 1.013 | 94.1 |
| global | 5 | 10 | 0.05 | 39.1 | 0.768 | -0.25 | 1.341 | 0.856 | 94.8 |
| global | 5 | 20 | 0.05 | 39.0 | 0.76 | -0.26 | 1.327 | 0.863 | 94.8 |
| global | 5 | 40 | 0.05 | 39.0 | 0.759 | -0.26 | 1.328 | 0.864 | 94.8 |
| global | 5 | 100 | 0.05 | 39.0 | 0.759 | -0.26 | 1.328 | 0.864 | 94.8 |
| global | 5 | 200 | 0.05 | 39.0 | 0.759 | -0.26 | 1.328 | 0.864 | 94.8 |
| global | 5 | 400 | 0.05 | 39.0 | 0.759 | -0.26 | 1.328 | 0.864 | 94.8 |
| global | 10 | 10 | 0.05 | 35.6 | 0.661 | -0.17 | 1.199 | 0.804 | 95.2 |
| global | 10 | 20 | 0.05 | 31.5 | 0.572 | -0.14 | 1.054 | 0.59 | 95.4 |
| global | 10 | 40 | 0.05 | 29.8 | 0.537 | -0.13 | 1.001 | 0.575 | 95.2 |
| global | 10 | 100 | 0.05 | 29.5 | 0.528 | -0.12 | 0.991 | 0.577 | 95.2 |
| global | 10 | 200 | 0.05 | 29.2 | 0.522 | -0.12 | 0.981 | 0.576 | 95.0 |
| global | 10 | 400 | 0.05 | 29.1 | 0.52 | -0.12 | 0.975 | 0.582 | 95.1 |
| global | 20 | 10 | 0.05 | 35.6 | 0.661 | -0.17 | 1.199 | 0.804 | 95.2 |
| global | 20 | 20 | 0.05 | 30.9 | 0.554 | -0.11 | 1.034 | 0.566 | 95.4 |
| global | 20 | 40 | 0.05 | 27.1 | 0.468 | -0.07 | 0.913 | 0.512 | 95.2 |
| global | 20 | 100 | 0.05 | 25.5 | 0.435 | -0.06 | 0.864 | 0.496 | 94.9 |
| global | 20 | 200 | 0.05 | 25.0 | 0.424 | -0.06 | 0.846 | 0.488 | 94.3 |
| global | 20 | 400 | 0.05 | 24.3 | 0.41 | -0.05 | 0.823 | 0.482 | 93.6 |
| global | 40 | 10 | 0.05 | 35.6 | 0.661 | -0.17 | 1.199 | 0.804 | 95.2 |
| global | 40 | 20 | 0.05 | 30.9 | 0.554 | -0.11 | 1.034 | 0.566 | 95.4 |
| global | 40 | 40 | 0.05 | 27.8 | 0.481 | -0.05 | 0.913 | 0.431 | 94.7 |
| global | 40 | 100 | 0.05 | 24.7 | 0.419 | -0.04 | 0.848 | 0.348 | 93.5 |
| global | 40 | 200 | 0.05 | 23.5 | 0.395 | -0.03 | 0.811 | 0.332 | 92.7 |
| global | 40 | 400 | 0.05 | 23.2 | 0.383 | -0.03 | 0.791 | 0.322 | 92.0 |

## G   LightGBM Baseline

Table 116: Ablation study of the LightGBM local model.

| type | max_depth | num leaves | learning rate | SMAPE | AAD | BIAS | RMSE | CRPS | COVERAGE @ 95 |
|------|-----------|-----------|---------------|-------|-----|------|------|------|---------------|
| local | -1 | 5 | 0.05 | 23.8 | 0.399 | -0.03 | 0.823 | 0.319 | 90.6 |
| local | -1 | 10 | 0.05 | 22.5 | 0.376 | -0.02 | 0.786 | 0.301 | 88.9 |
| local | -1 | 20 | 0.05 | 21.9 | 0.364 | -0.02 | 0.766 | 0.289 | 86.5 |
| local | -1 | 50 | 0.05 | 21.6 | 0.355 | -0.01 | 0.752 | 0.278 | 82.6 |
| local | 2 | 5 | 0.05 | 25.7 | 0.427 | -0.03 | 0.865 | 0.327 | 91.5 |
| local | 2 | 10 | 0.05 | 25.7 | 0.427 | -0.03 | 0.865 | 0.327 | 91.5 |
| local | 2 | 20 | 0.05 | 25.7 | 0.427 | -0.03 | 0.865 | 0.327 | 91.5 |
| local | 2 | 50 | 0.05 | 25.7 | 0.427 | -0.03 | 0.865 | 0.327 | 91.5 |
| local | 3 | 5 | 0.05 | 24.3 | 0.404 | -0.03 | 0.83 | 0.318 | 90.7 |
| local | 3 | 10 | 0.05 | 23.9 | 0.396 | -0.03 | 0.818 | 0.304 | 90.4 |
| local | 3 | 20 | 0.05 | 23.9 | 0.396 | -0.03 | 0.818 | 0.304 | 90.4 |
| local | 3 | 50 | 0.05 | 23.9 | 0.396 | -0.03 | 0.818 | 0.304 | 90.4 |
| local | 5 | 5 | 0.05 | 23.8 | 0.399 | -0.03 | 0.823 | 0.319 | 90.6 |
| local | 5 | 10 | 0.05 | 22.7 | 0.379 | -0.02 | 0.79 | 0.3 | 89.1 |
| local | 5 | 20 | 0.05 | 22.3 | 0.37 | -0.02 | 0.776 | 0.287 | 87.6 |
| local | 5 | 50 | 0.05 | 22.2 | 0.368 | -0.02 | 0.773 | 0.285 | 87.4 |

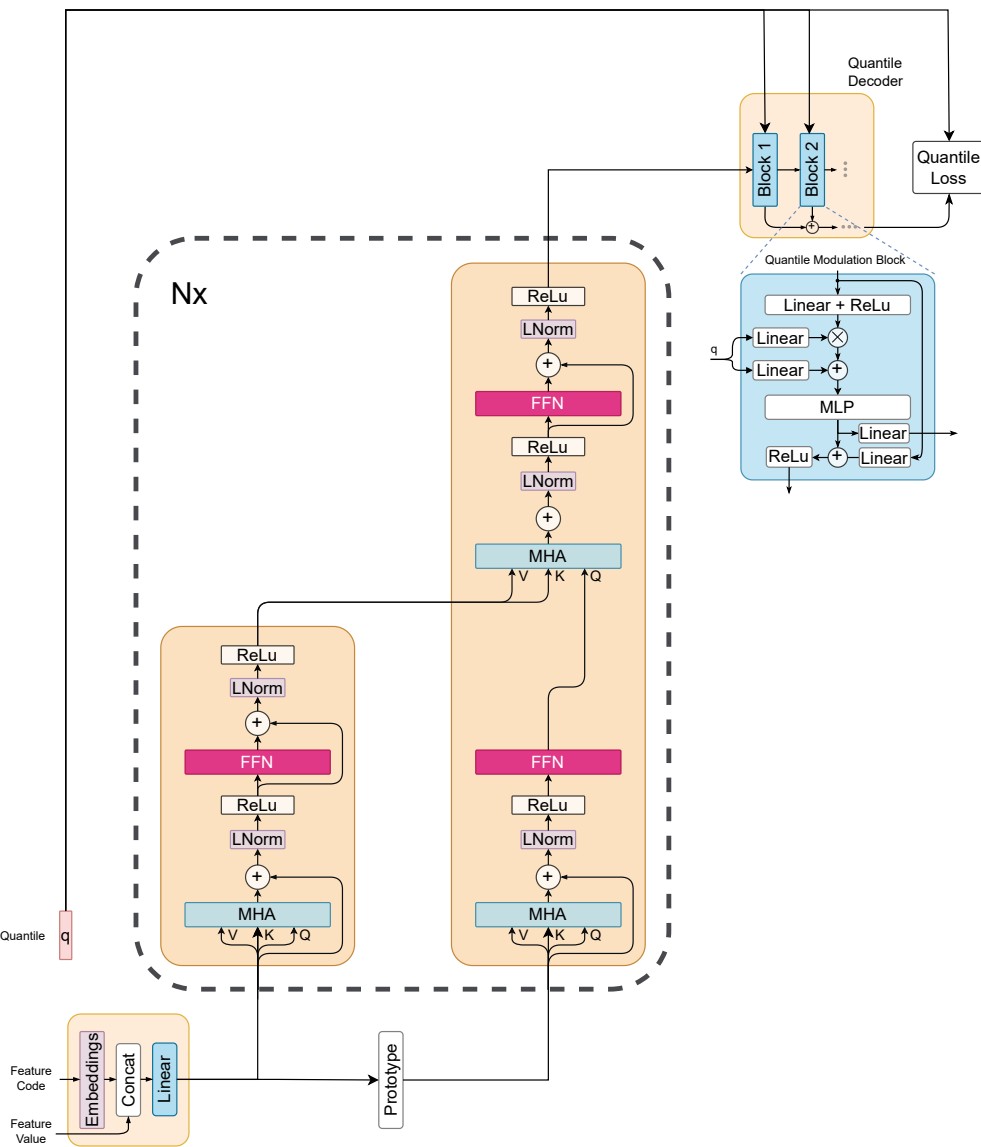

Figure 5: Transformer baseline used in our experiments. The feature encoding module is replaced with transformer block. Feature encoding is implemented via self-attention. The extraction of feature encoding is done by applying cross-attention between the prototype of input features and the output of self-attention. This operation is repeated several times corresponding to the number of blocks in transformer encoder.

## H    Transformer Baseline

The ablation study of the transformer architecture is presented in Table 117. It shows that in general, increasing the number of transformer blocks improves accuracy, however, at 8-10 blocks we clearly see diminishing returns. Dropout helps to gain better empirical coverage of the 95% confidence interval, but this happens at the expense of point prediction accuracy. Finally, the decoder query that is used to produce the feature embedding that is fed to the quantile decoder can be implemented in two principled ways. First, the scheme depicted in Figure 5, uses the prototype of features supplied to the encoder. We call it the prototype scheme. Second, the prototype can be replaced by a learnable embedding. Comparing the last and third rows in Table 117, we conclude that the prototype scheme is a clear winner.

Table 117: Ablation study of the Transformer architecture.

| query | d_model | width | blocks | dp | sMAPE | AAD | BIAS | RMSE | CRPS | COVERAGE @ 95 |
|---|---|---|---|---|---|---|---|---|---|---|
| proto | 256 | 256 | 4 | 0.1 | 25.6 | 0.462 | -0.01 | 0.918 | 0.313 | 95.2 |
| proto | 256 | 1024 | 4 | 0.1 | 24.5 | 0.414 | -0.02 | 0.845 | 0.292 | 95.1 |
| proto | 256 | 256 | 6 | 0.1 | 23.7 | 0.397 | -0.01 | 0.824 | 0.281 | 94.9 |
| proto | 256 | 512 | 6 | 0.2 | | | | | | |
| proto | 256 | 1024 | 6 | 0.1 | 24.3 | 0.407 | -0.01 | 0.840 | 0.287 | 94.9 |
| proto | 256 | 1024 | 6 | 0.0 | 26.5 | 0.477 | -0.04 | 0.980 | 0.334 | 93.0 |
| proto | 256 | 512 | 8 | 0.0 | 23.3 | 0.388 | -0.03 | 0.814 | 0.276 | 94.3 |
| proto | 256 | 1024 | 8 | 0.0 | 23.1 | 0.383 | -0.02 | 0.806 | 0.272 | 94.6 |
| proto | 256 | 1024 | 8 | 0.1 | 23.1 | 0.384 | -0.01 | 0.809 | 0.272 | 94.6 |
| proto | 256 | 512 | 10 | 0.0 | 23.0 | 0.384 | -0.03 | 0.814 | 0.273 | 94.2 |
| proto | 256 | 1024 | 10 | 0.1 | 24.3 | 0.407 | -0.01 | 0.840 | 0.287 | 94.9 |
| proto | 512 | 1024 | 6 | 0.1 | | | | | | |
| learn | 256 | 256 | 6 | 0.2 | 35.0 | 0.722 | -0.16 | 1.406 | 0.489 | 93.9 |

Table 118: Ablation study of NIAQUE-local model.

| blocks | width | dp | layers | sMAPE | AAD | BIAS | RMSE | CRPS | COVERAGE @ 95 |
|---|---|---|---|---|---|---|---|---|---|
| 2 | 64 | 0.0 | 3 | 24.2 | 0.414 | -0.03 | 0.848 | 0.292 | 95.1 |
| 2 | 128 | 0.0 | 3 | 22.8 | 0.381 | -0.02 | 0.804 | 0.270 | 94.5 |
| 2 | 256 | 0.0 | 3 | 22.1 | 0.365 | -0.02 | 0.786 | 0.260 | 94.0 |
| 2 | 512 | 0.0 | 3 | 21.9 | 0.360 | -0.02 | 0.781 | 0.257 | 92.7 |
| 2 | 64 | 0.1 | 3 | 24.7 | 0.431 | -0.07 | 0.855 | 0.305 | 93.3 |
| 2 | 128 | 0.1 | 3 | 23.1 | 0.389 | -0.04 | 0.81 | 0.276 | 94.0 |
| 2 | 256 | 0.1 | 3 | 22.2 | 0.369 | -0.02 | 0.79 | 0.263 | 94.0 |
| 2 | 512 | 0.1 | 3 | 22.0 | 0.361 | -0.02 | 0.779 | 0.257 | 93.5 |
| 2 | 64 | 0.0 | 2 | 24.5 | 0.419 | -0.03 | 0.852 | 0.296 | 95.0 |
| 2 | 128 | 0.0 | 2 | 23.4 | 0.391 | -0.02 | 0.815 | 0.276 | 94.7 |
| 2 | 256 | 0.0 | 2 | 22.3 | 0.368 | -0.02 | 0.783 | 0.262 | 94.1 |
| 2 | 512 | 0.0 | 2 | 22.1 | 0.363 | -0.03 | 0.780 | 0.259 | 92.9 |
| 4 | 64 | 0.0 | 2 | 23.8 | 0.399 | -0.02 | 0.828 | 0.282 | 95.1 |
| 4 | 128 | 0.0 | 2 | 22.8 | 0.377 | -0.03 | 0.797 | 0.267 | 94.9 |
| 4 | 256 | 0.0 | 2 | 22.0 | 0.363 | -0.02 | 0.788 | 0.259 | 93.5 |
| 4 | 512 | 0.0 | 2 | 22.0 | 0.359 | -0.02 | 0.785 | 0.257 | 92.0 |
| 4 | 64 | 0.1 | 2 | 23.8 | 0.401 | -0.03 | 0.829 | 0.284 | 94.3 |
| 4 | 128 | 0.1 | 2 | 22.9 | 0.379 | -0.03 | 0.801 | 0.267 | 94.6 |
| 4 | 256 | 0.1 | 2 | 22.1 | 0.363 | -0.03 | 0.786 | 0.259 | 93.5 |
| 4 | 512 | 0.1 | 2 | 22.0 | 0.360 | -0.03 | 0.781 | 0.257 | 92.4 |
| 8 | 128 | 0.0 | 2 | 23.0 | 0.381 | -0.02 | 0.798 | 0.27 | 95.7 |

## I  NIAQUE-Local Baseline

NIAQUE-local baseline is trained on each dataset individually using the same overall training framework as discussed in the main manuscript for the NIAQUE-global, with the following exceptions. The number of training epochs for each dataset is fixed at 1200, the batch size is set to 256, feature dropout is disabled. Finally, for each dataset we select the best model to be evaluated by monitoring the loss on validation set every epoch.

Table 119: Ablation study of NIAQUE model.

| blocks | width | dp | layers | singles | log input | sMAPE | AAD | BIAS | RMSE | CRPS | COVERAGE @ 95 |
|---|---|---|---|---|---|---|---|---|---|---|---|
| 1 | 1024 | 0.2 | 2 | 5% | yes | 25.6 | 0.433 | -0.04 | 0.864 | 0.306 | 96.5 |
| 2 | 1024 | 0.2 | 2 | 5% | yes | 23.1 | 0.384 | -0.02 | 0.802 | 0.272 | 95.7 |
| 2 | 1024 | 0.2 | 3 | 5% | yes | 22.7 | 0.377 | -0.03 | 0.796 | 0.267 | 95.6 |
| 4 | 1024 | 0.2 | 2 | 5% | yes | 22.1 | 0.367 | -0.02 | 0.787 | 0.261 | 94.6 |
| 4 | 1024 | 0.2 | 3 | 5% | yes | 22.1 | 0.367 | -0.02 | 0.792 | 0.262 | 94.6 |
| 8 | 1024 | 0.2 | 2 | 5% | yes | 22.0 | 0.366 | -0.02 | 0.798 | 0.264 | 92.7 |
| 4 | 512 | 0.2 | 2 | 0% | yes | 22.5 | 0.372 | -0.02 | 0.791 | 0.264 | 95.4 |
| 4 | 1024 | 0.2 | 2 | 0% | yes | 22.1 | 0.366 | -0.02 | 0.791 | 0.261 | 94.2 |
| 4 | 1024 | 0.3 | 2 | 0% | yes | 22.1 | 0.367 | -0.02 | 0.787 | 0.260 | 94.7 |
| 4 | 1024 | 0.4 | 2 | 0% | yes | 22.2 | 0.370 | -0.02 | 0.791 | 0.263 | 95.1 |
| 4 | 2048 | 0.3 | 2 | 0% | yes | 22.1 | 0.366 | -0.02 | 0.795 | 0.263 | 93.4 |
| 4 | 1024 | 0.2 | 2 | 5% | no | 31.4 | 0.530 | -0.066 | 1.017 | 0.371 | 95.6 |

## J  NIAQUE Training Details and Ablation Studies

To train both NIAQUE and Transformer models we use feature dropout defined as follows. Given dropout probability dp, we toss a coin with probability $\sqrt{dp}$ to determine if the dropout event is going to happen at all for a given batch. If this happens, we remove each feature from the batch, again with probability $\sqrt{dp}$. This way each feature has probability dp of being removed from a given batch and there is a probability $\sqrt{dp}$ that the model will see all features intact in a given batch. The intuition behind this design is that we want to expose the model to all features most of the time, but we also want to create many situations with some feature combinations missing.

**Input log transformation** defined in eq. (10) is important to ensure the success of the training, as follows both from Table 119 and Figure 6. The introduction of log-transform makes learning curves well-behaved and smooth and translates into much better accuracy.

**Adding samples containing only one of the features** as input does not significantly affect accuracy. At the same time, the addition of single-feature training rows has very strong effect on the effectiveness of NIAQUE's interpretability mechanism. When rows with single feature input are added (Figures 7a and 7b), NIAQUE demonstrates very clear accuracy degradation when top features are removed and insignificant degradation when bottom features are removed. When rows with single feature input are *not* added (Figure 7c), the discrimination between strong and weak features is poor, with removal of top and bottom features having approximately the same effect across datasets.

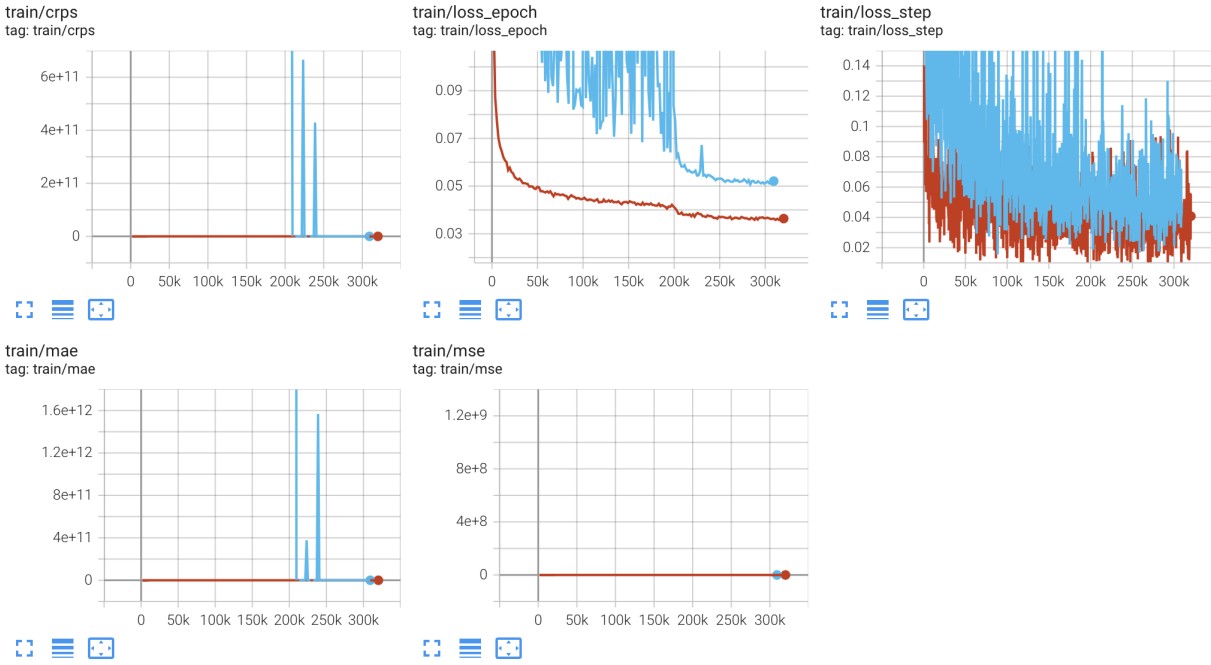

Figure 6: Training losses with (dark red) and without (blue) input value log-transform eq. (10). The introduction of log-transform makes learning curves well-behaved and smooth.

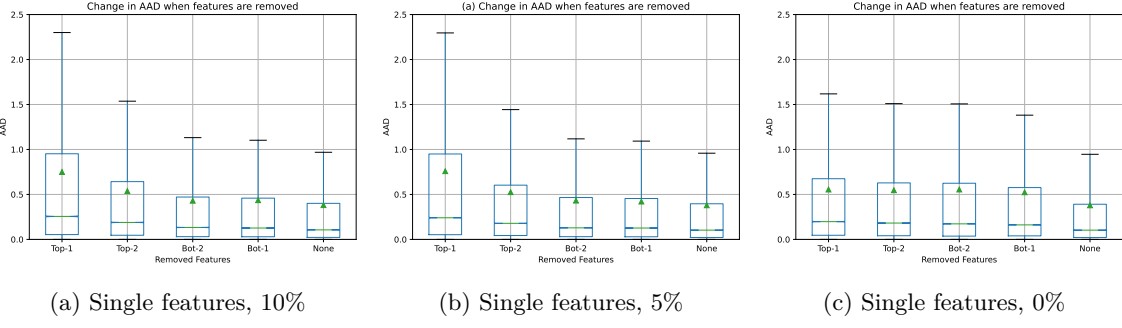

(a) Single features, 10%     (b) Single features, 5%     (c) Single features, 0%

Figure 7: The effect of adding training rows containing only one of the input features as NIAQUE input. When rows with single feature input are added (Figures 7a and 7b), NIAQUE demonstrates very clear accuracy degradation when top features are removed and insignificant degradation when bottom features are removed. When rows with single feature input are *not* added (Figure 7c), the discrimination between strong and weak features is poor, with removal of top and bottom features having approximately the same effect across datasets.

