# OpenReview forum: "Probabilistic Pretraining for Improved Neural Regression"
_TMLR — Accepted by TMLR_

### Review · Reviewer_sz4U · 2025-10-31

**Summary Of Contributions:**

The paper presents NIAQUE, a neural probabilistic regression model for tabular data. The proposed learning objective, architecture, and transfer learning via pre-training are sufficiently supported to be a desired modification through strong empirical results, especially when compared to the baselines. As outlined by the authors, the main novelty of the paper lies in bridging several ideas in different fields that were shown to be desirable in different fields.

**Audience:**

Yes

**Audience Explanation:**

Despite the extensive experiments across diverse datasets and the consistently strong empirical results compared to conventional baselines, the paper’s theoretical novelty is limited. The theoretical component primarily revisits established properties of the quantile loss and CRPS rather than introducing a new analytical insight. The main contribution is therefore empirical and integrative rather than theoretical. However, tabular learning with transfer and probabilistic pretraining is a practically relevant and desired area, and large-scale empirical evidence of this kind would likely attract interest from the readers, particularly those working on uncertainty-aware or data-efficient tabular modeling.

**Broader Impact Concerns:**

No specific concerns regarding the ethical implications.

**Claims And Evidence:**

Yes

**Claims Explanation:**

1. **Clear connection between motivation and solution**: The task that the paper wants to solve is clearly formulated in Section 3, and it is precisely addressed through Section 4 with a simple theoretical background in Theorem 1. Specifically, a transformer-based any-quantile learning approach is presented with a simple convergence analysis in Section 4.1.

2. **Principled probabilistic framing**: The connection between the quantile regression objective and the continuous ranked probability score (CRPS) is correctly established, providing a coherent probabilistic interpretation of the proposed loss. The proof sketch leading to the identification of the true inverse conditional CDF as the minimizer is accurate and aligns with prior quantile regression theory.

3. **Strong empirical results**: Most importantly, the authors claimed and demonstrated the effect of transfer learning in NIAQUE with an appropriate experimental design. The proposed confidence-interval-based feature importance offers a simple yet interpretable post-hoc analysis. While not as rigorous as SHAP or integrated gradients, the strong correlation between CI width ranking and SHAP-based importance across multiple datasets is convincing and adds practical value.

**Requested Changes:**

There are a few limitations of the paper that could improve the paper's readability.

1. **Reporting the model size**: While the paper describes the overall model architecture of the models, including NIAQUE and the transformer baseline, reporting the overall model size by parameter counts would improve the clarity of the paper.

2. **Table designs for readability**: The metrics reported in the paper have several different ways to interpret them, such as having lower values is better, lower absolute values are better, or higher the better in Table 1. Thus, implying the best numbers in each column could provide better readability.

3. **More neural models/methods in Table 1**: The results in Table 1 are the main experiments on the effect of transfer learning and effectiveness on a large-scale dataset when trained from scratch. While there are four baselines listed, XGBoost, LightGBM, CatBoost, and Transformer, comparisons on several modern neural tabular models are not sufficiently addressed here. As in Table 3, the paper would benefit from comparing with more neural tabular models, such as FT-Transformer, as a baseline for tabular data.

---

> ### Author Response · Authors · 2025-12-04
> **Response to Reviewer sz4U**
>
> We sincerely thank you for constructive feedback and careful review. We have addressed all points as detailed below:
>
> > Reporting the model size
>
> We have added parameter counts for all deep learning models in the paper. These details are now included in the Implementation Details section for clarity:
> * NIAQUE: ~28M parameters
> * Transformer baseline: ~16M parameters
> * FT-Transformer: ~12M parameters
>
> > Table designs for readability:
>
> We agree that the varied interpretation of metrics (lower is better vs. higher is better) could be confusing. We have now bolded the best performing values in each column for Table 1 to improve readability and make comparisons more intuitive.
>
> > More neural models/methods in Table 1:
>
> We appreciate this suggestion to strengthen our experimental evaluation. We have added FT-Transformer (Gorishniy et al., 2021) to Table 1, as you requested. This serves as a strong modern neural baseline specifically designed for tabular data with feature-wise tokenization and attention mechanisms. FT-Transformer outperforms the vanilla Transformer baseline, making it a more competitive comparison. Our results show that NIAQUE-Global still achieves superior performance compared to FT-Transformer across all metrics (SMAPE, MAE, RMSE, Coverage, and CRPS).

---

### Review · Reviewer_2p9A · 2025-11-02

**Summary Of Contributions:**

The paper proposes NIAQUE as a probabilistic tabular regressor. It conditions on a quantile q to learn the inverse conditional CDF and uses a training objective tied to CRPS. The work further suggests a prototype-aggregating encoder as an alternative to Transformer-based regressors.
The authors construct a 101-dataset corpus for evaluating tabular regression models with pretraining. NIAQUE is shown to outperform tree/transformer baselines and its quantile outputs enable uncertainty-aware feature-importance scoring.

### Strengths
* Probabilistic regression and transfer across heterogeneous tabular datasets is a relevant and underexplored research question.
* The suggested any-quantile setup combined with and efficient encoder-decoder models is a novel and practical approach for probabilistic regression over tables.
* The experimental evaluation covers 101 datasets. This collection of regression benchmark is an additional contribution and may benefit future work in this direction.

### Weaknesses
* Table 1 reports metrics “aggregated across datasets at sample level”. This may overweight larger datasets even after capping. Additionally reporting per-dataset results in the appendix would provide a more fine-grained picture.
* The paper lacks details on how/whether categorical feature columns are used by NIAQUE. Are these omitted or somehow mapped to numerical features?

Overall I believe the presented work is technically sound and interesting to the TMLR audience and should be accepted if the mentioned weaknesses are addressed.

**Audience:**

Yes

**Audience Explanation:**

Effective pretraining of tabular regression models is of interest to the TMLR audience and the paper contributes novel methods and evaluation datasets for this research direction.

**Claims And Evidence:**

Yes

**Claims Explanation:**

The experimental evaluation is convincing and adequate ablations are provided.

**Requested Changes:**

I think the following changes are critical:
1. Add more fine-grained per-dataset results underlying Table 1 in the appendix, or at least provide the macro average to balance the weights of each dataset.
2. Clarify if/how categorical features are leveraged by NIAQUE

---

> ### Author Response · Authors · 2025-12-04
> **Response to Reviewer 2p9A**
>
> We sincerely thank you for constructive feedback and careful review. We have addressed all points as detailed below:
>
> > Add more fine-grained per-dataset results underlying Table 1
>
> We have added comprehensive per-dataset results for all 101 datasets in Appendix D.1. This includes individual performance metrics (SMAPE, MAE, RMSE, Coverage, CRPS) for each dataset, allowing readers to examine fine-grained performance patterns. As an example, we found that NIAQUE was better than CatBoost-Local on 65 datasets in terms of SMAPE, and on 85 datasets in terms of CRPS.
>
> > Clarify if/how categorical features are leveraged by NIAQUE
>
> We have clarified our handling of categorical features in Section Neural Encoder-Decoder Architecture. Specifically:
>
> *NIAQUE processes both continuous and categorical features in a unified manner. Categorical features are first label-encoded to integers during preprocessing. For each feature in the observation vector $x$, NIAQUE incorporates both its raw integer value and a learnable embedding based on its feature index (position in the feature vector).*

---

### Review · Reviewer_qBhK · 2025-11-19

**Summary Of Contributions:**

The paper investigates transfer learning for probabilistic regression for tabular data. The authors propose NIAQUE -- an encoder-decoder neural network that enables pre-training followed by task-specific fine-tuning in probabilistic regression for tabular data.

Strengths:
* Introduction of Tabular Regression Set 101, a collection of 101 publicly available regression datasets, is a valuable contribution that may set a standard benchmark in the field.
* The proposed approach, NIAQUE, outperforms other baselines. Notably, global learning (slightly) helps in cross-dataset learning (Tab1) and pre-training is effective, especially when downstream task data is scarce (Tab2)

Weaknesses:
* The proposed architecture is not properly ablated it is unclear to my why each feature block or quantile modulation block have the proposed structure, why is prototype layer needed, etc. (Appendix J only searches over the hyperparameters given the proposed architecture, eg number of blocks, width, dropout probability, etc.)

**Additional Comments:**

I am not very familiar with probabilistic regression and tabular data literature. Therefore, my confidence is quite low.

**Audience:**

Yes

**Audience Explanation:**

NIAQUE seems like a state-of-the-art method for transfer learning that should be interesting to many people working on that. Also, the proposed benchmark may be useful for the community.Critical:
* proper ablations of the proposed architecture are necessary (see weakness)

Non-critical changes:
* The authors notoriously use `\citet{}` instead of `\citep` making the cited papers' authors blend into the text.

**Broader Impact Concerns:**

Broader Impact Statement is not present in the paper and I agree that it is not needed for this paper.

**Claims And Evidence:**

Yes

**Claims Explanation:**

The claims are properly supported by experimental evidence.

**Requested Changes:**

Critical:
* proper ablations of the proposed architecture are necessary (see weakness)

Non-critical changes:
* The authors notoriously use `\citet{}` instead of `\citep` making the cited papers' authors blend into the text.

---

> ### Author Response · Authors · 2025-12-04
> **Response to Reviewer qBhK**
>
> We sincerely thank you for constructive feedback and careful review. We have addressed all points as detailed below:
>
> > The proposed architecture is not properly ablated it is unclear to my why each feature block or quantile modulation block have the proposed structure, why is prototype layer needed, etc. (Appendix J only searches over the hyperparameters given the proposed architecture, eg number of blocks, width, dropout probability, etc.)
>
> Thank you for this insightful comment. We have added prototype layer ablation as you requested. Specifically, we ablated the proto-type layer by removing the prototype connection in equation (14). We summarized this additional experimental result in the revised manuscript as follows:
>
> *Finally, we performed the ablation of the prototype layer by disabling the prototype connection in eq. (14) We found that this drastically degrades the performance: SMAPE rises from 22.1 to 94.612, AAD from 0.367 to 2.585, and RMSE from 0.787 to 3.432, proving the effectiveness of the prototype layer.*
>
> We have clarified the structure of feature encoder block and the role of prototype connection in the discussion of equations (11-14):
>
> *b) Eq. (14) implements interactions between features (akin to attention, but with linear compute cost) and introduces an inductive bias by enforcing a delta-mode constraint, ensuring that feature contributions are only relevant when they deviate from the existing observation embedding, $p_{r-1}$*
>
> We have further clarified the structure of the quantile modulation block in the revised manuscript:
>
> *Its primary function is to implement the any-quantile functionality by injecting the quantile value inside the MLP block using FiLM modulation principle (Perez et al., 2018).*
>
> As such, this block cannot be removed, because then the any-quantile functionality will be lost and the training / inference will not be functional.
>
> > The authors notoriously use \citet{} instead of \citep making the cited papers' authors blend into the text.
>
> We have carefully revised all citations throughout the paper, replacing \citet{} with \citep{} where appropriate to improve readability. Citations are now properly integrated into the text flow without author names unnecessarily blending into sentences.

---

### Decision · Action_Editor_mDnp · 2026-01-15

**Recommendation:** Accept as is

**Audience:**

Yes

**Audience Explanation:**

The paper addresses probabilistic regression and transfer learning for tabular data, a relevant and underexplored area of interest to the TMLR community. Its combination of a scalable neural architecture, probabilistic modeling with proper scoring rules, and large-scale empirical evaluation across 101 datasets makes the findings particularly relevant to researchers working on tabular learning, uncertainty estimation, and data-efficient modeling.

**Claims And Evidence:**

Yes

**Claims Explanation:**

The claims are supported by accurate, convincing, and clearly presented evidence. The paper provides a comprehensive empirical evaluation across 101 diverse tabular regression datasets using appropriate accuracy, calibration, and proper scoring metrics (including CRPS). The experimental design directly tests the stated claims on transfer learning, scalability, and probabilistic modeling. Reviewer concerns regarding ablations, per-dataset results, baselines, and categorical feature handling were fully addressed in the revision.